# Revealing global stoichiometry conservation architecture in cells from Raman spectral patterns

Ken-ichiro F Kamei[1]*, Koseki J Kobayashi-Kirschvink[2], Takashi Nozoe[1,3,4], Hidenori Nakaoka[5], Miki Umetani[6], Yuichi Wakamoto[1,3,4]*

[1]Department of Basic Science, Graduate School of Arts and Sciences, The University of Tokyo, Tokyo, Japan; [2]Department of Medicine, The University of Chicago, Chicago, United States; [3]Research Center for Complex Systems Biology, The University of Tokyo, Tokyo, Japan; [4]Universal Biology Institute, The University of Tokyo, Tokyo, Japan; [5]Department of Optical Imaging, Advanced Research Promotion Center Tokushima University, Tokushima, Japan; [6]Department of Biology, New York University, New York, United States

## eLife Assessment

This paper reports the **fundamental** finding of how Raman spectral patterns correlate with proteome profiles using Raman spectra of *E. coli* cells from different physiological conditions and found global stoichiometric regulation on proteomes. The authors' findings provide **compelling** evidence that stoichiometric regulation of proteomes is general through analysis of both bacterial and human cells. In the future, similar methodology can be applied on various tissue types and microbial species for studying proteome composition with Raman spectral patterns.

*For correspondence:
kenichiro_kamei@cell.c.u-tokyo.ac.jp (KFK);
cwaka@mail.ecc.u-tokyo.ac.jp (YW)

**Abstract** Cells can adapt to various environments by changing their biomolecular profiles while maintaining physiological homeostasis. What organizational principles in cells enable the simultaneous realization of adaptability and homeostasis? To address this question, we measure Raman scattering light from *Escherichia coli* cells under diverse conditions, whose spectral patterns convey their comprehensive molecular composition. We reveal that dimension-reduced Raman spectra can predict condition-dependent proteome profiles. Quantitative analysis of the Raman-proteome correspondence characterizes a low-dimensional hierarchical stoichiometry-conserving proteome structure. The network centrality of each gene in the stoichiometry conservation relations correlates with its essentiality and evolutionary conservation, and these correlations are preserved from bacteria to human cells. Furthermore, stoichiometry-conserving core components obey growth law and ensure homeostasis across conditions, whereas peripheral stoichiometry-conserving components enable adaptation to specific conditions. Mathematical analysis reveals that the stoichiometrically constrained architecture is reflected in major changes in Raman spectral patterns. These results uncover coordination of global stoichiometric balance in cells and demonstrate that vibrational spectroscopy can decipher such biological constraints beyond statistical or machine-learning inference of cellular states.

## Introduction

Biological cells can change their gene expression and metabolic profiles globally to adapt to their biological contexts and external conditions, while maintaining the homeostasis of their core physiological states. The simultaneous realization of adaptability and homeostasis is a hallmark of biological systems and is assumed to be a system-level property of gene expression profiles in cells (*Waddington, 1957*; *Waddington, 1959*). However, understanding the underlying organizational principles in comprehensive gene expression profiles remains to be a fundamental problem in biology.

Vibrational spectroscopy such as Raman spectroscopy might help us investigate such principles in gene expression profiles. Raman spectroscopy is a light scattering technique that measures energy shifts of light caused by interaction with sample molecules. Raman spectra are obtainable non-destructively even from biological samples such as individual cells. In principle, cellular Raman spectra are optical signatures conveying comprehensive molecular composition of targeted cells (*Goodacre*

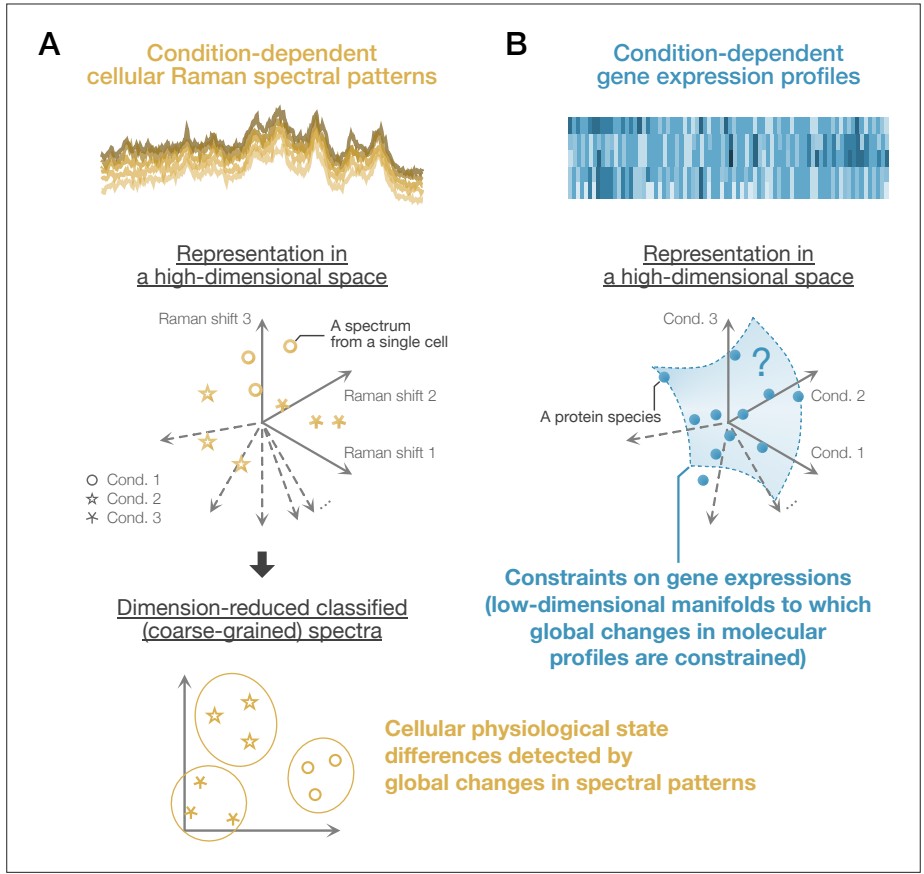

**Figure 1.** Cellular physiological state differences detected by Raman spectral global patterns and gene expression profiles. (**A**) Condition-dependent cellular Raman spectral patterns. Raman spectra obtained from cells reflect their molecular profiles. Therefore, systematic differences in global spectral patterns may indicate their physiological states. A Raman spectrum from each cell can be represented as a vector and a point in a high-dimensional Raman space. If condition-dependent differences exist in the spectral patterns, appropriate dimensional reduction methods allow us to classify the spectra and detect cellular physiological states in a low-dimensional space. (**B**) Condition-dependent gene expression profiles. Global gene expression profiles (proteomes and transcriptomes) are also dependent on conditions. For each gene, we can consider a high-dimensional vector whose elements represent expression levels under different conditions. It has been suggested that these expression-level vectors are constrained to some low-dimensional manifolds (*Eisen et al., 1998*; *Segal et al., 2003*; *Bergmann et al., 2003*; *Keren et al., 2013*; *You et al., 2013*; *Kaneko et al., 2015*; *Hui et al., 2015*; *Heimberg et al., 2016*; *Biswas et al., 2017*; *Husain and Murugan, 2020*; *Sato and Kaneko, 2020*). This study characterizes the statistical correspondence between dimension-reduced Raman spectral patterns and gene expression profiles. Analyzing the correspondence, we reveal a stoichiometry conservation principle that constrains gene expression profiles to low-dimensional manifolds.

*et al., 1998*; *Huang et al., 2004*; *Ichimura et al., 2014*; *Germond et al., 2018*). Furthermore, no prior treatments, such as staining and tagging, are necessary to obtain cellular Raman spectra. However, although some biomolecules have separable and intense Raman signal peaks, Raman spectra of most biomolecules overlap and are masked by signals of other molecules due to the diversity and complexity of molecular compositions of cells. Therefore, it is impractical to comprehensively determine the amounts of biomolecules by spectral decomposition.

Despite the intractability of spectral decomposition, reconstruction of comprehensive molecular profiles may be achievable by analyzing detectable global spectral patterns (*Figure 1A*), thanks to effective low dimensionality of changes in molecular profile of targeted cells (*Eisen et al., 1998*; *Segal et al., 2003*; *Bergmann et al., 2003*; *Keren et al., 2013*; *You et al., 2013*; *Kaneko et al., 2015*; *Hui et al., 2015*; *Heimberg et al., 2016*; *Biswas et al., 2017*; *Husain and Murugan, 2020*; *Sato and Kaneko, 2020*; *Figure 1B* and *Appendix 1—figure 1*). Indeed, it has been demonstrated that condition-dependent global transcriptome profiles of cells can be inferred from cellular Raman spectra based on their statistical correspondence (*Kobayashi-Kirschvink et al., 2018*; *Kobayashi-Kirschvink et al., 2024*). Importantly, this Raman-spectroscopic transcriptome inference was possible from dimension-reduced Raman spectra. Therefore, dominant changes in global Raman spectral patterns may contain vital information about the constraints on the molecular profiles in cells; an inspection of their correspondence might give us insights into architectural principles of omics profiles and biological foundation for global omics inference from spectral patterns (*Appendix 1—figure 1*).

In this report, we first reveal that, in addition to transcriptomes, condition-dependent proteome profiles of *Escherichia coli* are predictable from cellular Raman spectra. Next, we scrutinize the correspondence between Raman and proteome data, identifying several stoichiometrically conserved groups (SCGs) whose expression tightly correlates with the major changes in cellular Raman spectra. Finally, we reveal that the stoichiometry conservation centrality of each gene correlates with its essentiality, evolutionary conservation, and condition specificity of gene expression levels, which turns out general across different omics layers and organisms.

## Results

### Statistical correspondence between Raman spectra and proteomes

To examine the correspondence between Raman spectra and proteomes in *E. coli*, we reproduced 15 environmental conditions for which absolute quantitative proteome data are already available (*Schmidt et al., 2016*) and measured Raman spectra of *E. coli* cells under those conditions (*Figure 2A and B*). The culture conditions we adopted include (i) exponential growth phase in minimal media with various carbon sources, (ii) exponential growth phase in rich media, (iii) exponential growth phase with various stressors, and (iv) stationary phases (*Appendix 1—table 1*). We measured Raman spectra of single cells sampled from each condition and focused on the fingerprint region of biological samples, where the signals from various biomolecules concentrate (spectral range of 700–1800 cm$^{-1}$, *Figure 2B* and *Appendix 1—figure 2*). The Raman spectra were classified on the basis of the environmental conditions using a simple linear classifier, linear discriminant analysis (LDA) (*Goodacre et al., 1998*; *Huang et al., 2004*; *De Bie et al., 2005*; *Figure 2C–E* and *Appendix 1—figure 1*). This classifier calculates the most discriminatory axes by maximizing the ratio of between-condition variance to within-condition variance and reduces the dimensions of Raman data to $m - 1$, where $m = 15$ is the number of conditions (see 'Experimental methods, data acquisition, and data analyses' in Materials and methods and Section 2.1 in Appendix).

The result shows that Raman spectral points from different environmental conditions are distinguishable in the $(m - 1)$-dimensional LDA space (*Figure 2C–E*). For example, the first and second LDA axes clearly distinguish the conditions 'LB' and 'stationary3days' (*Figure 2C*), and the third axis distinguishes 'Glucose42C' and 'GlycerolAA' (*Figure 2D*). Notably, the first principal axis LDA1 correlated with growth rate significantly (Pearson correlation $r = 0.81 \pm 0.09$, *Appendix 1—figure 2*). Visualizing the Raman LDA data by embedding them on a two-dimensional plane using t-distributed stochastic neighbor embedding (t-SNE) (*van der Maaten and Hinton, 2008*) confirms that the points for each condition form a distinctive cluster (*Figure 2F*). These results imply that positions in the Raman LDA space reflect condition-dependent differences in cellular physiological states.

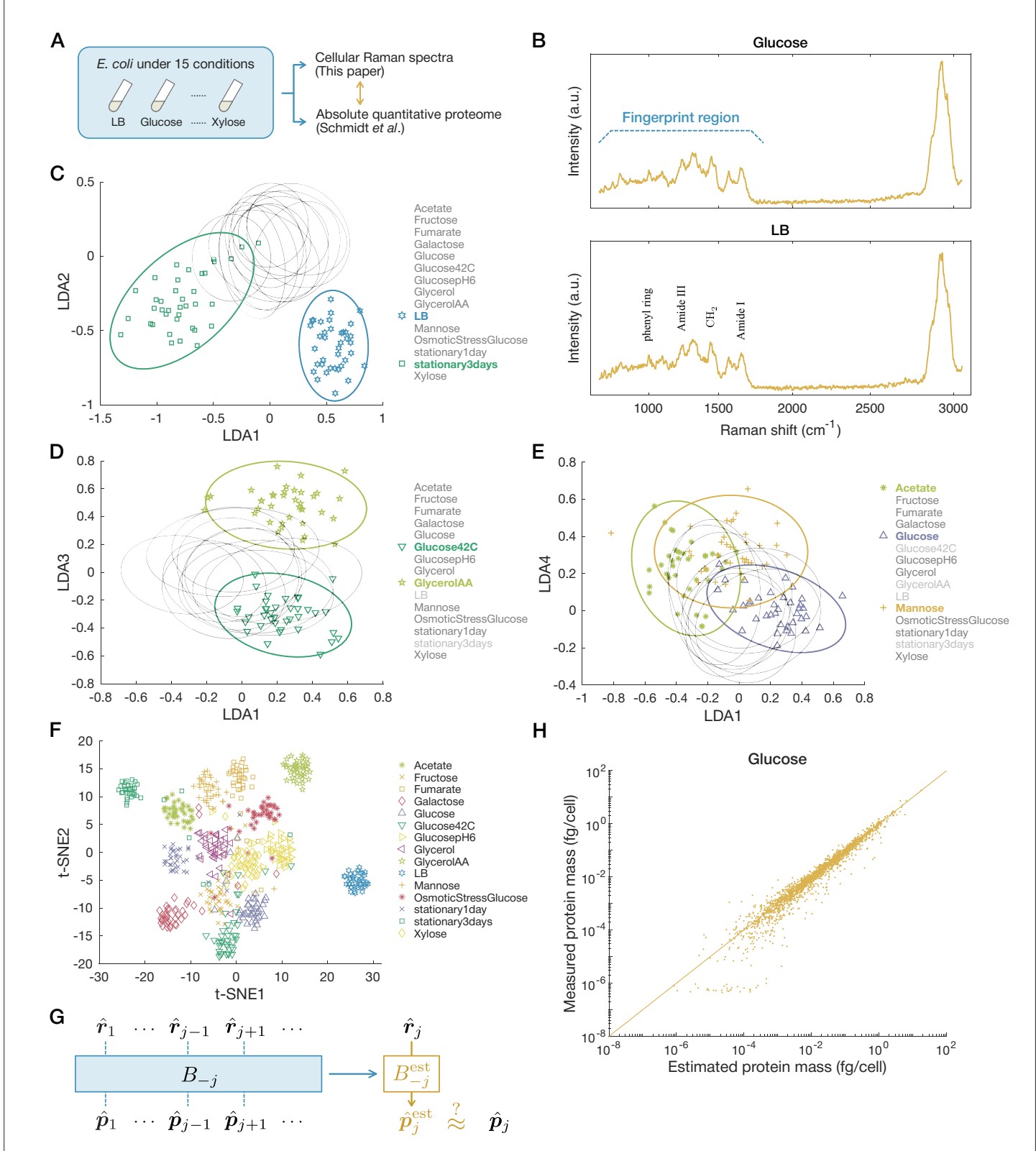

**Figure 2.** Estimation of proteomes from Raman spectra. (**A**) The experimental design. We cultured *E. coli* cells under 15 different conditions and measured single cells' Raman spectra. We then examined the correspondence between the measured Raman spectra and the absolute quantitative proteome data reported by ***Schmidt et al., 2016***. (**B**) Representative Raman spectra from single cells, one from the 'Glucose' condition, and the other from the 'LB' condition. The fingerprint region and representative peaks are annotated. (**C–E**) Cellular Raman spectra in linear discriminant analysis

*Figure 2 continued on next page*

*Figure 2 continued*

(LDA) space. The dimensionality of the spectra is reduced to $14\,(=15-1)$. Each point represents a spectrum from a single cell, and each ellipse shows the 95% concentration ellipse for each condition. Their projections to the LDA1-LDA2 plane (**C**), the LDA1-LDA3 plane (**D**), and the LDA1-LDA4 plane (**E**) are shown. (**F**) Visualization of the 14-dimensional LDA space embedded in two-dimensional space with t-distributed stochastic neighbor embedding (t-SNE). (**G**) The scheme of leave-one-out cross-validation. The Raman and proteome data of one condition (here $j$) are excluded, and the matrix $B$ is estimated using the data of the rest of the conditions as $B_{-j}^{\mathrm{est}}$. The proteome data under the condition $j$ is estimated from the Raman data $\hat{r}_j$ with $B_{-j}^{\mathrm{est}}$ and compared with the actual data to calculate estimation errors. (**H**) Comparison of measured and estimated proteome data. The plot for the 'Glucose' condition is shown as an example. Each dot corresponds to one protein species. The straight line indicates $x=y$. Proteins with negative estimated values are not shown.

We next asked whether these Raman spectral differences in the LDA space could be linked to the different proteome profiles (*Appendix 1—figure 1*). To examine this, we hypothesized linear correspondence between the $n$-dimensional proteome column vector $\hat{p}_j$, where $n = 2058$ is the number of protein species in the proteome data, and the low-dimensional ($(m-1)$-dimensional) Raman column vector $\hat{r}_j$ in condition $j$,

$$\hat{p}_j = B \cdot \begin{bmatrix} 1 \\ \hat{r}_j \end{bmatrix}. \tag{1}$$

**Table 1.** List of scalars, vectors, and matrices in the main text.

Scalars, vectors, and matrices in the main text are listed with their sizes and descriptions. $m$ is the number of conditions, and $n$ is the number of protein species. ($m = 15$ and $n = 2058$ in the main text.) Note that the notation summarized in this table differs in some respect from that in Materials and methods and Appendix.

| | Size (#columns × #rows) | Description |
|---|---|---|
| $\hat{r}_j$ <br> $(j = 1, \ldots, m)$ | $(m-1) \times 1$ (vector) | Mean LDA Raman profile <br><br> of single cells under condition $j$ |
| $\hat{p}_j$ <br> $(j = 1, \ldots, m)$ | $n \times 1$ (vector) | Proteome profile <br> of cell population under condition $j$ |
| $B$ <br> $= \begin{bmatrix} b_0 & \cdots & b_{m-1} \end{bmatrix}$ <br> $= (b_{ik})_{1 \le i \le n, 0 \le k \le m-1}$ | $n \times m$ | Set of condition-independent <br> coefficients that linearly connect $\hat{r}_j$ and $\hat{p}_j$ for all conditions $j$ (*Equation 1*) |
| $p_i$ <br> $(i = 1, \ldots, n)$ | $m \times 1$ (vector) | Expression levels of protein species <br> $i$ across $m$ conditions |
| $\cos \theta_{p_i p_j}$ <br> $= (p_i / \|p_i\|_2) \cdot (p_j / \|p_j\|_2)$ <br> $(i, j = 1, \ldots, n)$ | $1 \times 1$ (scalar) | Stoichiometry (abundance ratio) <br> conservation strength between two protein species $i$ and $j$ (*Figure 4A*) |
| $A = \left( \cos \theta_{p_i p_j} \right)_{1 \le i,j \le n}$ | $n \times n$ | Set of stoichiometry conservation strengths between all pairs of protein species (*Figure 5J*) |
| $d_i = \Sigma_{j=1}^{n} \cos \theta_{p_i p_j}$ <br> $(i = 1, \ldots, n)$ | $1 \times 1$ (scalar) | Stoichiometry conservation centrality of protein species $i$ |
| $g_i = \|p_i\|_1 / \|p_i\|_2$ <br> $(i = 1, \ldots, n)$ | $1 \times 1$ (scalar) | Expression generality of protein species $i$ |

$B$ is an $n \times m$ matrix that connects $\hat{\boldsymbol{p}}_j$ and $\hat{\boldsymbol{r}}_j$. We calculated $\hat{\boldsymbol{r}}_j$ as the average of the low-dimensional LDA Raman data of single cells in condition $j$ since the proteomes were measured for cell populations (**Table 1**).

We conducted leave-one-out cross-validation (LOOCV) to verify this linear correspondence (**Figure 2G**). We excluded one condition (here, $j$) as a test condition and estimated $B$ as $B_{-j}^{\mathrm{est}}$ by simple ordinary least squares (OLS) regression using the data of the rest of the conditions. We thereby estimated the proteome in condition $j$ as $\hat{\boldsymbol{p}}_j^{\mathrm{est}} = B_{-j}^{\mathrm{est}} \cdot \begin{bmatrix} 1 \\ \hat{\boldsymbol{r}}_j \end{bmatrix}$.

The proteome profile estimated using the first four major LDA axes (LDA1–LDA4) agreed well with the actual proteome data under most conditions (**Figure 2H** and **Appendix 1—figure 3**; see 'Raman-proteome statistical correspondence' in Materials and methods for the estimation with all the LDA axes). Changing the condition to exclude, we estimated the proteomes for all the 15 conditions and calculated the overall estimation error by the Euclidean distance $\sum_j \|\hat{\boldsymbol{p}}_j^{\mathrm{est}} - \hat{\boldsymbol{p}}_j\|^2$. The result shows that the overall estimation error is significantly small ($p = 0.00005$ by permutation test; **Fisher, 1935**; **Pitman, 1937**; **Phipson and Smyth, 2010**). Adopting other distance measures does not change the conclusion (**Appendix 1—tables 2 and 3**). These results, therefore, validate the assumption of linear correspondence between cellular Raman spectra and proteomes and confirm that condition-dependent changes in proteomes can be inferred from the corresponding low-dimensional Raman spectra.

## Stoichiometry conservation of proteins in the ISP COG class

Since the dimensionality of the proteome data is significantly higher than that of the Raman data, the result above suggests that changes in proteome profiles are constrained in low-dimensional space. The regression matrix $B$ considered above determines how the proteomes relate to the Raman LDA axes. Therefore, analyzing $B$ should provide some insights into constraints on condition-dependent changes in the proteomes (**Appendix 1—figure 1**).

The $n \times m$ matrix $B$ is represented as $B = \begin{bmatrix} \boldsymbol{b}_0 & \boldsymbol{b}_1 & \cdots & \boldsymbol{b}_{m-1} \end{bmatrix}$, where the $(k+1)$-th column $\boldsymbol{b}_k = \begin{pmatrix} b_{1k} & b_{2k} & \cdots & b_{nk} \end{pmatrix}^\top$ $(0 \le k \le m-1)$ is the collection of coefficients of all $n$ proteins for the $k$-th LDA axis (**Table 1**). In the case of $k = 0$, the coefficients are constant terms. We first asked whether any shared features might exist in the coefficients of $B$ depending on biological functions of corresponding proteins. We then classified the proteins according to functional annotations of Clusters of Orthologous Group (COG) classes (**Tatusov et al., 1997**; **Tatusov et al., 2003**; **Galperin et al., 2015**) and found that, for many proteins belonging to the 'information storage and processing' (ISP) COG class, the coefficients corresponding to different LDA axes are approximately proportional to the constant terms, i.e., $b_{lk} \approx c_k b_{l0}$, where $l$ is the index of an ISP COG class protein species and $c_k$ is the proportionality constant common to many ISP COG class protein species for the $k$-th LDA axis (**Figure 3A**). The ISP COG class contains various proteins involved in processing genetic information such as translation, transcription, DNA replication, and DNA repair (**Schmidt et al., 2016**). Simple calculations show that these proportionality relationships imply that proteins in the ISP COG class conserve their mutual abundance ratios, i.e., stoichiometry, irrespective of environmental conditions (see 'Characterizing an SCG by analyzing the Raman-proteome correspondence matrix' in Materials and methods).

Since this is an implication from the Raman-proteome correspondence, we next examined the stoichiometry conservation only with the proteome data, evaluating the expression levels with Pearson correlation coefficients for all the pairs of the conditions for each COG class (**Figure 3B**). For the ISP COG class, the correlation coefficients were close to 1, whereas those for the other COG classes were significantly smaller depending on condition pairs. We also evaluated the coordination of gene expression patterns within each COG class using cosine similarity and obtained consistent results (**Appendix 1—figure 4**). Therefore, stoichiometry conservation is stronger in the ISP COG class than in the other COG classes. Remarkably, neither shared transcription factors nor chromosome locations can account for the observed stoichiometry conservation of many protein pairs. Indeed, although the ISP COG class shows highly coordinated expression patterns (**Figure 3C**) compared to the non-ISP COG class (**Figure 3D**), the gene loci are not chromosomally clustered in either example. Additionally,

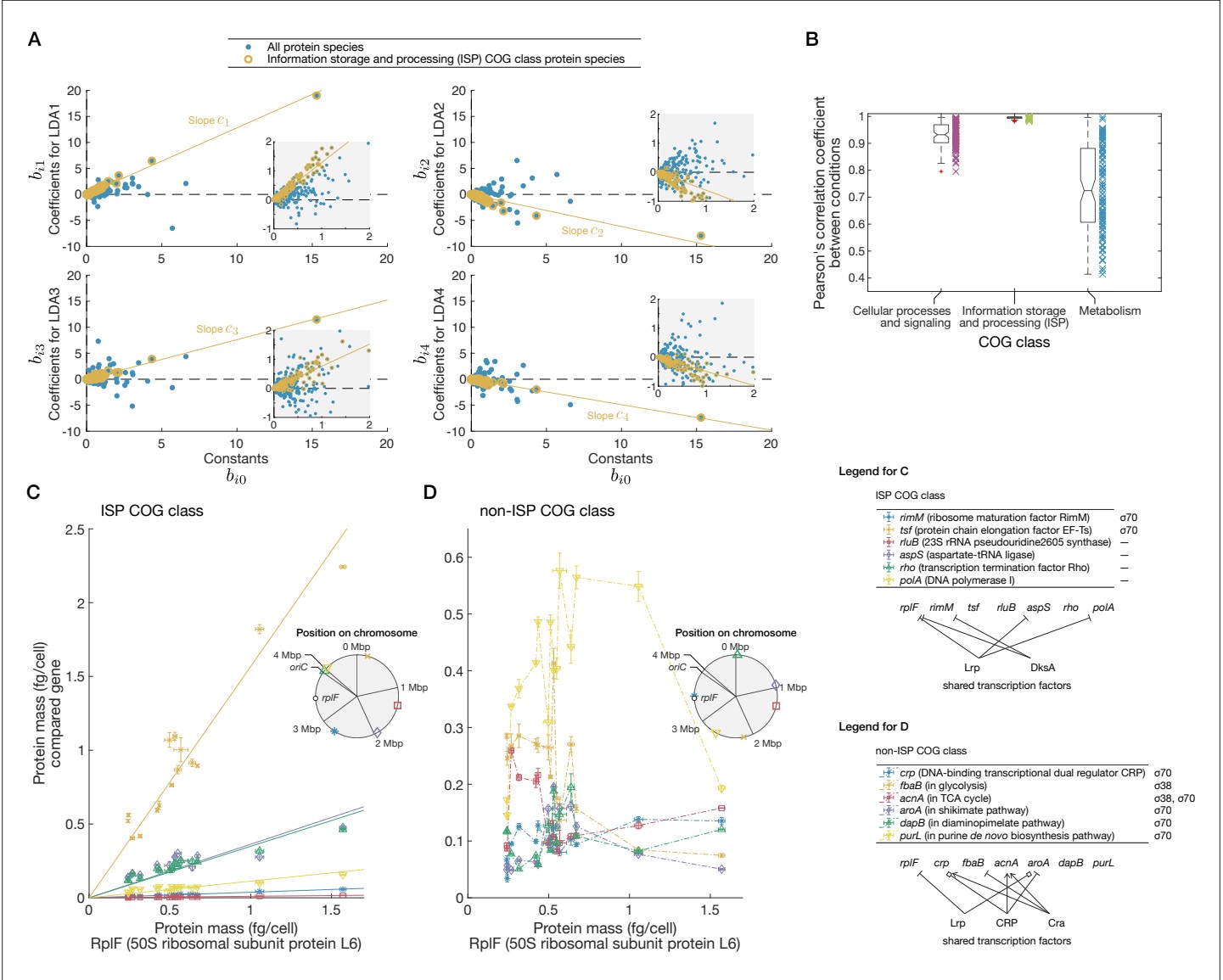

**Figure 3.** A stoichiometrically conserved protein group identified by an analysis of the Raman-proteome coefficient matrix. (**A**) Scatterplots of Raman-proteome transformation coefficients. The horizontal axes are constant terms ($b_0$) in all the plots. The vertical axis is coefficients for LDA1 ($b_1$), LDA2 ($b_2$), LDA3 ($b_3$), or LDA4 ($b_4$) in each plot. The proteins in the information storage and processing (ISP) Clusters of Orthologous Group (COG) class are indicated in yellow. Yellow solid straight lines are least squares regression lines passing through the origins for the ISP proteins. Insets are enlarged views of area around the origins. In this figure, we used the average of $B_{-i}^{\mathrm{est}}$ as an estimate of $B$. (**B**) Similarity of expression patterns between culture conditions for each COG class. We divided the proteome into COG classes (***Tatusov et al., 2003***; ***Galperin et al., 2015***) and calculated Pearson correlation coefficient of expression patterns for all the combinations of culture conditions. Since the data are from 15 conditions, there are 105 (=15·14/(2·1)) points for each COG class in the graph. The box-and-whisker plots summarize the distributions of the points. The lines inside the boxes denote the medians, the top and bottom edges of the boxes do the 25th percentiles and 75th percentiles, respectively. The numbers of protein species are 376 for the Cellular Processes and Signaling COG class, 354 for the ISP COG class, and 840 for the Metabolism COG class. See ***Appendix 1—figure 4*** for the evaluation with Pearson correlation coefficient of log abundances and with cosine similarity. ***Appendix 1—figure 4*** also contains figures directly showing expression-level changes of different protein species across conditions for each COG class. (**C**) Examples of stoichiometry-conserving proteins in the ISP COG class. The horizontal axis represents the abundance of RplF under 15 conditions, and the vertical axis represents those of several ISP COG class proteins. These proteins are also contained in the *homeostatic core* defined later (see ***Figure 4***). The solid straight lines are linear regression lines with an intercept of zero. (**D**) Examples of abundance ratios of non-ISP COG class proteins. The horizontal axis represents the abundance of RplF under 15 conditions, and the vertical axis represents those of compared non-ISP COG class proteins. Crp belongs to the Cellular Processes and Signaling COG class; the other proteins belong to the Metabolism COG class. In both (**C**) and (**D**), we selected the proteins expressed from distant loci on the chromosome. All sigma factors participating in the regulation of the proteins examined in (**C**) and (**D**) are listed on the right of the gene name legends. All transcription factors known to regulate multiple genes listed here are shown in the right diagrams. Arrows show activation; bars represent

*Figure 3 continued on next page*

*Figure 3 continued*

inhibition; and squares indicate that a transcription factor activates or inhibits depending on other factors. The information on gene regulation and functions was obtained from EcoCyc (**Keseler et al., 2017**) in August 2022. The error bars are standard errors calculated by using the data of **Schmidt et al., 2016**. The insets show the positions of the genes on the *E. coli* chromosome determined based on ASM75055v1.46 (**Howe et al., 2020**). No genes are in the same operon.

the similarity/dissimilarity of expression patterns cannot easily be inferred from transcription factor regulation patterns. These results imply multi-level regulation of their abundance.

We consulted other public quantitative proteome data of *Mycobacterium tuberculosis* (**Schubert et al., 2015**), *Mycobacterium bovis* (**Schubert et al., 2015**), and *Saccharomyces cerevisiae* (**Lahtvee et al., 2017**) under environmental perturbations and consistently found strong stoichiometry conservation of the ISP COG class (**Appendix 1—figure 4**). Furthermore, the same trend was observed for the genotype-dependent expression changes in *E. coli* proteomes (**Schmidt et al., 2016**; **Appendix 1—figure 4**).

## Identifying SCGs

Inspired by the existence of a large class of proteins that conserves their stoichiometry, we considered a systematic way to extract SCGs without relying on artificial functional classification of COG (**Appendix 1—figure 1**). Focusing only on the proteome data, we evaluated stoichiometry conservation for all the pairs of proteins in the proteome by calculating the cosine similarity of expression patterns (i.e. all $\cos\theta_{\boldsymbol{p}_i\boldsymbol{p}_j} := \left(\boldsymbol{p}_i/\|\boldsymbol{p}_i\|_2\right) \cdot \left(\boldsymbol{p}_j/\|\boldsymbol{p}_j\|_2\right)$ in **Figure 4A** and **Table 1**, where each element of the $m$-dimensional vector $\boldsymbol{p}_i$ denotes the expression level of protein species $i$ under one of the $m$ conditions), and extracted groups in each of which the component proteins exhibit coherent expression change patterns by setting a high threshold of cosine similarity ($\geq 0.995$, **Figure 4B**; see 'Direct characterization of SCGs in omics data' in Materials and methods for details).

The largest SCG (SCG 1) included many proteins in the ISP COG class (91 out of 191 SCG 1 members), such as ribosomal proteins and RNA polymerase, and also proteins in the other COG classes (**Figure 4B**, **Appendix 1—table 4**). We call this largest SCG *homeostatic core*, as it constitutes the largest stoichiometry-conserving unit in cells. We found that the abundance of each protein in the homeostatic core (SCG 1) increased approximately linearly with the growth rate in each condition (**Figure 4C**). This relationship is reminiscent of the growth law: The total ribosomal contents for translation increase linearly with growth rate (**Neidhardt and Magasanik, 1960**; **Scott et al., 2010**; **Bremer and Dennis, 2008**). The linear increase in the abundance of each protein in **Figure 4C** indicates that the growth law is valid even at the single-gene level for a large class of ribosomal and non-ribosomal proteins in the homeostatic core (**Appendix 1—figure 5**) (see Section 3.1 in Appendix).

Though not evenly distributed, the gene loci of the proteins in the homeostatic core are scattered throughout the chromosome (**Figure 4D**). Therefore, localization of gene loci to a single or a small number of operons is not likely a cause of the observed stoichiometry conservation.

The proteins in the second largest SCG (SCG 2) are expressed at high levels in the fast growth conditions, especially in the 'LB' condition (**Figure 4C**). The SCG 2 includes many proteins in the metabolism COG class (21 out of 26 SCG 2 members) (**Appendix 1—table 5**), and their abundance increases approximately exponentially with growth rate (**Figure 4C**). We also identified other condition-specific small SCGs, such as a group most expressed in the 'GlycerolAA' condition (SCG 3) (**Appendix 1—table 6**), a group mainly expressed in the 'Fructose' condition (SCG 4) (**Appendix 1—table 7**), and a group most expressed in the stationary phase conditions (SCG 5) (**Appendix 1—table 8**; **Figure 4C**).

## Biological relevance of stoichiometry conservation

To understand the overall strength of stoichiometry conservation of the proteins in the different SCGs, we calculated the sum of cosine similarity, $d_i = \sum_j \cos\theta_{\boldsymbol{p}_i\boldsymbol{p}_j}$, for each protein species $i$, where $\cos\theta_{\boldsymbol{p}_i\boldsymbol{p}_j}$ is cosine similarity between the $m$-dimensional expression level vectors of protein $i$ and protein $j$ (**Figure 4A**), and the sum is taken over all the protein species (see 'Global proteome structures based on stoichiometric balance' in Materials and methods). We refer to $d_i$ as 'stoichiometry conservation centrality' (**Table 1**).

The proteins in the homeostatic core had high centrality scores (**Figure 5A**). Therefore, these proteins tend to have more connections with other proteins in terms of stoichiometry conservation.

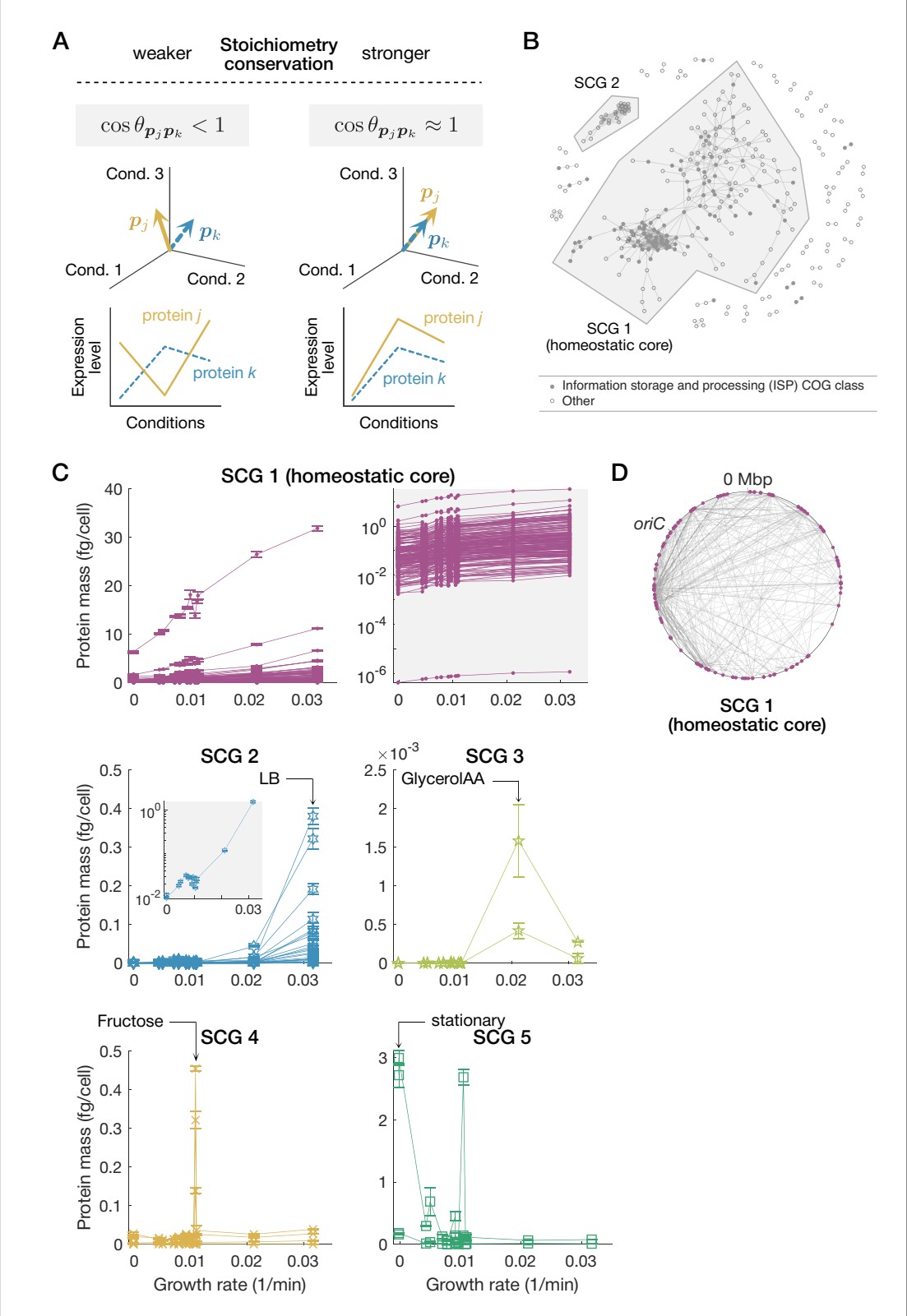

**Figure 4.** Extracting stoichiometrically conserved groups (SCGs) from proteome data. (**A**) Quantifying stoichiometry conservation by cosine similarity. We consider an $m$-dimensional expression vector for each protein species whose elements represent its abundance under different conditions. The cosine similarity between the $m$-dimensional expression vectors of two protein species becomes nearly 1 when they conserve mutual stoichiometry strongly across conditions, whereas lower than 1 when their expression patterns are incoherent. (**B**) Extracted SCGs. We extracted proteins with high

*Figure 4 continued*

cosine similarity relationships. Each node represents a protein species. An edge connecting two nodes represents that the expression patterns of the two connected protein species have high cosine similarity exceeding a threshold of 0.995. Proteins that have no edge with the other proteins are not shown. The largest and the second largest protein groups, which we refer to as SCG 1 and SCG 2, respectively, are indicated by shaded polygons. (**C**) Expression patterns of the extracted SCGs. The horizontal and vertical axes represent growth rate and protein abundance, respectively. Line-connected points represent expression-level changes of different protein species across conditions. SCG 1 (homeostatic core) is shown in two ways: the left panel with a linear-scaled vertical axis and the right panel with a log-scaled vertical axis. The inset for SCG 2 shows the total abundances of SCG 2 proteins with a log-scaled vertical axis. Error bars are standard errors. (**D**) The gene loci of the homeostatic core (SCG 1) proteins on the chromosome. Magenta dots are nodes (genes), and gray lines are edges (high cosine similarity relationships). We determined the gene loci based on ASM75055v1.46 (*Howe et al., 2020*).

On the other hand, the proteins in the condition-specific SCGs tend to have low centrality scores among all the proteins (*Figure 5A*), which suggests that their stoichiometry conservation is localized within each SCG.

The stoichiometry conservation centrality is biologically relevant because it correlates with gene essentiality. Fractions of essential genes almost monotonically decrease with the ranks of centrality score (*Figure 5B* and *Appendix 1—figure 6*). We also noted that genes with high centrality scores have more orthologs determined by OrthoMCL-DB (*Chen et al., 2006*) across the three domains of life (*Figure 5C* and *Appendix 1—figure 6*). Likewise, genes with many orthologs tend to have higher centrality scores (*Figure 5C* and *Appendix 1—figure 6*). Therefore, the stoichiometry conservation in cells correlates with the evolutionary conservation of proteins.

To determine if the correlation of stoichiometry conservation centrality with gene essentiality and evolutionary conservation is general, we analyzed the transcriptome data from other organisms and found comparable correlations in *Schizosaccharomyces pombe* (*Appendix 1—figure 6*). In addition, we found that fractions of coding genes almost monotonically decreased with ranks of centrality score in the *S. pombe* data (*Appendix 1—figure 6*).

We further analyzed two kinds of *Homo sapiens* transcriptome data. One is a human cell atlas, in which expression of both coding and non-coding genes in 15 fetal organs was quantified (*Cao et al., 2020*), and the other is genome-wide Perturb-seq data (*Replogle et al., 2022*), in which genetically perturbed transcriptomes were measured mainly for coding genes. Our analysis of the human cell atlas data revealed that, while the overall distribution of stoichiometry conservation centrality was broad (*Figure 5D*, top), the centrality distribution of coding genes was skewed to higher values (*Figure 5D*, bottom) as observed for the *E. coli* proteome. Fractions of coding genes almost monotonically decreased with ranks of centrality (*Figure 5E*) as seen in the *S. pombe* data (*Appendix 1—figure 6*). Essentiality of each gene in human cells was quantified by an index called CRISPR score, which measures the fitness cost imposed by CRISPR-based inactivation of the gene (*Wang et al., 2015*). Genes with lower CRISPR scores are considered more essential. Our analysis revealed that genes with higher stoichiometry conservation centrality scores tend to have lower CRISPR scores, thus more essential (*Figure 5F*). Similarly, genes with lower CRISPR scores tend to have higher stoichiometry conservation centrality scores. Furthermore, genes with higher centrality scores have more orthologs across the three domains of life and vice versa (*Figure 5G*). Comparable correlations of stoichiometry conservation with essentiality and evolutionary conservation were also found in the genome-wide Perturb-seq data (*Figure 5H and I*). Together, these results suggest that correlations of stoichiometry conservation centrality with gene essentiality and evolutionary conservation are general and preserved from *E. coli* to human cells regardless of the type of perturbation (see 'Relevance of centrality of csLE structure to biological functions' in Materials and methods for details).

## Revealing global stoichiometry conservation architecture of the proteomes with csLE

Although the previous analysis revealed the biological relevance of stoichiometry conservation centrality, it is a one-dimensional quantity and cannot capture the global architecture of omics profiles. To gain further insights into genome-wide stoichiometry-conserving relationships among genes, we next analyzed the proteomes using a method similar to Laplacian eigenmaps (LE) (*Appendix 1—figure 1*; *Belkin and Niyogi, 2003*). We consider a symmetric $n \times n$ matrix $A$ whose $(i, j)$ entry is $\cos \theta_{p_i p_j}$ (*Figure 5J, Table 1*). The entire proteome structure can be represented using the eigenvectors

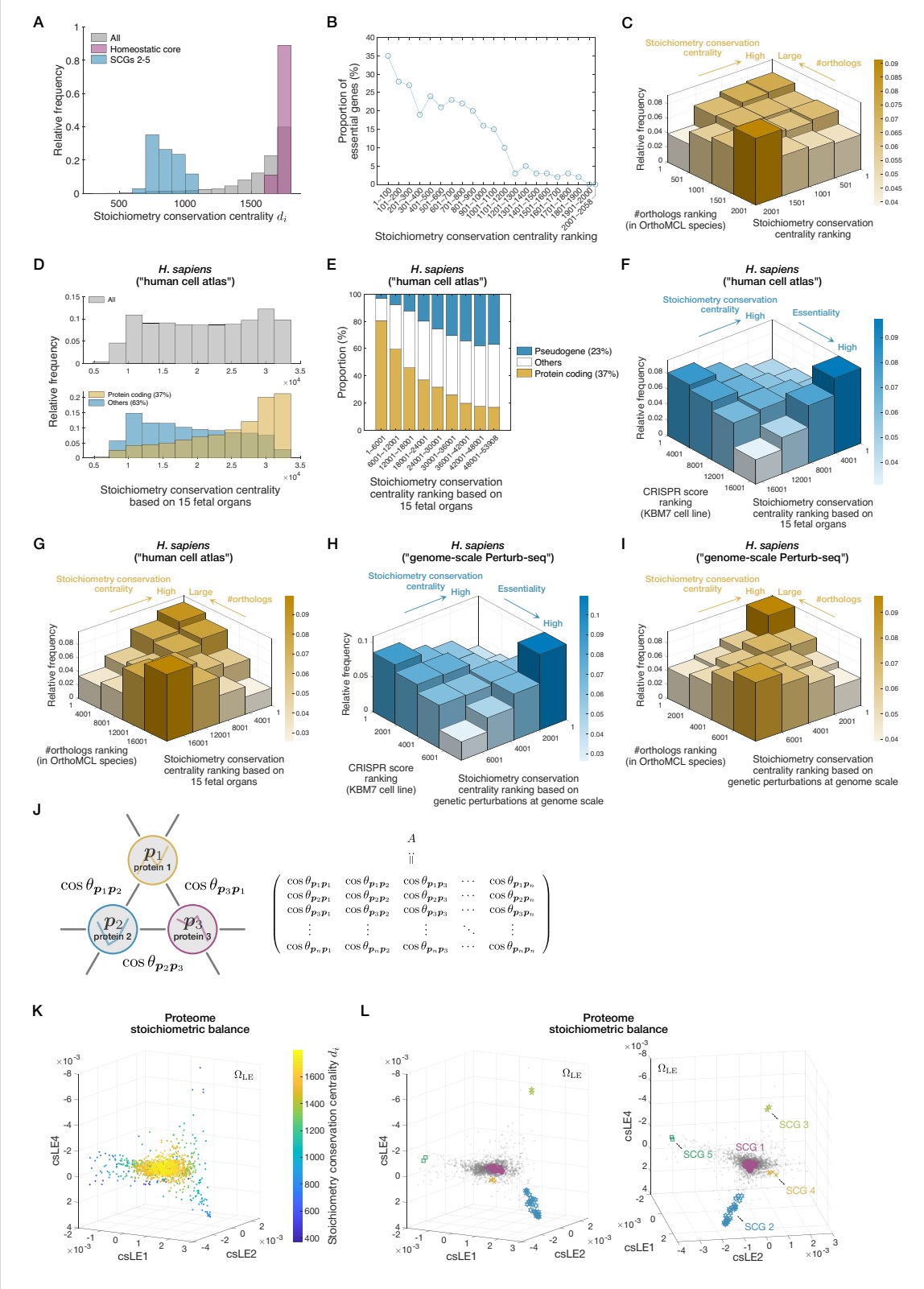

**Figure 5.** A proteome structure characterized by global stoichiometry conservation relationships. (**A**) Distributions of stoichiometry conservation centrality values for all the proteins (gray), the homeostatic core (SCG 1) proteins (magenta), and the proteins belonging to the other stoichiometrically conserved groups (SCGs) (cyan). (**B**) Correlation between stoichiometry conservation centrality and gene essentiality. The proportion of essential genes within each class of stoichiometry conservation ranking is shown. The list of essential genes was downloaded from EcoCyc (*Keseler et al., 2017*).

*Figure 5 continued on next page*

*Figure 5 continued*

(**C**) Correlation between stoichiometry conservation and evolutionary conservation. The strength of evolutionary conservation of each protein species was estimated by the number of orthologs found in the OrthoMCL species (**Chen et al., 2006**). The genes with more orthologs tend to have higher stoichiometry conservation centrality ($p = 3.42 \times 10^{-14}$ by one-sided Brunner-Munzel test between the top 25% and the bottom 25% fractions of ortholog number ranking). Likewise, the genes with higher stoichiometry conservation centrality scores tend to have more orthologs ($p = 8.44 \times 10^{-12}$ by one-sided Brunner-Munzel test, top 25%–bottom 25% comparison; $p$-values in the captions for (**F–I**) were evaluated with the same statistical test scheme). (**D–G**) Stoichiometry conservation analyses of human cell atlas transcriptome data of fetal 15 organs (**Cao et al., 2020**). The top gray histogram in (**D**) shows the distribution of stoichiometry conservation centrality values for all genes. The bottom histograms in (**D**) show the distribution for coding genes (yellow) and that for the other genes (cyan). (**E**) shows a correlation between the ratio of coding genes and stoichiometry conservation centrality calculated from the human cell atlas data. (**F**) shows a correlation between gene essentiality and stoichiometry conservation centrality calculated from the human cell atlas data. The essentiality of each human gene was quantified by CRISPR score, which is the fitness cost imposed by CRISPR-based inactivation of the gene in KBM7 chronic myelogenous leukemia cells (**Wang et al., 2015**). Genes with lower CRISPR score are regarded as more essential. The fraction with low CRISPR scores (i.e. high essentiality fraction) tends to have higher stoichiometry conservation centrality ($p < 10^{-15}$). The fraction with high centrality scores tends to be more essential ($p < 10^{-15}$). (**G**) shows a correlation between evolutionary conservation and stoichiometry conservation centrality based on the human cell atlas data. The gene fraction with many orthologs tends to have higher stoichiometry conservation centrality ($p < 10^{-15}$). The gene fraction with high centrality scores tends to have more orthologs ($p < 10^{-15}$). (**H**) and (**I**) Stoichiometry conservation analyses of genome-wide Perturb-seq data (**Replogle et al., 2022**). (**H**) shows a correlation between stoichiometry conservation centrality calculated from the Perturb-seq data and gene essentiality. The essentiality of each gene was quantified by the CRISPR score as in (**F**). The gene fraction with low CRISPR scores (i.e. high essentiality fraction) tends to have higher stoichiometry conservation centrality ($p < 10^{-15}$). The gene fraction with high centrality scores tends to be more essential ($p < 10^{-15}$). (**I**) shows a correlation between stoichiometry conservation based on the Perturb-seq data and evolutionary conservation of genes. The gene fraction with many orthologs tends to have higher stoichiometry conservation centrality ($p < 10^{-15}$). The gene fraction with high centrality scores tends to have more orthologs ($p < 10^{-15}$). (**J**) Representation of the proteomes as a graph. A node corresponds to a protein species, and the weight of an edge is taken as the cosine similarity between the $m$-dimensional expression vectors of the two connected protein species. The $n \times n$ matrix $A$ can specify the whole graph. Note that the diagonal elements of $A$ are ones, which were introduced just for simplicity. (**K**) Cosine similarity LE (csLE) structure in a three-dimensional space. Each dot represents a different protein species and is color-coded on the basis of its stoichiometry conservation centrality value. We selected the axes considering the structural similarity to the Raman-based proteome structure in $\Omega_B$ (see **Figure 6**). (**L**) The csLE structure in a three-dimensional space. The views from two different angles are shown. Each gray dot represents a different protein species. The proteins belonging to each SCG are indicated with distinct markers. Colors of the two-dimensional histograms in (**C**), (**F**), (**G**), (**H**), and (**I**) represent the height of each bar.

of normalized $A$. Major differences of this method from the ordinary LE are that we consider an edge for all node pairs and that we adopt cosine similarity for weighting edges. This method places the proteins with higher cosine similarity closer in the resulting $(m-1)$-dimensional space (see 'Global proteome structures based on stoichiometric balance' in Materials and methods and Section 2.1 in Appendix); we call this linear method cosine similarity LE (csLE).

In this $(m-1)$-dimensional csLE space $\Omega_{LE}$, the stoichiometry conservation centrality of the proteins decreased from center to periphery (**Figure 5K**), which confirms that it indeed measures the extent to which each protein is close to the center in the entire stoichiometry conservation architecture. Furthermore, the proteins formed polyhedral distributions with the cluster of the proteins in the homeostatic core at the center and the clusters of the proteins in the other condition-specific SCGs at distinct vertices (**Figure 5L**). This distribution is consistent with the fact that the condition-specific SCGs are the components whose expression patterns are distant from the homeostatic core and also between each other.

## Representing the proteomes using the Raman LDA axes

Given that the analysis of the LDA Raman-proteome regression coefficients $B$ (**Figure 3A**) eventually led us to identify the stoichiometry conservation architecture in the proteome data (**Figure 5**), the low-dimensional proteome structure in $\Omega_{LE}$ might be related to major changes in cellular Raman spectra in the LDA space and provide insight into the Raman-proteome correspondence. To investigate this, we considered representing the proteomes on the basis of the Raman LDA axes (**Appendix 1—figure 1**).

The coefficients in the $n \times m$ regression matrix $B$ must satisfy the proportionality $b_{ik}/b_{i0} = b_{jk}/b_{j0}$ for all $k$-th LDA axes ($1 \le k \le m-1$) for the pair of protein $i$ and protein $j$ that perfectly conserve their stoichiometry, as previously mentioned in the analysis of the ISP COG class (**Figure 3A**; see 'Characterizing an SCG by analyzing the Raman-proteome correspondence matrix' in Materials and methods and Section 2.1 in Appendix). Noting this property, we constructed another $(m-1)$-dimensional proteome space $\Omega_B$, assigning each protein species $i$ a coordinate $\left( \beta_i^{LDA1} \; \beta_i^{LDA2} \; \cdots \; \beta_i^{LDA(m-1)} \right)$, where $\beta_i^{LDAk} := b_{ik}/b_{i0}$ is the normalized coefficient of gene $i$ corresponding to the $k$-th LDA axis. As

in $(m-1)$-dimensional $\Omega_{\mathrm{LE}}$, a pair of proteins with strong stoichiometry conservation is expected to position closely in this $(m-1)$-dimensional proteome space $\Omega_{\mathrm{B}}$. Note that the proximity of the coordinates $\beta_i^{\mathrm{LDA}k}$ of different proteins $i$ in $\Omega_{\mathrm{B}}$ is equivalent to the approximate proportionality of different proteins $i$ in *Figure 3A*, demonstrated for the ISP COG class using the proportionality constants (normalized coefficients) $c_k$ common to different proteins.

We then found that the distribution of the proteins in $\Omega_{\mathrm{B}}$ closely resembled the one in $\Omega_{\mathrm{LE}}$ when visualized using the first few major axes (*Figure 5L* and *Figure 6A*). This similarity is nontrivial because $\Omega_{\mathrm{LE}}$ is constructed only from the proteome data, whereas $\Omega_{\mathrm{B}}$ depends on the $(m-1)$-dimensional Raman LDA space (*Figure 2C–E*).

We remark that each axis of $\Omega_{\mathrm{B}}$ is directly linked to the corresponding Raman LDA axis. Consequently, the orthants in $\Omega_{\mathrm{B}}$ where the condition-specific protein species reside agree with those in the Raman LDA space where the cellular Raman spectra under corresponding conditions reside (*Appendix 1—figure 10*) (see 'Global omics structures characterized by Raman-omics correspondences' in Materials and methods and Section 2.1 in Appendix). Indeed, we find such orthant agreement between the proteins in the condition-specific SCGs (SCG 2–SCG 5) and the cellular Raman spectra under the corresponding conditions (*Figure 6B and C*). This straightforward correspondence between $\Omega_{\mathrm{B}}$ and the Raman LDA space allows us to examine the relationship between changes in cellular Raman spectra and omics components' stoichiometry conservation architecture by comparing the two proteome structures in $\Omega_{\mathrm{B}}$ and $\Omega_{\mathrm{LE}}$.

## Omics-level interpretation of cellular Raman spectra and a quantitative constraint between expression generality and stoichiometry conservation centrality

To understand rigorously what the similarity of the proteome structures in $\Omega_{\mathrm{B}}$ and $\Omega_{\mathrm{LE}}$ signifies (*Figure 6C and D*), we clarified the mathematical relation between the coordinates of the proteins in these two spaces (*Figure 6E* and *Appendix 1—figure 1*; see Sections 2.1 and 2.2 in Appendix for details). We then characterized the two mathematical conditions that must be satisfied simultaneously (*Figure 6E*).

The first condition is that major axes of the Raman LDA space and those of the proteome csLE space correspond (*Figure 6E*). Consequently, cellular Raman spectra under a condition accompanying the expression of a condition-specific SCG must be significantly different from those under conditions with the expression of other condition-specific SCGs in a manner distinguishable by LDA. Mathematically, this condition is related to the $m \times m$ orthogonal matrix $\Theta$ that appears in the equation in *Figure 6E*. For the distributions of the proteome components to be similar in the low-dimensional subspaces of $\Omega_{\mathrm{LE}}$ and $\Omega_{\mathrm{B}}$, $\Theta$ must be close to the identity matrix with small off-diagonal elements (*Figure 6E*). We verified this first condition with the data (*Appendix 1—figure 9*; see 'Evaluating similarity between orthogonal matrix $\Theta$ and identity matrix' in Materials and methods for details).

The second condition relates to the proportionality of the $n$-dimensional vectors $\boldsymbol{b}_0$ and $\boldsymbol{b}_0^{\mathrm{est}}$ in *Figure 6E*. This proportionality relation can be transformed into another relation that $d_i$ is proportional to $g_i := \|\boldsymbol{p}_i\|_1 / \|\boldsymbol{p}_i\|_2$, where $\|\boldsymbol{p}_i\|_1$ and $\|\boldsymbol{p}_i\|_2$ are the $L^1$ and $L^2$ norms of the expression-level $m$-dimensional vector of protein $i$ across conditions (*Figures 4A and 6E*, *Table 1*).

$g_i$ can be interpreted as *the expression generality score*. When $g_i$ is large, the protein $i$ is expressed generally across conditions; when $g_i$ is small, this is expressed only under specific conditions (*Appendix 1—figure 8*) (see 'Interpretation of $L^1$ norm/$L^2$ norm ratio of an expression vector as a quantitative measure of expression generality' in Materials and methods). Therefore, the proportionality between $d_i$ and $g_i$ indicates that the proteins with high stoichiometry conservation centrality must be expressed nonspecifically to conditions. We also verified this condition with the data, confirming that it is indeed satisfied (*Figure 7A* and *Appendix 1—figure 9*).

The spread of the points from the proportionality diagonal line of the *E. coli* proteome data in *Figure 7A* was found related to the growth rate under the condition where each protein is expressed the most (see Section 2.2 in Appendix for a detailed analysis on the origin of the deviation). Consequently, one can envisage a growth-rate-dependent expression pattern of each protein on the basis of its relative position in this $g_i$-$d_i$ plot (*Figure 7B and C*). For example, both BamB and YqjD are expressed nonspecifically to the conditions with nearly identical expression generality scores. However, BamB is expressed at higher levels under fast growth conditions, whereas YqjP is expressed

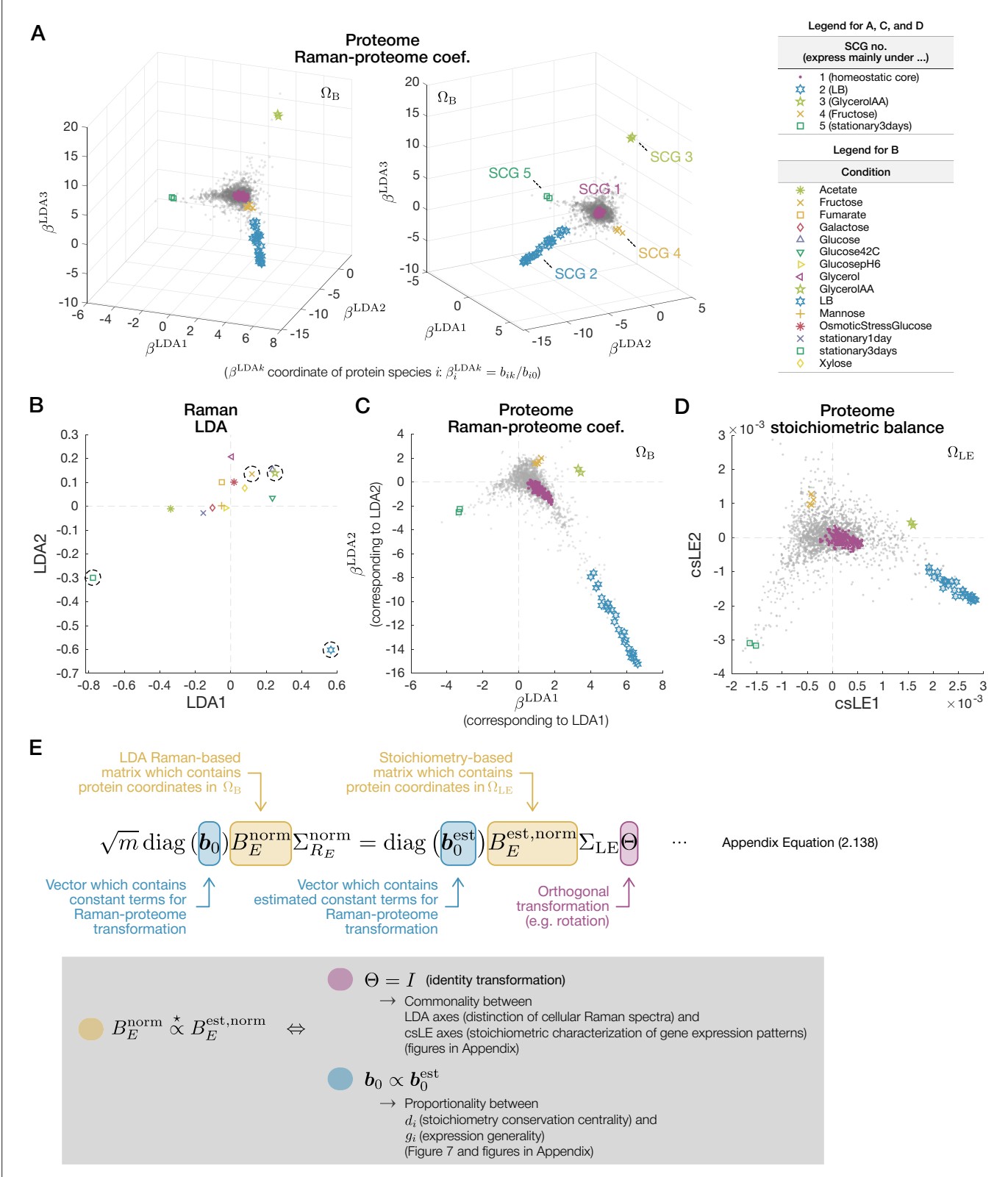

**Figure 6.** Raman-based proteome structure and its similarity to stoichiometry-based proteome structure. (**A**) Proteome structure determined by Raman-proteome coefficients visualized in a three-dimensional space. The views from two different angles are shown. Each gray dot represents a protein species. The proteins belonging to each stoichiometrically conserved group (SCG) are indicated with distinct markers. We note that SCGs are defined without referring to Raman data (*Figure 4*). (**B–D**) Similarity among the distribution of linear discriminant analysis (LDA) Raman spectra (**B**), the proteome

*Figure 6 continued on next page*

*Figure 6 continued*

structure determined by Raman-proteome coefficients (**C**), and the proteome structure determined by stoichiometry conservation (**D**). (**E**) Mathematical relation between the coordinates of the proteins in $\Omega_B$ (**C**) and $\Omega_{LE}$ (**D**). The two conditions, one with $\Theta$ (magenta) and the other between $\boldsymbol{b}_0$ and $\boldsymbol{b}_0^{est}$ (cyan), must hold for the similarity between the two proteome structures (yellow), as described in the gray box. $\overset{*}{\propto}$ denotes column-wise proportionality.

at higher levels under slow growth conditions due to their relative positions to the proportionality line. A similar growth rate dependence is observed for PaaE and DgoA, but with more prominent condition specificity because these proteins are characterized by their low expression generality scores. These growth-rate-dependent deviation patterns might hint at a new growth law that governs *the total relative expression changes* of the proteome components (see Section 2.2 in Appendix for detailed discussion).

## Generality

We also examined the generality of the aforementioned two conditions using the Raman and proteome data of *E. coli* strains with different genotypes (BW25113, MG1655, and NCM3722) under two culture conditions (*Schmidt et al., 2016*) and the Raman and transcriptome data of *S. pombe* under 10 culture conditions (*Kobayashi-Kirschvink et al., 2018*). Applying csLE to the omics data, we again found similar omics structures between $\Omega_{LE}$ and $\Omega_B$ when visualized using the first few major axes, with homeostatic cores at the centers and condition-specific SCGs at the vertices (*Appendix 1— figures 11 and 12*).

Proportionality between stoichiometry conservation centrality and expression generality score was also confirmed in both additional datasets (*Appendix 1—figure 7*). We further used publicly available quantitative proteome data of *M. tuberculosis*, *M. bovis*, and *S. cerevisiae* (*Schubert et al., 2015*; *Lahtvee et al., 2017*) to examine this relation and confirmed that the proportionality universally holds (*Appendix 1—figures 7 and 13*). Almost no deviation from the proportionality line existed in the *S. cerevisiae* proteome data measured for the cells in different media but cultured in chemostats with an identical dilution rate (thus, identical growth rate), which is consistent with the result of *E. coli* in which the deviations were related to the growth rate differences.

## Discussion

A Raman spectrum obtained from a single cell is a superposition of the spectra of all of its constituent biomolecules. Therefore, cellular Raman spectra potentially contain rich information on essential state differences in targeted cells. The fact that both transcriptomes and proteomes are inferable from cellular Raman spectra, as demonstrated in this and previous (*Kobayashi-Kirschvink et al., 2018*) studies, endorses this speculation. The detailed analyses of the relationship between Raman and omics data have identified functionally relevant constraints on omics changes and provided an interpretation of cellular Raman spectra (*Appendix 1—figure 1*). Specifically, it has been revealed that major changes in cellular Raman spectra distinguishable by LDA reflect the changes in omics profiles under the constraints of stoichiometry conservation. This correspondence would help us interpret global changes in cellular Raman spectra by translating them into the differences in omics profiles.

We remark that linearity in our formulation enabled us to find the rigorous connection between the two omics spaces $\Omega_B$ and $\Omega_{LE}$ (*Figure 6E*). Unlike the original LE, we adopted cosine similarity as weights of edges between all node pairs to measure expression stoichiometry conservation of proteins. This modification was indispensable in terms of interpretation; relative proximity of positions in $\Omega_{LE}$ reflects the strength of stoichiometry conservation. We also remark that simple principal component analysis (PCA) applied to the normalized *E. coli* proteome data also finds a similar low-dimensional proteome structure (*Appendix 1—figure 6*) (see 'Proteome structure obtained with PCA' in Materials and methods). Therefore, besides interpretability, omics structures in $\Omega_{LE}$ might reflect dominant relationships among omics components commonly characterized by several methods of omics representation.

It should be noted that the quantitative analysis of Raman-omics correspondence resulted in the characterization of stoichiometry-conserving architecture in cells (*Appendix 1—figure 1*). This shows that besides distinguishing different cellular states or quantifying specific biomolecular species by focusing on spectral peaks, Raman spectra can also characterize the system-level constraints behind

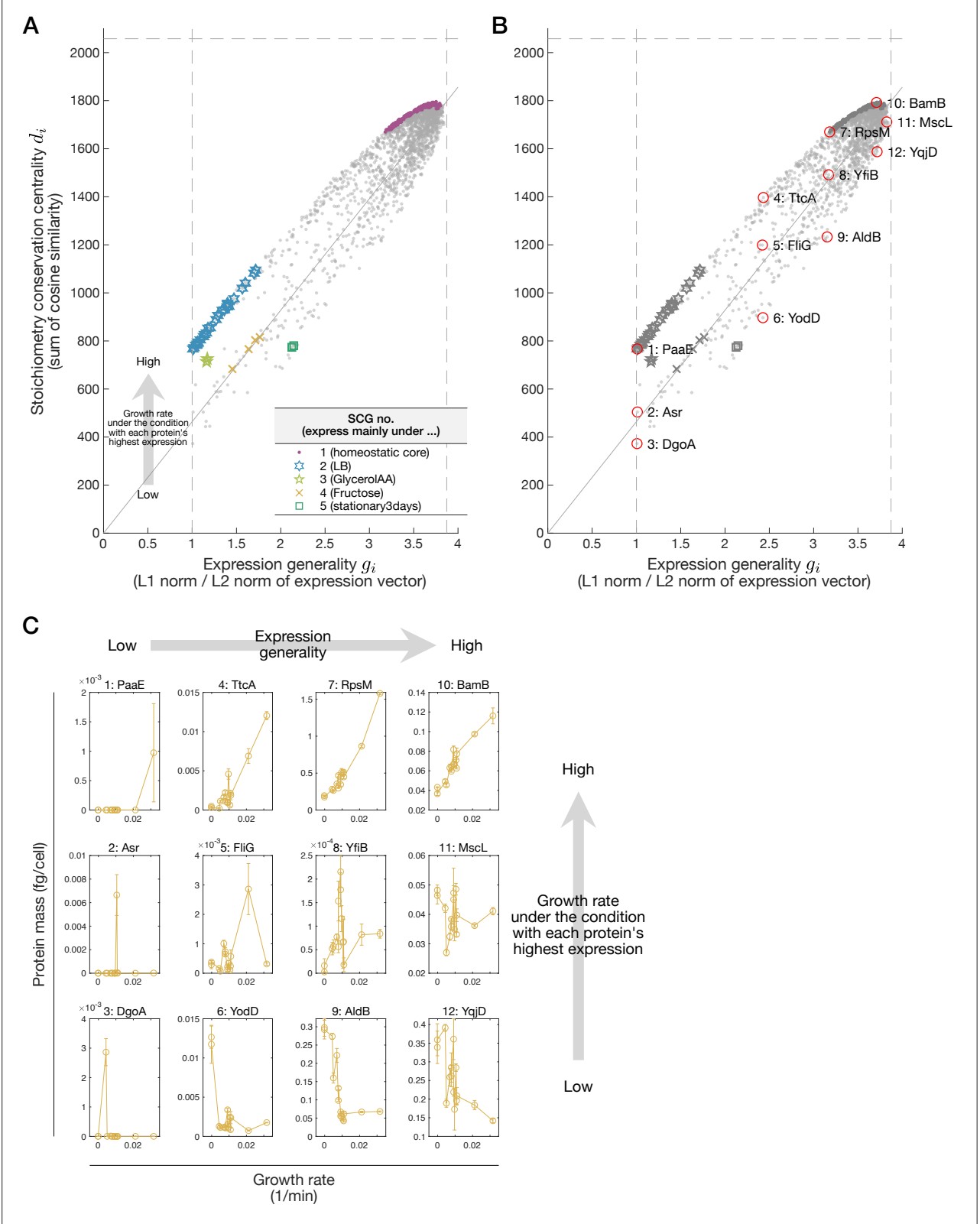

**Figure 7.** Proportionality between stoichiometry conservation centrality and expression generality. (**A**) Relationships between stoichiometry conservation centrality ($d_i$) and expression generality ($g_i$). Each gray dot represents a protein species. The proteins belonging to each stoichiometrically conserved group (SCG) are indicated with distinct markers. The dashed lines are $y = n$, $x = 1$, $\sqrt{m}$ ($n = 2058$, $m = 15$). The solid lines represent

*Figure 7 continued on next page*

*Figure 7 continued*

$y = \left\{ \left( \sum_{j=1}^{n} d_j \right) / m \right\}^{1/2} x$ (see Section 2.2 in Appendix). The deviation of a point from the solid line is related to the growth rate under the condition where each protein is expressed the most. (**B**) The same plot as (**A**) in black and white. Overlaid red circles indicate proteins featured in (**C**). (**C**) Expression patterns of the proteins indicated by red circles in (**B**) across conditions. The condition differences are shown by the growth rate differences on the horizontal axes. The arrangement of the plots for the proteins corresponds to their relative positions in (**B**).

changes in global gene expression profiles. While the identified features, such as stoichiometry conservation centrality, expression generality score, and csLE space, can be calculated without Raman data, it is difficult to reach them directly without scrutinizing the Raman-omics correspondence. Furthermore, the definition of expression generality and its relation to stoichiometry conservation centrality were directly derived from the Raman-omics correspondence analysis (*Figures 6E and 7*). Therefore, as a signal reflecting comprehensive molecular profiles in cells, Raman spectra are an important modality for dissecting system-level properties and constraints in cells.

In this study, we mainly analyzed the Raman and proteome data of *E. coli* under 15 different environmental conditions. However, the resulting low-dimensional structures and correspondence of $\Omega_{\text{LE}}$ and $\Omega_{\text{B}}$ can change depending on what and how many conditions are included in the analysis. Thus, an intriguing question is how the Raman-proteome correspondence is affected by the conditions used in the analysis. A subsampling analysis focusing on the orthogonal matrix $\Theta$, which represents low-dimensional correspondence precision of $\Omega_{\text{LE}}$ and $\Omega_{\text{B}}$ (*Figure 6E*), reveals that correspondence precision tends to increase with an increasing number of conditions (*Appendix 1—figure 14*). This result suggests that increasing the number of conditions generally improves the low-dimensional correspondence rather than disrupting it.

Since the proteome data that we referenced (*Schmidt et al., 2016*) represent the averaged expression profile of the cells in each condition, we likewise averaged the single-cell Raman data in each condition in the LDA space to determine their correspondence. Once this correspondence is established, it becomes technically feasible to infer the proteomes of individual cells from their Raman spectra. However, verifying the accuracy of the inferred proteome profiles requires quantitative ground truth of single-cell proteomes, which are not yet readily obtainable, especially for bacterial cells. Despite this limitation, future studies may clarify the correspondence at the single-cell level as omics technology advances.

Stoichiometry conservation is plausibly crucial for cellular functions and physiology. For example, the enzymes involved in evolutionarily conserved metabolic pathways conserve their stoichiometry across microorganism species despite their diverse transcriptional and translational rates (*Lalanne et al., 2018*). It is suggested that stoichiometry conservation is achieved by optimizing the metabolic flux for fast growth (*Lalanne and Li, 2021*). Furthermore, a ribosome-targeting antibiotic causes an imbalance of ribosomal proteins and growth arrest in *E. coli*, but the balance is restored alongside growth recovery through physiological adaptation (*Koganezawa et al., 2022*). These results suggest that disruption of stoichiometric balance among core components could impose significant fitness cost.

It is known that functions, essentiality, and evolutionary conservation of genes can be linked to the topologies of gene networks (*Jeong et al., 2001*; *He and Zhang, 2006*; *Yu et al., 2007*; *Fraser et al., 2002*; *Wuchty et al., 2003*; *Li et al., 2020*). However, networks that have been previously analyzed, such as protein-protein interaction networks, depend on known interactions. Therefore, as our understanding of the molecular interactions evolves with new findings, the conclusions may change. Furthermore, analysis of a particular interaction network cannot account for effects of different types of interactions or multilayered regulations affecting each protein species, thus highlighting only one aspect of the inherently global coordination of molecular compositions in cells. In contrast, the stoichiometry conservation network in this study focuses solely on expression patterns as the net result of interactions and regulations among all types of molecules in cells. Consequently, the stoichiometry conservation networks are not affected by the detailed knowledge of molecular interactions and naturally reflect the global effects of multilayered interactions behind cellular physiological state changes. Additionally, stoichiometry conservation networks can easily be obtained for non-model organisms, for which detailed molecular interaction information is usually unavailable. Therefore, analysis with the

stoichiometry conservation network has several advantages over existing methods from both biological and technical perspectives.

It is intriguing to ask how cells conserve stoichiometry among the components in each SCG. In particular, the homeostatic core (SCG 1) contains many components whose gene loci are scattered throughout the genome. It is known that both transcriptional and translational negative autoregulation contributes to controlling the stoichiometry of many ribosomal proteins (*Nomura et al., 1980*; *Dean et al., 1981*; *Kaczanowska and Rydén-Aulin, 2007*; *Portier and Grunberg-Manago, 1993*; *Aseev et al., 2008*; *Roy et al., 2020*). The genes for the ribosomal proteins are scattered in multiple operons and co-regulated with many other non-ribosomal proteins, such as RNA polymerase subunits, translation initiation/elongation factors, and transmembrane transporters (*Keseler et al., 2017*). Therefore, the stoichiometry-conserving mechanisms established for ribosomes might be partially exploited for the stoichiometry conservation within the homeostatic core.

The existence of condition-specific SCGs and genes with similar expression patterns confirms that adaptation to specific conditions is not necessarily achieved by a small number of functionally relevant genes, but is often accompanied by changes in the expression of many seemingly unrelated genes. Indeed, condition-specific SCGs contain genes with unclear roles in adaptation, including some that are functionally uncharacterized (*Appendix 1—table 5–8*). Therefore, it would be important to investigate whether the coexpression of multiple genes is crucial for cellular adaptation to a wide range of perturbations while maintaining homeostasis.

The proportionality between stoichiometry conservation centrality and expression generality score suggests that proteins with high stoichiometry conservation centrality govern basal cellular functions required under any conditions. In fact, both essential genes and evolutionarily conserved genes are enriched in the omics fractions with high centrality scores. On the contrary, proteins of low centrality scores might have been acquired in later stages of the evolution and exploited to survive or increase fitness under specific conditions. Such hierarchy in the stoichiometry conservation centrality among core and peripheral processes might promote the adaptability of cells since cells can respond to diverse environments without restructuring a large body of the functional homeostatic core. This architectural principle in omics might underlie the robustness and adaptability of biological cells.

## Materials and methods

**Key resources table**

| Reagent type (species) or resource | Designation | Source or reference | Identifiers | Additional information |
|---|---|---|---|---|
| Chemical compound, drug | Difco LB Broth, Miller (Luria-Bertani) | Becton, Dickinson and Company | | |
| Chemical compound, drug | Bacto Yeast Extract | Becton, Dickinson and Company | | |
| Chemical compound, drug | Bacto Tryptone | Becton, Dickinson and Company | | |
| Chemical compound, drug | Sodium Chloride | Wako Pure Chemical Industries, Ltd. | | |
| Chemical compound, drug | Disodium Hydrogenphosphate | Wako Pure Chemical Industries, Ltd. | | |
| Chemical compound, drug | Potassium Dihydrogenphosphate | Wako Pure Chemical Industries, Ltd. | | |
| Chemical compound, drug | Ammonium Sulfate | Wako Pure Chemical Industries, Ltd. | | |
| Chemical compound, drug | Zinc Sulfate Heptahydrate | Wako Pure Chemical Industries, Ltd. | | |
| Chemical compound, drug | Cooper(II) Chloride Dihydrate | Wako Pure Chemical Industries, Ltd. | | |
| Chemical compound, drug | Manganese(II) Sulfate Pentahydrate | Wako Pure Chemical Industries, Ltd. | | |
| Chemical compound, drug | Cobalt(II) Chloride Hexahydrate | Wako Pure Chemical Industries, Ltd. | | |
| Chemical compound, drug | Calcium Chloride Dihydrate | Wako Pure Chemical Industries, Ltd. | | |
| Chemical compound, drug | Magnesium Sulfate Heptahydrate | Wako Pure Chemical Industries, Ltd. | | |
| Chemical compound, drug | Thiamin Hydrochloride | Wako Pure Chemical Industries, Ltd. | | |
| Chemical compound, drug | Iron(III) Chloride Hexahydrate | Wako Pure Chemical Industries, Ltd. | | |

*Continued on next page*

*Continued*

| Reagent type (species) or resource | Designation | Source or reference | Identifiers | Additional information |
|---|---|---|---|---|
| Chemical compound, drug | Sodium Acetate | Wako Pure Chemical Industries, Ltd. | | |
| Chemical compound, drug | Disodium Fumarate | FUJIFILM Wako Pure Chemical Corporation | | |
| Chemical compound, drug | D-Galactose | Wako Pure Chemical Industries, Ltd. | | |
| Chemical compound, drug | D-Glucose | Wako Pure Chemical Industries, Ltd. | | |
| Chemical compound, drug | Glycerol | Wako Pure Chemical Industries, Ltd. | | |
| Chemical compound, drug | D-Fructose | FUJIFILM Wako Pure Chemical Corporation | | |
| Chemical compound, drug | D-Mannose | FUJIFILM Wako Pure Chemical Corporation | | |
| Chemical compound, drug | D-Xylose | Wako Pure Chemical Industries, Ltd. | | |
| Chemical compound, drug | L-Alanine | Wako Pure Chemical Industries, Ltd. | | |
| Chemical compound, drug | L-Asparagine Monohydrate | Wako Pure Chemical Industries, Ltd. | | |
| Chemical compound, drug | L-Cysteine | FUJIFILM Wako Pure Chemical Corporation | | |
| Chemical compound, drug | L-Glutamic acid | Wako Pure Chemical Industries, Ltd. | | |
| Chemical compound, drug | L-Glutamine | Wako Pure Chemical Industries, Ltd. | | |
| Chemical compound, drug | Glycine | Wako Pure Chemical Industries, Ltd. | | |
| Chemical compound, drug | L-Histidine | FUJIFILM Wako Pure Chemical Corporation | | |
| Chemical compound, drug | L-Isoleucine | Wako Pure Chemical Industries, Ltd. | | |
| Chemical compound, drug | L-Phenylalanine | Wako Pure Chemical Industries, Ltd. | | |
| Chemical compound, drug | L-Proline | Wako Pure Chemical Industries, Ltd. | | |
| Chemical compound, drug | L-Serine | Wako Pure Chemical Industries, Ltd. | | |
| Chemical compound, drug | Adenine | FUJIFILM Wako Pure Chemical Corporation | | |
| Chemical compound, drug | L-Arginine | FUJIFILM Wako Pure Chemical Corporation | | |
| Chemical compound, drug | L-Aspartic acid | Wako Pure Chemical Industries, Ltd. | | |
| Chemical compound, drug | L-Leucine | FUJIFILM Wako Pure Chemical Corporation | | |
| Chemical compound, drug | L-Lysine | Wako Pure Chemical Industries, Ltd. | | |
| Chemical compound, drug | L-Methionine | Wako Pure Chemical Industries, Ltd. | | |
| Chemical compound, drug | L-Threonine | Wako Pure Chemical Industries, Ltd. | | |
| Chemical compound, drug | L-Tryptophan | Wako Pure Chemical Industries, Ltd. | | |
| Chemical compound, drug | L-Tyrosine | Wako Pure Chemical Industries, Ltd. | | |
| Chemical compound, drug | L-Valine | Wako Pure Chemical Industries, Ltd. | | |
| Chemical compound, drug | Uracil | Wako Pure Chemical Industries, Ltd. | | |
| Chemical compound, drug | 8 mol/L Sodium Hydroxide Solution | Wako Pure Chemical Industries, Ltd., FUJIFILM Wako Pure Chemical Corporation | | |
| Chemical compound, drug | 35–37% (mass/mass) Hydrochloric Acid | Wako Pure Chemical Industries, Ltd. | | |
| Chemical compound, drug | 0.1 mol/L Hydrochloric Acid | Wako Pure Chemical Industries, Ltd. | | |
| Chemical compound, drug | Agar | Wako Pure Chemical Industries, Ltd., FUJIFILM Wako Pure Chemical Corporation | | |
| Strain, strain background (*Escherichia coli*) | BW25113 | Wakamoto Laboratory stock | | |
| Strain, strain background (*Escherichia coli*) | MG1655 | Wakamoto Laboratory stock | | |

| Reagent type (species) or resource | Designation | Source or reference | Identifiers | Additional information |
|---|---|---|---|---|
| Strain, strain background (*Escherichia coli*) | NCM3722 | Coli Genetic Stock Center | | |

Note that mathematical notation in Materials and methods differs in some respects from that in the main text, *Table 1*, and main figures.

## Experimental methods, data acquisition, and data analyses

### Absolute quantitative proteome data

We utilized high-quality absolute quantitative proteome data reported by *Schmidt et al., 2016*. In these data, expression levels of more than 55% of genes of *E. coli* BW25113 strain (more than 95% of total proteome mass) were quantified under various environmental conditions.

We also used additional absolute quantitative proteome data (*Schmidt et al., 2016*; *Schubert et al., 2015*; *Lahtvee et al., 2017*) for checking the generality of our findings (see Appendix 3.2). In addition to the proteome data across environmental conditions, Schmidt et al. also reported proteomes of *E. coli* strains with different genotype backgrounds (BW25113, MG1655, and NCM3722) cultured in a rich medium or a minimal medium supplemented with glucose. *Schubert et al., 2015* quantified proteomes of *M. tuberculosis* H37Rv strain and *M. bovis* BCG strain under time-course environmental change conditions starting from exponential growth conditions, followed by dormant states induced by decreasing oxygen levels, and finally regrowth conditions with re-aeration. *Lahtvee et al., 2017* quantified proteomes of *S. cerevisiae* under a reference condition and three stressed conditions (ethanol, osmotic pressure, and high temperature, with three stress intensity steps for each type of stress) using chemostat.

For checking the generality of our findings across omics classes, we also used the transcriptome data reported by our previous paper (*Kobayashi-Kirschvink et al., 2018*). The data include the transcriptomes of *S. pombe* in rich and minimal media, in nutrient-depleted media, and under various stress conditions.

### *E. coli* strains and culture conditions

To quantitatively analyze a linkage between the absolute proteome data generated by *Schmidt et al., 2016* and Raman data, we reproduced the culture conditions used in *Schmidt et al., 2016* as closely as possible in our lab. We obtained three biological replicates.

### *E. coli* strains

We used BW25113, MG1655, and NCM3722 as in *Schmidt et al., 2016*. In particular, BW25113 (*Datsenko and Wanner, 2000*) was used for the main data in this study. The genotype of BW25113 is F⁻ Δ(*araD-araB*)567 Δ*lacZ4787* (::rrnB-3) $\lambda^-$ *rph-1* Δ(*rhaD-rhaB*)568 *hsdR514*, that of MG1655 is F⁻ $\lambda^-$ *rph-1* , and that of NCM3722 is F⁺, respectively (*Baba et al., 2006*; *Blattner et al., 1997*; *Soupene et al., 2003*).

### Culture conditions

We prepared 15 batch culture conditions listed in *Appendix 1—table 1*. We excluded three culture conditions among the 18 conditions reported in *Schmidt et al., 2016* because we could not obtain sufficiently strong cellular Raman signals under those excluded conditions. See *Schmidt et al., 2016* for the detail of medium compositions. For 'GlucosepH6' medium, 37% HCl was titrated to the 'Glucose' medium. Medium for 'stationary1day' and 'stationary3days' was the same as 'Glucose' medium. LB agar plates were prepared by adding 15 g/L agar to 'LB' medium.

### Cultivation

Culturing *E. coli* cells proceeded in four steps:

- **Step 1: Growth on LB agar plates.** Cells were taken from a −80°C glycerol stock and streaked on LB agar plates. The plates were incubated at 37°C overnight and stored at 4°C. All subsequent experiments were conducted using colonies on the LB agar plates. Picking colonies from the plates for cultivation was done within 4 days of storage at 4°C.

- **Step 2: Liquid culture under 'Glucose' condition.** Several colonies picked from LB agar plates were inoculated into 'Glucose' liquid culture medium and grown for about 16 hr. Cells for the 'Glucose42C' condition were cultured at 42°C, and those for the other conditions were grown at 37 °C.
- **Step 3: Liquid culture under each condition.** Cells from Step 2 were passaged into each type of medium and grown to exponential phase. Cells for the 'Glucose42C' condition were grown at 42°C, and those for the other conditions were cultured at 37°C.
- **Step 4: Liquid culture under each condition.** Cells from Step 3 were passaged into the respective fresh medium and grown to almost the same level of turbidity as that at the end of Step 3. Cells for the 'Glucose42C' condition were cultured at 42°C, and those for the other conditions were grown at 37°C.

For the exponential conditions, cell cultivation was conducted as described above. For the stationary conditions, cultivation of cells at Step 3 was continued instead of proceeding to Step 4 and ended 1 or 3 days after they reached the stationary phase.

The medium volume was $2\,\mathrm{mL}$ for all the liquid cultures in our experiments. Borosilicate glass test tubes with a diameter of 16.5 mm and a length of 165 mm were used. A fresh medium was pre-warmed before passage so that its temperature was the same as that of cultivation. All the liquid cultures were under reciprocal shaking at $200\,\mathrm{r/min}$ and at an inclination of 45°. Liquid cultures were diluted to an $OD_{600}$ of around 0.01 for passage.

Main differences between our cultivation conditions and those of *Schmidt et al., 2016* are the periods of storage at 4°C at Step 1 (a maximum of 3 weeks in *Schmidt et al., 2016*), the number of colonies inoculated from plates to liquid medium at the second step (one colony per inoculation in *Schmidt et al., 2016*), and medium volumes and shaking conditions of liquid cultures ($50\,\mathrm{mL}$ liquid culture in $500\,\mathrm{mL}$ unbaffled wide-neck Erlenmeyer flasks under orbital shaking at $300\,\mathrm{r/min}$ in *Schmidt et al., 2016*).

## Growth rate measurements

Growth curves were obtained by continuing the Step 3 in cultivation. Cultivation of cells for growth measurements was conducted with $5\,\mathrm{mL}$ culture media, not $2\,\mathrm{mL}$, due to a requirement of the device used for continuous turbidity recording (ODBox-C, TAITEC Corporation). In addition, cells were washed with each type of fresh medium before inoculation at the beginning of Step 3, and cultivation for growth recording started from an $OD_{600}$ of around 0.001. Growth rates were calculated from the growth curves using the fitting algorithm based on Gaussian processes (*Swain et al., 2016*).

## Raman measurements and preprocessing of spectra

Cells were washed three times with 0.9% aqueous solution of NaCl, and 5 µL of the suspension was placed on a synthetic quartz slide glass (Toshin Riko Co., Ltd.) and dried. Raman spectra of cells were measured with a Raman microscope (*Appendix 1—figure 2*), where a custom-built Raman system (STR-Raman, AIRIX) was integrated into a microscope (Ti-E, Nikon). Excitation light was generated by a $532\,\mathrm{nm}$ continuous-wave diode-pumped solid-state laser (Gem 532, Laser Quantum). We altered the first version of this Raman microscope (*Kobayashi-Kirschvink et al., 2018*), and light from the laser oscillator was transmitted by mirrors in this research. A $100\times$ and $\mathrm{NA} = 0.9$ air objective lens (MPLN100X, Olympus) was used. Raman scattered light was collected by an optical fiber and transmitted to a spectrometer (Acton SP2300i, Princeton Instruments). Dispersed light by a $300\,\mathrm{gr/mm}$ grating was projected onto an image sensor of an sCMOS camera (OrcaFlash 4.0 v2, Hamamatsu Photonics). The sCMOS camera was water-cooled at 15°C to reduce dark noise. The exposure time for each cell was $10\,\mathrm{s}$. Randomly selected 15 cells were measured per condition per replicate. Raman spectrum of background was measured for each cell with $10\,\mathrm{s}$ exposure in an area close to a targeted cell where neither cells nor NaCl crystals existed.

In our setup, the laser power at the sample stage was $21\,\mathrm{mW}$. The measurement system and processes were controlled using Micro-Manager 1.4 (*Edelstein et al., 2014*) and a plugin we made.

Readout noise of sCMOS image sensors is pixel-dependent. A noise reduction filter developed in *Kobayashi-Kirschvink et al., 2018*, on the basis of *Huang et al., 2013*, was applied to measured spectral images by using 10,000 blank images obtained with the same sCMOS sensor with exposure time of $10\,\mathrm{s}$. See *Kobayashi-Kirschvink et al., 2018* for details.

After noise reduction with the filter, pixel counts were summed up along the direction perpendicular to wavenumber. A background spectrum was subtracted from a cellular Raman spectrum. A pixel region corresponding to the range from $632\,\text{cm}^{-1}$ to $1862\,\text{cm}^{-1}$ was cropped. The cropped spectrum was smoothed with a Savitzky-Golay filter (*Savitzky and Golay, 1964*). To minimize the effect of laser excitation variations, each spectrum was normalized by subtracting the average and dividing it by the standard deviation.

## Data analysis

We wrote scripts and analyzed data using MATLAB (R2019a and R2023b), except for Brunner-Munzel test, for which we used R (version 4.0.3) (see 'Centrality-evolutionary conservation correlation' in Materials and methods).

Related to *Figure 2* in the main text, we first performed LDA against the Raman data. LDA is a linear classifier; it finds the most discriminatory bases by maximizing the ratio of the between-class variance to the within-class variance and reduces the dimensions of the data to $m - 1$, where $m$ is the number of classes (*Huang et al., 2004*; *De Bie et al., 2005*; *Goodacre et al., 1998*). In the case of our main data, classes are culture conditions. In the verification step of the correspondence between the LDA Raman and omics data, we conducted LOOCV. In LOOCV, one condition is used as test data and the remaining conditions are used as training data. This is repeated by changing the condition to exclude.

The details of the data analyses are provided in the sections below.

## Raman-proteome statistical correspondence

### Notation

We write the population-averaged 14-dimensional LDA Raman spectrum vector of each condition as a row vector $\hat{r}_i$ ($i = 1, ..., 15$) and the 2058-dimensional absolute proteome vector of each condition as a row vector $\hat{p}_i$ ($i = 1, ..., 15$). Note that we regarded $\hat{r}_i$ and $\hat{p}_i$ as column vectors in the main text for simple expression of equations.

Our hypothesis of Raman-proteome linear correspondence (*Equation 1* in the main text) is expressed as

$$\hat{p}_i^{\top} = B \cdot \begin{bmatrix} 1 \\ \hat{r}_j^{\top} \end{bmatrix}, \tag{2}$$

where $B$ is a 2058 × 15 matrix and $\top$ denotes transpose. In LOOCV, one condition is excluded (let $i$ be the excluded condition) and the remaining 14 conditions are used to estimate $B$. We write the estimated $B$ as $B_{-i}^{\text{est}}$, which is also a 2058 × 15 matrix. Let $\hat{p}_i^{\text{est}}$ be the estimated proteome of the excluded condition in LOOCV (*Figure 2G*).

### OLS in LOOCV scheme

In the case of LOOCV, 14 (= 15 − 1) conditions are included in a training data. Thus, if all the 14 LDA axes of the low-dimensional Raman data are considered, OLS becomes underdetermined. We excluded higher dimensions of the Raman space to conduct OLS in LOOCV unless otherwise noted. The results described in the main text were obtained using the first four axes (LDA1 to LDA4). In this case, $B_{-i}^{\text{est}}$ is a 2058 × 5 matrix.

### Permutation test

Let a permutation of all the 15 conditions be $\sigma$. In our permutation test, we calculated overall estimation errors as $\sum_i \text{dist}\left(\hat{p}_i, \hat{p}_{\sigma(i)}^{\text{est}}\right)$, where $\text{dist}\left(\hat{p}_i, \hat{p}_{\sigma(i)}^{\text{est}}\right)$ is one of the distance measures between $\hat{p}_i$ and $\hat{p}_{\sigma(i)}^{\text{est}}$ listed in *Appendix 1—table 2*. There exist 15! sets of $\sigma$, and calculating all of them is computationally intensive. Thus, we randomly generated $10^5$ permutation sets.

The result presented in the main text is the case where Euclidean metric (PRESS) was used as a distance measure. Likewise, we also obtained small $p$-values with the other metrics (*Appendix 1—table 2*).

We could also estimate the proteomes with high accuracy using all the 14 dimensions of the LDA space (*Appendix 1—table 3*). As noted in 'OLS in LOOCV scheme' in Materials and methods, the regression is underdetermined in this case. Thus, we simply adopted the minimum-norm solution from among all least-squares solutions.

## Characterizing an SCG by analyzing the Raman-proteome correspondence matrix

### Notation

The component representation of *Equation 2* is

$$
\underbrace{\begin{pmatrix} p_{i1} \\ p_{i2} \\ \vdots \\ p_{in} \end{pmatrix}}_{\hat{\boldsymbol{p}}_i^\top} = \underbrace{\begin{pmatrix} b_{10} & b_{11} & \cdots & b_{1(m-1)} \\ b_{20} & b_{21} & \cdots & b_{2(m-1)} \\ \vdots & \vdots & \ddots & \vdots \\ b_{n0} & b_{n1} & \cdots & b_{n(m-1)} \end{pmatrix}}_{B} \underbrace{\begin{pmatrix} 1 \\ r_{i1} \\ \vdots \\ r_{i(m-1)} \end{pmatrix}}_{\begin{bmatrix} 1 & \hat{r}_i \end{bmatrix}^\top},
$$

(3)

where $n$ is the number of proteins and $m$ is the number of culture conditions. $n = 2058$ and $m = 15$ in our case. Let $\boldsymbol{b}_{h-1}$ be the $h$-th row of th column of $B$. For example, $\boldsymbol{b}_0 = (b_{10} \cdots b_{n0})^\top$ denotes the constant term for each protein, and $\boldsymbol{b}_1 = (b_{11} \cdots b_{n1})^\top$ the coefficient of LDA1 for each protein. The expression level of protein $j$ in the condition $i$ is

$$
p_{ij} = b_{j0} + b_{j1}r_{i1} + \ldots + b_{j(m-1)}r_{i(m-1)}.
$$

(4)

### Stoichiometry conservation of ISP COG class

In the main text, we revealed that many proteins belonging to ISP COG class were aligned on a straight line passing through the origin when the relations between the columns of $B$ were shown in scatterplots (*Figure 3A*). Consider hypothetical proteins that align perfectly on a straight line through the origin. Let $e_1, \ldots, e_k$ be the indices of such perfectly aligning protein species. Extracting only these rows for the proteins from *Equation 3*, we obtain

$$
\begin{pmatrix} p_{ie_1} \\ p_{ie_2} \\ \vdots \\ p_{ie_k} \end{pmatrix} = \begin{bmatrix} \tilde{\boldsymbol{b}}_0 & \tilde{\boldsymbol{b}}_1 & \cdots & \tilde{\boldsymbol{b}}_{m-1} \end{bmatrix} \begin{pmatrix} 1 \\ r_{i1} \\ \vdots \\ r_{i(m-1)} \end{pmatrix}
$$

(5)

$$
= \begin{bmatrix} \tilde{\boldsymbol{b}}_0 & c_1\tilde{\boldsymbol{b}}_0 & \cdots & c_{m-1}\tilde{\boldsymbol{b}}_0 \end{bmatrix} \begin{pmatrix} 1 \\ r_{i1} \\ \vdots \\ r_{i(m-1)} \end{pmatrix}
$$

(6)

$$
= \left(1 + c_1 r_{i1} + \cdots + c_{m-1} r_{i(m-1)}\right) \tilde{\boldsymbol{b}}_0,
$$

(7)

where $c_i$ $(i = 1, 2, \ldots, m-1)$ are constants and $\tilde{\boldsymbol{b}}_h := \left(b_{e_1 h} \cdots b_{e_k h}\right)^\top$. For our data, *Appendix 1—figure 4A and B* correspond to *Equation 7*. In these plots, the $y$-axis represents $\hat{\boldsymbol{p}}_i^\top$, and the $x$-axis $\tilde{\boldsymbol{b}}_0$. Many ISP proteins indeed align on a straight line through the origin with different slopes for different conditions (*Appendix 1—figure 4A*). In contrast, many proteins in other COG classes do not align on a straight line (*Appendix 1—figure 4B*).

Importantly, for a pair of proteins $e_\alpha, e_\beta$ that align on the straight line,

$$
\frac{p_{ie_\alpha}}{p_{ie_\beta}} = \frac{b_{e_\alpha 0}}{b_{e_\beta 0}}
$$

(8)

holds from *Equation 7*. The right-hand side of *Equation 8* does not contain condition index $i$, which means that the abundance ratio of the proteins remains constant regardless of the conditions.

## On the evaluation of stoichiometry conservation by Pearson correlation coefficient

In the main text, we used Pearson correlation coefficients to confirm the stoichiometry conservation of many ISP COG class members (*Figure 3B*). However, strictly speaking, cosine similarity is a more appropriate measure to evaluate stoichiometry conservation. In this analysis, cosine similarity can be written as

$$\cos \theta_{\tilde{\hat{p}}_i \tilde{\hat{p}}_j} = \frac{\tilde{\hat{p}}_i \cdot \tilde{\hat{p}}_j}{\left\| \tilde{\hat{p}}_i \right\|_2 \left\| \tilde{\hat{p}}_j \right\|_2}, \tag{9}$$

where $\tilde{\hat{p}}_i$ and $\tilde{\hat{p}}_j$ are the vectors representing the protein abundance for the proteome subgroups ('ISP' COG class, 'Cellular processes and signaling' COG class, and 'Metabolism' COG class) for conditions $i$ and $j$, respectively ($1 \leq i, j \leq m$). Cosine similarity version of *Figure 3B* is *Appendix 1—figure 4F*. The cosine similarity takes the maximum value 1 only when abundance ratios between all considered proteins are perfectly the same between the two compared conditions.

In addition, we also examined differences between COG classes by calculating Pearson correlation coefficients of log abundances (*Appendix 1—figure 4E*).

## Direct characterization of SCGs in omics data

### Notation

Let $p_i$ be a column vector representing the abundances of protein $i$. Each component of this vector indicates the abundance of protein $i$ under each condition. Therefore,

$$p_i = \begin{pmatrix} p_{1i} \\ p_{2i} \\ \vdots \\ p_{mi} \end{pmatrix}, \tag{10}$$

where $m = 15$ in our case. Note that $p_i$ defined here is a 15-dimensional column vector and different from $\hat{p}_i$ introduced previously, which was a 2058-dimensional row vector.

### Identifying SCGs in omics data

As explained in the main text, we extracted SCGs directly from the omics data, without referring to Raman data or COG classification. We evaluated the similarity of expression patterns for all the combinations of proteins using cosine similarity. Specifically, cosine similarity between proteins $i$ and $j$ is calculated as

$$\cos \theta_{p_i p_j} := \frac{p_i \cdot p_j}{\|p_i\|_2 \|p_j\|_2}. \tag{11}$$

This is the inner product of normalized $p_i$ and $p_j$. Note that $0 \leq \cos \theta_{p_i p_j} \leq 1$ as protein abundances of any proteins are non-negative. $\cos \theta_{p_i p_j}$ takes the maximum value 1 if and only if $p_i$ and $p_j$ point in identical direction, i.e., the abundance ratios of proteins $i$ and $j$ are constant under all the conditions. Therefore, if we extract only the proteins connected with high cosine similarity from all $\binom{n}{2}$ protein pairs, they would constitute proteome fractions in each of which the abundance ratios of the proteins remain almost constant across all the $m$ conditions. We hence extracted only the protein pairs whose cosine similarity was above a high threshold of 0.995. As a result, we obtained several SCGs, in each of which the protein species are linked to each other with high cosine similarity (*Figure 4B and C*).

The genes in each SCG are listed in *Appendix 1—table 4–8*. Note that there are many other minor components (*Figure 4B*), some of which may have an expression pattern similar to another component but are separated due to the high threshold.

The positions of members of the SCGs on the chromosome are shown in **Figure 4D** (SCG 1 [homeostatic core]) and **Appendix 1—figure 5E** (SCGs 2–5).

## Global proteome structures based on stoichiometric balance

In the previous section, we identified SCGs by setting a threshold of cosine similarity for extracting protein pairs. We next removed the threshold and considered the 'distance' with respect to cosine similarity for all the protein pairs to capture the global proteome structure that includes SCGs.

The cosine similarity for all the $\binom{n}{2}$ pairs of proteins can be summarized in one matrix as

$$A := \left( \cos\theta_{\boldsymbol{p}_i\boldsymbol{p}_j} \right)_{1 \leq i,j \leq n} = \begin{pmatrix} \cos\theta_{\boldsymbol{p}_1\boldsymbol{p}_1} & \cdots & \cos\theta_{\boldsymbol{p}_1\boldsymbol{p}_n} \\ \vdots & \ddots & \vdots \\ \cos\theta_{\boldsymbol{p}_n\boldsymbol{p}_1} & \cdots & \cos\theta_{\boldsymbol{p}_n\boldsymbol{p}_n} \end{pmatrix}, \tag{12}$$

where $(i,j)$ component represents cosine similarity between proteins $i$ and $j$. Assuming that this matrix is an adjacency matrix in graph theory and network theory, the entire proteomes are considered as a weighted undirected complete graph (with loops), where nodes correspond to protein types and any protein pair is connected by an undirected edge. Each edge is weighted by the cosine similarity between the two protein species at both ends. Note that all the diagonal elements of $A$ are one, which represents that each node has a loop with weight of one. These were introduced just for simplicity.

To ask whether the SCGs identified in the previous section have any unique features in this network, we evaluated the degree to which each node is central in the network structure. In graph theory, 'centrality' is known as an index to measure how 'important' or 'influential' each node is. In particular, we employed a measure called 'degree centrality' (for weighted graphs) (**Nieminen, 1973**; **Segarra and Ribeiro, 2015**). Degree centrality, which is also called 'degree', simply measures 'influence' of a node on a network on the basis of links with its direct neighborhood. One can obtain a degree centrality value by calculating the sum of the weights of all the edges connected to each node (see also the definition of the degree matrix $D$ in **Equation 14** below). We note that in our graph, degree centrality vector $A\mathbf{1}_n(= D\mathbf{1}_n)$, where $\mathbf{1}_n$ is an $n$-dimensional column vector of which all elements are one, is equal to the eigenvector corresponding to the largest eigenvalue of a 'normalized' adjacency matrix $(D^{-1}A)^\top = AD^{-1}$ up to multiplication by a constant. From this perspective, the centrality index we adopted measures 'influence' of a node in a recursive manner depending on 'influence' of its neighboring nodes. A well-known example of this centrality indicator is Google's PageRank (**Brin and Page, 1998**) used for ranking web pages on the World Wide Web. It can be regarded as a variant of 'eigenvector centrality' (the eigenvector corresponding to the largest eigenvalue of the adjacency matrix $A$) (**Bonacich, 1972**; **Segarra and Ribeiro, 2015**). As explained in the main text, the protein species in the homeostatic core (the largest SCG) had high centrality scores, while those in the other condition-specific SCGs had low centrality scores (**Figure 5A**).

We directly observed the global stoichiometry conservation structure of this proteome graph using Laplacian eigenmaps (**Figures 5K, L and 6D**). In general, a graph can be uniquely specified not only by the adjacency matrix $A$, but also by the Laplacian matrix $L$ defined as

$$L := D - A, \tag{13}$$

where $D = (d_{ij})$ is the degree matrix with the components of

$$d_{ij} = \begin{cases} (A\mathbf{1}_n)_i & (i = j) \\ 0 & (i \neq j) \end{cases}, \tag{14}$$

where $\mathbf{1}_n$ is an $n$-dimensional column vector of which all elements are one. The $(i,i)$-element of $D$ represents the sum of the weights of all the edges connected to node $i$. In our case, it represents the sum of cosine similarity values between protein $i$ and the other proteins. To see the entire proteome graph structure, we specifically employed the normalized Laplacian,

$$L_{\mathrm{rw}} = D^{-1}L = I - D^{-1}A. \tag{15}$$

We remark that there are two types of often-used normalized Laplacian matrices, $L_{rw}$ and $L_{sym} = D^{-1/2}LD^{-1/2} = I - D^{-1/2}AD^{-1/2}$, in the field of machine learning (**von Luxburg, 2007**), and our mathematical analysis can provide a clear interpretation to each of them in the context of the Raman-proteome linear correspondence as described in Appendix 2.1.5.

There exist $m-1$ nontrivial eigenvalues of $L_{rw}$ that are greater than zero and less than one. We write these $m-1$ eigenvalues as $\lambda_{LE1}, \ldots, \lambda_{LE(m-1)}$ from the smallest and the corresponding eigenvectors as $v_{rw,1}, \ldots, v_{rw,(m-1)}$. Additionally, we denote the eigenvector corresponding to the eigenvalue zero as $v_{rw,0}$. Using these eigenvectors, one can construct a matrix $\tilde{V}_{rw} = \begin{bmatrix} v_{rw,0} & v_{rw,1} & \cdots & v_{rw,(m-1)} \end{bmatrix}$ and visualize a proteome, assigning protein $j$ with a coordinate specified by the elements after the second column in the $j$-th row of $\tilde{V}_{rw}$, i.e., by the $j$-th row of $\begin{bmatrix} v_{rw,1} & \cdots & v_{rw,(m-1)} \end{bmatrix}$. The csLE structure we illustrate in the main figures was produced by using these eigenvectors. For example, the csLE1-csLE2 figure in the main text (**Figure 6D**) is a scatterplot between $v_{rw,1}$ and $v_{rw,2}$. Note that the closer to one the cosine similarity of a protein pair is (the more similar their expression patterns are), the 'closer' the two protein species are placed (see Section 2.1.5 in Appendix for details).

This method of obtaining low-dimensional representation of data using eigenvectors of a graph Laplacian is known as Laplacian eigenmaps (LE) (**Belkin and Niyogi, 2001**; **Belkin and Niyogi, 2003**). Thus, what we explained above is the LE of a graph with edges weighted with cosine similarity of expression patterns of nodes (protein species). It differs from the original and common usages of LE in that the graph we considered is a complete graph (with loops) and that the weight of edges (pairwise similarity of nodes) is cosine similarity. It has made all the mathematical formulations linear, which allowed us to biologically interpret the results with mathematically rigorous analyses. We also remark that our graph representation of proteome does not rely on existing knowledge on the underlying interaction and regulatory networks of proteins and is based only on final expression levels of the proteins. Therefore, the results are robust against the uncertainty of underlying molecular detail.

## Relevance of centrality of csLE structure to biological functions

### Centrality-essentiality correlation

As mentioned in the main text, centrality of protein species with regard to stoichiometry conservation correlates with gene essentiality (**Figure 5B**). We analyzed the proteome data from all the 22 conditions reported by **Schmidt et al., 2016** in **Figure 5B**. Interestingly, the centrality-essentiality correlation becomes weaker when the analysis was conducted with the data from fewer conditions (**Appendix 1—figure 6A**).

We obtained the list of essential genes of *E. coli* from EcoCyc (**Keseler et al., 2017**) on September 23, 2020. The list contained 318 essential genes in total. The essentiality of the genes in this list was determined on the basis of whether single-gene knockouts of BW25113 (Keio Collection) could grow under LB condition at 37°C (**Baba et al., 2006**).

We also confirmed centrality-essentiality correlation for *S. pombe* transcriptome data (**Kobayashi-Kirschvink et al., 2018**; **Appendix 1—figure 6B**, see Appendix 3.2). For this analysis, we downloaded the list of essential genes of *S. pombe* from PomBase (**Harris et al., 2022**) on May 13, 2022. The list contained 1221 essential genes in total. Here, the essentiality data by PomBase was based on the Fission Yeast Phenotype Ontology terms 'inviable vegetative cell population' (FYPO:0002061) and 'viable vegetative cell population' (FYPO:0002060) (**Harris et al., 2013**). Note that in our *S. pombe* essentiality analysis, we focused only on coding genes, whereas the csLE structure was calculated using both coding and non-coding genes. See 'Centrality-coding/non-coding correlation' in Materials and methods and **Appendix 1—figure 6C** for the proportion of coding genes in each bin in **Appendix 1—figure 6B**. Eleven coding genes in the *S. pombe* transcriptome data were not found in current PomBase. Thus, some bins do not show 100% in total in **Appendix 1—figure 6B**.

Stoichiometry conservation centrality in human cells was evaluated using two kinds of *H. sapiens* transcriptome data: (i) human cell atlas data reported in **Cao et al., 2020** (**Figure 5F**) and (ii) genome-wide Perturb-seq data reported in **Replogle et al., 2022** (**Figure 5H**).

The human cell atlas data (**Cao et al., 2020**) contain gene expression profiles in cells from 15 fetal organs. To calculate stoichiometry conservation centrality from the human cell atlas, we analyzed the pseudobulk data (GSE156793_S4_gene_expression_tissue.txt provided at the Gene Expression Omnibus). We calculated stoichiometry conservation centrality value of each gene using expression level data of 53,908 genes that are expressed at least in one organ.

The Perturb-seq data we used are gene expression profiles in a chronic myeloid leukemia cell line K562 (*Replogle et al., 2022*). This dataset contains single-cell RNA sequencing data of genetically perturbed cells in which expression of targeted genes is inhibited by CRISPR interference. We analyzed the pseudobulk data (K562_gwps_raw_bulk_01.h5ad provided at Figshare) to calculate stoichiometry conservation centrality. We evaluated stoichiometry conservation centrality value of each gene using the expression data of all the 8248 genes in the Perturb-seq data. We remark that this dataset did not contain genes that showed no expression under all the reported genetic perturbation conditions.

Human gene essentiality was determined by referring to another dataset reported in *Wang et al., 2015*, in which fitness cost imposed by gene inactivation was evaluated by a CRISPR-based method (*Wang et al., 2015*). The fitness cost was quantified by an index called CRISPR score; genes with lower CRISPR scores are considered more essential (*Wang et al., 2015*). We used the CRISPR scores calculated with a human chronic myelogenous leukemia cell line KBM7.

The CRISPR scores of 16,996 genes and 7462 genes were found in *Wang et al., 2015* among the genes whose stoichiometry conservation centrality was evaluated using the human cell atlas data (*Cao et al., 2020*) and the Perturb-seq data (*Replogle et al., 2022*), respectively. We evaluated the correlations between stoichiometry conservation centrality and gene essentiality (CRISPR scores) for these common genes in *Figure 5F and H*. The correlations were examined with the Brunner-Munzel test (*Brunner and Munzel, 2000*) using R (version 4.0.3) and 'brunnermunzel' package (version 2.0) (*Ara, 2022*).

## Centrality-evolutionary conservation correlation

As mentioned in the main text, centrality of proteins with regard to expression stoichiometry conservation weakly correlates with evolutionary conservation represented by the number of orthologs based on protein sequences (*Figure 5C*). In *Figure 5C*, we analyzed the proteome data from all the 22 conditions reported in *Schmidt et al., 2016*. We also confirmed the relation for the *E. coli* proteome data from fewer conditions which we had used for our Raman-proteome correspondence analyses (*Appendix 1—figure 6D*).

We obtained the ortholog data from OrthoMCL-DB (*Chen et al., 2006*) (release 6.12). We used the number of orthologs in all of the 'Core species' and the 'Peripheral species' of OrthoMCL, which are across the three domains (Bacteria, Archaea, and Eukaryota), as a proxy for evolutionary conservation of each protein. To examine the correlation, we performed the Brunner-Munzel test (*Brunner and Munzel, 2000*) using R (version 4.0.3) and 'brunnermunzel' package (version 2.0) (*Ara, 2022*). The *E. coli* proteome data contain 15 proteins with IDs that were not found in OrthoMCL-DB for technical reasons such as changes in IDs in the past, and thus, we manually processed these 15 proteins.

We also examined *S. pombe* transcriptome data (*Kobayashi-Kirschvink et al., 2018*; *Appendix 1—figure 6E–G*, see Appendix 3.2). We obtained ortholog data from OrthoMCL-DB (*Chen et al., 2006*) (release 6.12). The *S. pombe* transcriptome data have 11 coding genes which were not found in both current PomBase and OrthoMCL-DB, and two coding genes which were found in PomBase but not in OrthoMCL-DB. The *S. pombe* transcriptome data contain not only coding genes but also non-coding genes, and we obtained the csLE structure using both.

We also evaluated stoichiometry conservation-evolutionary conservation correlation using the human cell atlas data (*Cao et al., 2020*; *Figure 5G*) and the genome-wide Perturb-seq data (*Replogle et al., 2022*; *Figure 5I*). Ortholog data for these analyses were obtained from OrthoMCL-DB (release 6.20). We found the ortholog data in OrthoMCL-DB for 18,959 genes among the 53,908 genes with stoichiometry conservation centrality evaluated with the human cell atlas data. We remark that 98.7% of the 18,959 genes were classified as coding genes in the human cell atlas data. We also found the ortholog data for 7957 genes among the 8248 genes with stoichiometry conservation centrality evaluated with the Perturb-seq data. The correlations were examined with the Brunner-Munzel test (*Brunner and Munzel, 2000*) using R (version 4.0.3) and 'brunnermunzel' package (version 2.0) (*Ara, 2022*).

## Centrality-coding/non-coding correlation

As mentioned in the main text and 'Centrality-essentiality correlation' in Materials and methods, centrality of genes with regard to stoichiometry conservation clearly correlates with coding/non-coding classification of genes in *S. pombe*. We observed this trend using *S. pombe* transcriptome data

(*Kobayashi-Kirschvink et al., 2018*; *Appendix 1—figure 6C*). The coding/non-coding assignment of each gene is based on PomBase (*Harris et al., 2022*) data downloaded on October 11, 2022.

We observed a comparable correlation even in the human cell atlas data (*Figure 5E*). The gene type assignment is based on the human cell atlas data. Note that almost all the genes in the Perturb-seq data were coding genes.

## Global omics structures characterized by Raman-omics correspondences

### Notation

Let $\hat{\boldsymbol{b}}_j$ denote the $j$-th row in $B$ (see *Equation 3*). It is an $m$-dimensional row vector whose components represent coefficients of protein $j$. The first component is the constant term, and the $i$-th component is the coefficient for LDA($i-1$) Raman. Below, we consider the coefficients normalized with the constant terms,

$$\hat{\boldsymbol{b}}_j^{\mathrm{norm}} := \begin{pmatrix} 1 & \dfrac{b_{j1}}{b_{j0}} & \cdots & \dfrac{b_{j(m-1)}}{b_{j0}} \end{pmatrix}. \tag{16}$$

### Raman-proteome correspondence matrix as a low-dimensional representation of proteome changes

We asked whether the stoichiometry conservation structure of the proteomes revealed by LE (*Figures 5K, L and 6D*) is relevant to the low-dimensional Raman LDA space. To address this, we focused on a proteome low-dimensional structure specified by the Raman-proteome coefficients, motivated by the fact that the analysis of $B$ led to the discovery of a proteome fraction that conserves mutual stoichiometry (*Figure 3*). We considered a space where $\hat{\boldsymbol{b}}_j^{\mathrm{norm}}$ represents the coordinate of each protein. From *Equation 7* or *Equation 8*, protein species whose abundance ratios remain constant have an identical coordinate in this normalized coefficient space. The proteome in this normalized coefficient space is shown in *Figure 6A and C*.

This structure (*Figure 6A and C*) is constructed using the Raman LDA axes (dual basis) and is different from the csLE structure (*Figures 5K, L and 6D*), which is independent of Raman information. Therefore, it is nontrivial that these two structures are similar. This similarity suggests that differences in cellular Raman spectra captured by LDA might be quantitatively related to the omics structure deduced from stoichiometry-conserving relations. We will mathematically analyze this similarity in Section 2 in Appendix.

## Evaluating similarity between orthogonal matrix Θ and identity matrix

As we see in Appendix 2.1.5, an orthogonal matrix Θ that appears in the relation connecting the two types of proteome structure must be close to an identity matrix to guarantee the structural similarity. To evaluate to what extent Θ is close to an identity matrix, we generated many random orthogonal matrices (*Mezzadri, 2006*) and compared Θ and the identity matrix with them.

We first multiplied each orthogonal matrix by itself in the sense of Hadamard product (element-wise product). Then, we regarded the resultant matrix as a scatterplot (*Appendix 1—figure 9B*) and calculated its Pearson correlation coefficient, assuming that $(i, j)$ element was the frequency of 'data points' at the coordinate $(i, j)$. The obtained Pearson correlation coefficient can be regarded as a measure of closeness to the identity matrix. In the case of the identity matrix, the correlation coefficient takes the maximum value, one, because non-zero values are concentrated on the diagonal part.

We calculated the square of each matrix in the sense of Hadamard product for two reasons. First, since all elements of the resultant matrix are non-negative, one can ensure that the number of 'points' is non-negative at any coordinate. Note that it is not necessarily an integer here. Second, the sum of all the elements of the resultant matrix is necessarily $m$. Thus, the total number of 'points' is equally $m$ for any $m \times m$ orthogonal matrices compared.

In addition to this method, we also evaluated the closeness of Θ to the identity matrix (i) by comparing the magnitudes of off-diagonal elements among Θ, the identity matrix, and random orthogonal matrices, and (ii) by comparing the magnitudes of elements of leading principal submatrices among Θ, the identity matrix, and random orthogonal matrices. In (i), from a part consisting of $(m-1)$- and $-(m-1)$-diagonals ($(1, m)$ and $(m, 1)$ elements) to the whole matrix, we expand step by

step the area to consider by including $i$- and $-i$-diagonals ($m - 1 \geq i \geq 0$, the final step is inclusion of the main diagonal), and calculated the sum of the square of the elements in the area at each step. In (ii), from the smallest leading principal submatrix (($1, 1$) element) to the whole matrix, we expand the area to consider step by step and calculate the sum of the square of the elements in the area at each step. See also schematic diagrams in the figures in Appendix (e.g. *Appendix 1—figure 9D and E*).

## Interpretation of $L^1$ norm/$L^2$ norm ratio of an expression vector as a quantitative measure of expression generality

In Appendix 2.1.5, we will also see that even if $\Theta$ is close to the identity matrix, there is another condition which must be met to guarantee the similarity of the two types of proteome structure. By considering the mathematics behind the condition, we will reveal that the two indices, stoichiometry conservation centrality (degree) $d_j = \sum_i \cos \theta_{\boldsymbol{p}_i \boldsymbol{p}_j}$ and expression generality score $g_j = \left\| \boldsymbol{p}_j \right\|_1 / \left\| \boldsymbol{p}_j \right\|_2$, must be mutually proportional. Here, we explain why $g_j$ is a quantitative measure of the generality (or constancy) of expression levels.

First, we note that the ratio $\left\| \boldsymbol{p}_j \right\|_1 / \left\| \boldsymbol{p}_j \right\|_2$ is independent of the magnitude of the expression vector $\boldsymbol{p}_j$. In other words, normalization does not affect the ratio:

$$\frac{\left\| \dfrac{\boldsymbol{p}_j}{\|\boldsymbol{p}_j\|_2} \right\|_1}{\left\| \dfrac{\boldsymbol{p}_j}{\|\boldsymbol{p}_j\|_2} \right\|_2} = \frac{\sum_{i=1}^{m} \dfrac{|p_{ij}|}{\|\boldsymbol{p}_j\|_2}}{\sqrt{\sum_{i=1}^{m} \dfrac{(p_{ij})^2}{\|\boldsymbol{p}_j\|_2^2}}} = \frac{\dfrac{\|\boldsymbol{p}_j\|_1}{\|\boldsymbol{p}_j\|_2}}{\dfrac{\|\boldsymbol{p}_j\|_2}{\|\boldsymbol{p}_j\|_2}} = \frac{\|\boldsymbol{p}_j\|_1}{\|\boldsymbol{p}_j\|_2}. \tag{17}$$

On the basis of this, we only consider normalized expression vectors $\boldsymbol{p}_j / \left\| \boldsymbol{p}_j \right\|_2$ without loss of generality.

By definition, $L^2$ norm of a normalized expression vector (the denominator of the most left-hand side of *Equation 17*) equals one. Thus, the ratio we are considering equals the $L^1$ norm of the normalized expression vector:

$$\frac{\|\boldsymbol{p}_j\|_1}{\|\boldsymbol{p}_j\|_2} = \left\| \frac{\boldsymbol{p}_j}{\|\boldsymbol{p}_j\|_2} \right\|_1. \tag{18}$$

Here, we write

$$\frac{\boldsymbol{p}_j}{\|\boldsymbol{p}_j\|_2} = \begin{pmatrix} \tilde{p}_{1j} \\ \tilde{p}_{2j} \\ \vdots \\ \tilde{p}_{mj} \end{pmatrix}, \tag{19}$$

where $\tilde{p}_{1j}, \tilde{p}_{2j}, \ldots, \tilde{p}_{mj} \geq 0$. Then,

$$\frac{\|\boldsymbol{p}_j\|_1}{\|\boldsymbol{p}_j\|_2} = \left\| \frac{\boldsymbol{p}_j}{\|\boldsymbol{p}_j\|_2} \right\|_1 = \sum_{i=1}^{m} \tilde{p}_{ij}. \tag{20}$$

Note that $\sum_{i=1}^{m} \left( \tilde{p}_{ij} \right)^2 = 1$ holds because of normalization. Therefore, any normalized expression vector corresponds to a point on the first orthant division of the unit ($m - 1$)-sphere $\sum_{i=1}^{m} \left( x_i \right)^2 = 1$ ($x_1, \ldots, x_m \geq 0$).

Next, we consider a hyperplane $\sum_{i=1}^{m} x_i = k$ which passes through the point in *Equation 19*. Since all the coefficients of this hyperplane are equal, all the $m$ intercepts are also equal. The intercept value is $k = \sum_{i=1}^{m} \tilde{p}_{ij}$, thus equals the ratio in *Equation 20*. In other words, the ratio from *Equation 20* appears as an intercept of the hyperplane passing through the point corresponding to the normalized vector $\boldsymbol{p}_j / \left\| \boldsymbol{p}_j \right\|_2$ with all the coefficients equal to one.

By simple calculation, one can see that the two surfaces $\sum_{i=1}^{m} (x_i)^2 = 1$ $(x_1, \ldots, x_m \geq 0)$ and $\sum_{i=1}^{m} x_i = k$ intersect when $1 \leq k \leq \sqrt{m}$. (In other words, $\|\boldsymbol{p}_j\|_2 \leq \|\boldsymbol{p}_j\|_1 \leq \sqrt{m} \|\boldsymbol{p}_j\|_2$ holds.) The intercept value $k = \sum_{i=1}^{m} \tilde{p}_{ij}$ takes the maximum $k = \sqrt{m}$ when the normalized expression vector points to the 'center' of the first orthant division of the unit $(m-1)$-sphere, i.e., when

$$\frac{\boldsymbol{p}_j}{\|\boldsymbol{p}_j\|_2} = \begin{pmatrix} \tilde{p}_{1j} \\ \tilde{p}_{2j} \\ \vdots \\ \tilde{p}_{mj} \end{pmatrix} = \frac{1}{\sqrt{m}} \begin{pmatrix} 1 \\ 1 \\ \vdots \\ 1 \end{pmatrix}. \tag{21}$$

This means that the expression level is even and constant across the conditions. When this evenness of expression level breaks, the intercept value $k$ decreases, and it attains the minimum $k = 1$ when the normalized expression vector overlaps with an axis, i.e., when

$$\frac{\boldsymbol{p}_j}{\|\boldsymbol{p}_j\|_2} = \begin{pmatrix} \tilde{p}_{1j} \\ \vdots \\ \tilde{p}_{(\mu-1)j} \\ \tilde{p}_{\mu j} \\ \tilde{p}_{(\mu+1)j} \\ \vdots \\ \tilde{p}_{mj} \end{pmatrix} = \begin{pmatrix} 0 \\ \vdots \\ 0 \\ 1 \\ 0 \\ \vdots \\ 0 \end{pmatrix}, \tag{22}$$

which corresponds to a completely 'condition-specific expression pattern' (μ is the condition's index).

See *Appendix 1—figure 8A and B* for a graphical explanation of the argument for the two- and three-dimensional cases.

## Proteome structure obtained with PCA

As mentioned in the main text, we confirmed that PCA could find a proteome structure (*Appendix 1—figure 6H*) similar to the csLE structure (*Figure 5L* and *Figure 6D*). Since cosine similarity of expression vectors is inner product of the $L^2$-normalized expression vectors, we also performed $L^2$ normalization of proteome data before applying PCA in this analysis. In other words, we applied PCA to a normalized proteome data $[\boldsymbol{p}_1 / \|\boldsymbol{p}_1\|_2 \cdots \boldsymbol{p}_n / \|\boldsymbol{p}_n\|_2]$ (see Appendix 2.1.5).

We remark that, despite the structural similarity between the PCA structure and the csLE structure, csLE has an advantage over PCA in that the relative proximity of positions reflects the strength of stoichiometry conservation between each element. In addition, as shown in the main text and Section 2 in Appendix, csLE of omics data has a direct quantitative connection to cellular Raman spectra, which is not the case for PCA.

## Acknowledgements

We thank Matthias Heinemann and Silke Bonsing-Vedelaar for detailed information on the *E. coli* culture conditions; Doeke R Hekstra, Tetsuya J Kobayashi, Takafumi Miyamoto, John Russell, and Ian Hunt-Isaak for reading the manuscript and providing critical comments; Kunihiko Kaneko, Chikara Furusawa, Yasushi Okada, and members of the Wakamoto Lab and the Universal Biology Institute for discussion and encouragement. This work was supported by JST CREST Grant Number JPMJCR1927 (YW); JST ERATO Grant Number JPMJER1902 (YW); JSPS KAKENHI Grant Numbers 19J22448 (KFK) and 21K20672 (TN).

## Additional information

### Competing interests
Koseki J Kobayashi-Kirschvink, Yuichi Wakamoto: Inventor on patents (JP6993682 and US10,379,052 B2) filed by The University of Tokyo. The other authors declare that no competing interests exist.

### Funding

| Funder | Grant reference number | Author |
|---|---|---|
| Japan Science and Technology Agency | JPMJCR1927 | Yuichi Wakamoto |
| Japan Science and Technology Agency | JPMJER1902 | Yuichi Wakamoto |
| Japan Society for the Promotion of Science | 19J22448 | Ken-ichiro F Kamei |
| Japan Society for the Promotion of Science | 21K20672 | Takashi Nozoe |

The funders had no role in study design, data collection and interpretation, or the decision to submit the work for publication.

### Author contributions
Ken-ichiro F Kamei, Conceptualization, Data curation, Software, Formal analysis, Funding acquisition, Validation, Investigation, Methodology, Writing – original draft, Writing – review and editing; Koseki J Kobayashi-Kirschvink, Hidenori Nakaoka, Conceptualization, Writing – review and editing; Takashi Nozoe, Formal analysis, Funding acquisition, Investigation, Writing – review and editing; Miki Umetani, Investigation, Writing – review and editing; Yuichi Wakamoto, Conceptualization, Supervision, Funding acquisition, Writing – original draft, Writing – review and editing

### Author ORCIDs
Ken-ichiro F Kamei ⓘ https://orcid.org/0009-0002-8026-4454
Koseki J Kobayashi-Kirschvink ⓘ https://orcid.org/0000-0001-6590-3823
Takashi Nozoe ⓘ https://orcid.org/0000-0003-2556-6484
Hidenori Nakaoka ⓘ https://orcid.org/0000-0001-8465-5853
Miki Umetani ⓘ https://orcid.org/0000-0002-3171-4327
Yuichi Wakamoto ⓘ https://orcid.org/0000-0002-6233-0844

Reviewer #1 (Public review): https://doi.org/10.7554/eLife.101485.3.sa1
Author response https://doi.org/10.7554/eLife.101485.3.sa2

## Additional files

### Supplementary files
MDAR checklist

### Data availability
All data and analysis codes have been deposited in Zenodo and are publicly available at https://doi.org/10.5281/zenodo.17090710.

The following dataset was generated:

| Author(s) | Year | Dataset title | Dataset URL | Database and Identifier |
|---|---|---|---|---|
| Kamei KF, Kobayashi-Kirschvink KJ, Nozoe T, Nakaoka H, Umetani M, Wakamoto Y | 2025 | Code and data for "Revealing global stoichiometry conservation architecture in cells from Raman spectral patterns" | https://doi.org/10.5281/zenodo.17090710 | Zenodo, 10.5281/zenodo.17090710 |

The following previously published datasets were used:

| Author(s) | Year | Dataset title | Dataset URL | Database and Identifier |
|---|---|---|---|---|
| Kobayashi-Kirschvink K, Nakaoka H, Oda A, Kamei KF, Nosho K, Fukushima H, Kanesaki Y, Yajima S, Masaki H, Ohta K, Wakamoto Y | 2018 | Data for: Linear Regression Links Transcriptomic Data and Cellular Raman Spectra | https://doi.org/10.17632/2fx3h2rx2m.1 | Mendeley Data, 10.17632/2fx3h2rx2m.1 |
| Cao J, O'Day DR, Pliner HA, Kingsley P, Deng M, Daza RM, Zager MA, Kimberly A, Blecher R, Zhang F, O'Day DR, Spielmann M, Palis J, Doherty D, Steemers FJ, Glass IA, Trapnell C, Shendure J | 2020 | A human cell atlas of fetal gene expression | https://www.ncbi.nlm.nih.gov/geo/query.acc.cgi?acc=GSE156793 | NCBI Gene Expression Omnibus, GSE156793 |
| Replogle J, Weissman J | 2022 | "Mapping information-rich genotype-phenotype landscapes with genome-scale Perturb-seq" Replogle et al. 2022 processed Perturb-seq datasets | https://doi.org/10.25452/figshare.plus.20029387.v1 | figshare, 10.25452/figshare.plus.20029387.v1 |
| Keseler IM | 2017 | The EcoCyc database | https://ecocyc.org/ | The EcoCyc, Version 24.1 |

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

## Appendix 1

### 1 Materials and methods

See Materials and methods section in the main text.

### 2 Mathematical analysis and details

To clarify what is nontrivial in the correspondence between LDA Raman and csLE proteome, here we derive rigorous mathematical relations for the correspondence through linear algebraic calculation.

#### 2.1 Mathematics behind the correspondence

#### 2.1.1 Notations

We use the following notations:

- $\mathbf{1}_x$ denotes an $x$-dimensional column vector of ones, and $\mathbf{0}_x$ does an $x$-dimensional zero column vector.
- A vector without a hat (e.g. $\boldsymbol{x}$) is a column vector, and a vector with a hat (e.g. $\hat{\boldsymbol{x}}$) is a row vector.
- $I$ denotes an identity matrix, and $O$ does a zero matrix.
- For a square matrix $X = (x_{ij})$, $\mathrm{diag}(X) = (\delta_{ij}x_{ij})$, and for a vector $\boldsymbol{x} = (x_i)$, $\mathrm{diag}(\boldsymbol{x}) = (\delta_{ij}x_i)$, where $\delta_{ij}$ is the Kronecker delta.
- For a matrix $X$, $X\left[i,:\right]$ denotes the $i$-th row of $X$, and $X\left[:,j\right]$ denotes the $j$-th column of $X$.

#### 2.1.2 Preparations

Let $l$ be the number of cells in each condition, $m$ be the number of conditions, $n$ be the original dimension of proteome, i.e., the number of protein species in the proteome data, and $s'$ be the original dimension of Raman spectra after the application of the Savitzky-Golay filter. In our main data, $l = 38, m = 15, n = 2058$, and $s' = 599$.

#### Raman Spectra

Original preprocessed Raman data: Let

$$\hat{\boldsymbol{x}}_j^{\prime(i)} = \left( x_{j1}^{\prime(i)} \cdots x_{js'}^{\prime(i)} \right) \tag{2.1}$$

be a preprocessed Raman spectrum from cell $j$ under condition $i$ (see 'Raman measurements and preprocessing of spectra' in Materials and methods). The prime ′ denotes that the variable is the original preprocessed data. The $\hat{\boldsymbol{x}}_j^{\prime(i)}(1 \leq j \leq l)$ are collected in an $l \times s'$ matrix:

$$X_i' = \begin{bmatrix} \hat{\boldsymbol{x}}_1^{\prime(i)} \\ \vdots \\ \hat{\boldsymbol{x}}_l^{\prime(i)} \end{bmatrix} = \left( x_{jk}^{\prime(i)} \right)_{1 \leq j \leq l, 1 \leq k \leq s'}. \tag{2.2}$$

Combining $X_i'$ from different conditions, one can define an $(lm) \times s'$ matrix

$$X' = \begin{bmatrix} X_1' \\ \vdots \\ X_m' \end{bmatrix}, \tag{2.3}$$

which contains all the preprocessed Raman data.

PCA and centering: Before LDA, PCA was first applied to the preprocessed Raman data to reduce noise. The covariance matrix is

$$C_{X'} := \frac{1}{lm - 1} \left( X' - \frac{1}{lm}\mathbf{1}_{lm}(\mathbf{1}_{lm})^\top X' \right)^\top \left( X' - \frac{1}{lm}\mathbf{1}_{lm}(\mathbf{1}_{lm})^\top X' \right), \tag{2.4}$$

which is positive semi-definite. PCA is formulated as the following eigenvalue problem of $C_{X'}$:

$$C_{X'} V_{\text{PCA}} = V_{\text{PCA}} \Lambda_{\text{PCA}}, \tag{2.5}$$

where $\Lambda_{\text{PCA}}$ is a diagonal matrix with the eigenvalues of $C_{X'}$ as its diagonal elements in decreasing order from the upper left, and $V_{\text{PCA}}$ is an orthogonal matrix consisting of the eigenvectors of $C_{X'}$ as its columns. Using the first $s$ ($1 \leq s \leq s'$) columns of $V_{\text{PCA}} = \begin{bmatrix} w_1 & \cdots & w_{s'} \end{bmatrix}$, i.e., the columns corresponding to the first $s$ largest eigenvalues, we obtain an $(lm) \times s$ matrix representing the post PCA data:

$$X := \left( X' - \frac{1}{lm} \mathbf{1}_{lm} \left( \mathbf{1}_{lm} \right)^{\top} X' \right) \begin{bmatrix} w_1 & \cdots & w_s \end{bmatrix}. \tag{2.6}$$

Here, $w_k$ is the $k$-th PCA coefficient vector, and $s$ is the reduced dimension of the Raman spectra. The subtraction of the $\frac{1}{lm} \mathbf{1}_{lm} (\mathbf{1}_{lm})^{\top} X'$ is for centering the data. In our case, the top 218 principal components (i.e. $s = 218$) explaining 98% of the variance were used to reduce noise and dimensionality.

Let $\hat{x}_j^{(i)}$ be the post PCA Raman spectrum from cell $j$ under condition $i$, i.e.,

$$\hat{x}_j^{(i)} = \begin{pmatrix} x_{j1}^{(i)} & \cdots & x_{js}^{(i)} \end{pmatrix}, \tag{2.7}$$

and $X_i$ be the collection of $\hat{x}_j^{(i)}$ ($1 \leq j \leq l$), i.e.,

$$X_i = \begin{bmatrix} \hat{x}_1^{(i)} \\ \vdots \\ \hat{x}_l^{(i)} \end{bmatrix} = \left( x_{jk}^{(i)} \right)_{1 \leq j \leq l, 1 \leq k \leq s}. \tag{2.8}$$

Then, $X$ is written as

$$X = \begin{bmatrix} X_1 \\ \vdots \\ X_m \end{bmatrix}. \tag{2.9}$$

From *Equation 2.6*,

$$\left( \mathbf{1}_{lm} \right)^{\top} X = \mathbf{0}, \tag{2.10}$$

namely, for any $k$,

$$\sum_{i=1}^{m} \sum_{j=1}^{l} x_{jk}^{(i)} = 0, \tag{2.11}$$

which means that the post PCA data is centered.

Population average in each condition: Let $\hat{x}_i$ be the population average of the post PCA spectra of cells under condition $i$. Then,

$$\hat{x}_i = \frac{1}{l} \left( \mathbf{1}_l \right)^{\top} X_i. \tag{2.12}$$

Also,

$$\hat{x}_i = \begin{pmatrix} \bar{x}_{i1} & \cdots & \bar{x}_{is} \end{pmatrix}, \tag{2.13}$$

where

$$\bar{x}_{ik} = \frac{1}{l} \sum_{j=1}^{l} x_{jk}^{(i)}.$$  (2.14)

We define an $m \times s$ matrix

$$\bar{X} = \begin{bmatrix} \hat{x}_1 \\ \vdots \\ \hat{x}_m \end{bmatrix} = (\bar{x}_{ik})_{1 \leq i \leq m, 1 \leq k \leq s} \ .$$  (2.15)

Each row of $\bar{X}$ corresponds to a condition. From *Equation 2.11* and *Equation 2.14*, for any $k$,

$$\sum_{i=1}^{m} \bar{x}_{ik} = 0.$$  (2.16)

Namely,

$$\left(\mathbf{1}_m\right)^\top \bar{X} = \mathbf{0}.$$  (2.17)

These relations mean that $\bar{X}$ is also centered.

LDA: The within-class covariance matrix is

$$C_I := \frac{1}{m} \sum_{i=1}^{m} \frac{1}{l-1} \left( X_i - \frac{1}{l}\mathbf{1}_l \left(\mathbf{1}_l\right)^\top X_i \right)^\top \left( X_i - \frac{1}{l}\mathbf{1}_l \left(\mathbf{1}_l\right)^\top X_i \right)$$  (2.18)

$$= \frac{1}{m} \sum_{i=1}^{m} \frac{1}{l-1} \left( X_i - \mathbf{1}_l\hat{x}_i \right)^\top \left( X_i - \mathbf{1}_l\hat{x}_i \right).$$  (2.19)

Here, *Equation 2.12* was used. The between-class covariance matrix is

$$C_B := \frac{1}{m-1} \left( \bar{X} - \frac{1}{m}\mathbf{1}_m \left(\mathbf{1}_m\right)^\top \bar{X} \right)^\top \left( \bar{X} - \frac{1}{m}\mathbf{1}_m \left(\mathbf{1}_m\right)^\top \bar{X} \right)$$  (2.20)

$$= \frac{1}{m-1}\bar{X}^\top \bar{X}.$$  (2.21)

Here, *Equation 2.17* was used. Assume $\text{rank}(C_I) = s$ and $\text{rank}(C_B) = m - 1$ (the maximum possible values). In fact, $\text{rank}(C_I) = s$ and $\text{rank}(C_B) = m - 1$ in our data. Note that $s > m - 1$. From the definitions of $C_I$ and $C_B$ above, both are positive semi-definite. LDA is formulated as the following generalized eigenvalue problem:

$$C_B V_{\text{LDA}} = C_I V_{\text{LDA}} \Lambda'_{\text{LDA}},$$  (2.22)

where $\Lambda'_{\text{LDA}}$ is a diagonal matrix, and $V_{\text{LDA}}$ is an $s \times (m-1)$ matrix that simultaneously diagonalizes $C_B$ and $C_I$ (to $\Lambda_{\text{LDA}}$):

$$\left(V_{\text{LDA}}\right)^\top C_B V_{\text{LDA}} = \left(V_{\text{LDA}}\right)^\top C_I V_{\text{LDA}} \Lambda'_{\text{LDA}} = \Lambda_{\text{LDA}}.$$  (2.23)

Here, the diagonal elements in $\Lambda'_{\text{LDA}}$ were in decreasing order from the upper left. In our analysis, the columns of $V_{\text{LDA}}$ were normalized.

Using $V_{\text{LDA}}$, we obtain an $m \times (m-1)$ matrix representing the post LDA data

$$R := \bar{X} V_{\text{LDA}}.$$  (2.24)

Each row of $R$ represents a dimension-reduced Raman spectrum of each condition. Let us write the $h$-th $(1 \leq h \leq m - 1)$ column of $R$ as

$$\boldsymbol{r}_h := \begin{pmatrix} r_{1h} \\ \vdots \\ r_{mh} \end{pmatrix}. \tag{2.25}$$

Then,

$$R = \begin{bmatrix} \boldsymbol{r}_1 & \cdots & \boldsymbol{r}_{m-1} \end{bmatrix} = (r_{ih})_{1 \le i \le m, 1 \le h \le m-1}. \tag{2.26}$$

Transforming $R^\top R$ gives

$$R^\top R = (V_{\mathrm{LDA}})^\top \bar{X}^\top \bar{X} V_{\mathrm{LDA}} \tag{2.27}$$

$$= (m-1)(V_{\mathrm{LDA}})^\top C_B V_{\mathrm{LDA}} \quad (\because 2.20) \tag{2.28}$$

$$= (m-1)\Lambda_{\mathrm{LDA}} \quad (\because 2.23). \tag{2.29}$$

Therefore, $R^\top R$ is a diagonal matrix, and $\boldsymbol{r}_h$ are orthogonal to each other. As all the diagonal elements of the diagonal matrix $\Lambda_{\mathrm{LDA}}$ is positive, $\mathrm{rank}(R) = m - 1$. Furthermore,

$$(\mathbf{1}_m)^\top R = (\mathbf{1}_m)^\top \bar{X} V_{\mathrm{LDA}} \tag{2.30}$$

$$= (\mathbf{0}_{m-1})^\top \quad (\because 2.17). \tag{2.31}$$

Namely, for any $h$,

$$\sum_{i=1}^m r_{ih} = 0. \tag{2.32}$$

This means that, as the data is centered, all the columns of $R$ are perpendicular to $\mathbf{1}_m$.

## Proteome

Let

$$\boldsymbol{p}_j = \begin{pmatrix} p_{1j} \\ \vdots \\ p_{mj} \end{pmatrix} \tag{2.33}$$

be the absolute abundances of the $j$-th protein. $\boldsymbol{p}_j$ are collected in an $m \times n$ matrix:

$$P := \begin{bmatrix} \boldsymbol{p}_1 & \cdots & \boldsymbol{p}_n \end{bmatrix} = (p_{ij})_{1 \le i \le m, 1 \le j \le n}. \tag{2.34}$$

$p_{ij}$ is the absolute abundance of the $j$-th protein in condition $i$.

We assume $\mathrm{rank}(P) = m$, i.e., proteome vectors for different conditions are linearly independent. Actually, $\mathrm{rank}(P) = m$ in our data. We also assume that proteins with zero expression in all the $m$ conditions had been excluded from the proteome data.

### 2.1.3 Linear transformation between LDA Raman and proteome

We define an $m \times m$ matrix

$$R_E := \begin{bmatrix} \mathbf{1}_m & R \end{bmatrix} = \begin{bmatrix} \mathbf{1}_m & \boldsymbol{r}_1 & \cdots & \boldsymbol{r}_{m-1} \end{bmatrix} \tag{2.35}$$

We denote the first column of $R_E$ as $\boldsymbol{r}_0$. Hence,

$$\boldsymbol{r}_0 = \mathbf{1}_m. \tag{2.36}$$

From *Equation 2.29* and *Equation 2.31*,

$$\left(R_E\right)^\top R_E = \begin{bmatrix} (\mathbf{1}_m)^\top \\ R^\top \end{bmatrix} \begin{bmatrix} \mathbf{1}_m & R \end{bmatrix} = \begin{bmatrix} m & \mathbf{0} \\ \mathbf{0} & R^\top R \end{bmatrix} = \begin{bmatrix} m & \mathbf{0} \\ \mathbf{0} & (m-1)\Lambda_{\mathrm{LDA}} \end{bmatrix}. \tag{2.37}$$

Therefore, $\left(R_E\right)^\top R_E$ is also a diagonal matrix, and $R_E$ has full rank. For convenience, we write

$$\left(R_E\right)^\top R_E = \Lambda_{R_E} = \begin{pmatrix} m & & & \\ & (m-1)\lambda_{\mathrm{LDA1}} & & \\ & & \ddots & \\ & & & (m-1)\lambda_{\mathrm{LDA}(m-1)} \end{pmatrix}. \tag{2.38}$$

Here, we consider singular value decomposition (SVD) of $R_E$; i.e.;

$$R_E = U_{R_E} \Sigma_{R_E} \left(V_{R_E}\right)^\top, \tag{2.39}$$

where $\Sigma_{R_E}$ is a diagonal matrix whose diagonal elements are the singular values of $R_E$, and $\left(U_{R_E}\right)^\top U_{R_E} = \left(V_{R_E}\right)^\top V_{R_E} = I$. Note that we can set $V_{R_E} = I$ in the following way. Let

$$U_{R_E} := R_E(\Lambda_{R_E})^{-1/2}. \tag{2.40}$$

Then,

$$(U_{R_E})^\top U_{R_E} = I, \tag{2.41}$$

$$(U_{R_E})^\top R_E = (\Lambda_{R_E})^{1/2}. \tag{2.42}$$

Thus, SVD of $R_E$ can be written as

$$R_E = U_{R_E}(\Lambda_{R_E})^{1/2}I. \tag{2.43}$$

As $\Lambda_{R_E}$ is the eigenvalue matrix of $(R_E)^\top R_E$,

$$\left(R_E\right)^\top R_E = \Lambda_{R_E} = \left(\Sigma_{R_E}\right)^2 \tag{2.44}$$

and

$$\Sigma_{R_E} = \begin{pmatrix} \sqrt{m} & & & \\ & \sqrt{(m-1)\lambda_{\mathrm{LDA1}}} & & \\ & & \ddots & \\ & & & \sqrt{(m-1)\lambda_{\mathrm{LDA}(m-1)}} \end{pmatrix}. \tag{2.45}$$

Now, we consider linear transformation between $P$ and $R_E$. We introduce the $n \times m$ coefficient matrix $B_E = \begin{bmatrix} \boldsymbol{b}_0 \cdots \boldsymbol{b}_{m-1} \end{bmatrix}$ that connects $P$ and $R_E$ as

$$P = R_E \left(B_E\right)^\top \tag{2.46}$$

$$= \begin{bmatrix} \boldsymbol{r}_0 & \boldsymbol{r}_1 & \cdots & \boldsymbol{r}_{m-1} \end{bmatrix} \begin{bmatrix} (\boldsymbol{b}_0)^\top \\ (\boldsymbol{b}_1)^\top \\ \vdots \\ (\boldsymbol{b}_{m-1})^\top \end{bmatrix} \tag{2.47}$$

$$= \mathbf{1}_m \left( \boldsymbol{b}_0 \right)^\top + \boldsymbol{r}_1 \left( \boldsymbol{b}_1 \right)^\top + \cdots + \boldsymbol{r}_{m-1} \left( \boldsymbol{b}_{m-1} \right)^\top. \tag{2.48}$$

$R_E$ has full rank and is therefore invertible. Thus, $B_E$ is obtained by

$$B_E := P^\top \left( \left( R_E \right)^{-1} \right)^\top. \tag{2.49}$$

From the viewpoint of linear regression, $\boldsymbol{b}_0$ can be regarded as the constant terms and $\boldsymbol{b}_h$ ($1 \le h \le m-1$) is the coefficients for the $h$-th LDA dimension.

We can rewrite $P$ using the row vectors in $R_E$ and $B_E$. Writing the $i$-th ($1 \le i \le m$) row of $R_E$ as $\hat{\boldsymbol{r}}_i = \left( 1 \; r_{i1} \cdots r_{i(m-1)} \right)$,

$$R_E = \begin{bmatrix} \hat{\boldsymbol{r}}_1 \\ \vdots \\ \hat{\boldsymbol{r}}_m \end{bmatrix}. \tag{2.50}$$

Likewise, writing the $j$-th ($1 \le j \le n$) row of $B_E$ as $\hat{\boldsymbol{b}}_j = \left( b_{j0} \cdots b_{j(m-1)} \right)$,

$$B_E = \begin{bmatrix} \hat{\boldsymbol{b}}_1 \\ \vdots \\ \hat{\boldsymbol{b}}_n \end{bmatrix}. \tag{2.51}$$

Then, **Equation 2.46** can be written in another way:

$$\begin{aligned} P \quad &= R_E \left( B_E \right)^\top \\ &= \begin{bmatrix} \hat{\boldsymbol{r}}_1 \\ \vdots \\ \hat{\boldsymbol{r}}_m \end{bmatrix} \left[ \left( \hat{\boldsymbol{b}}_1 \right)^\top \cdots \left( \hat{\boldsymbol{b}}_n \right)^\top \right] = \begin{pmatrix} \hat{\boldsymbol{r}}_1 \left( \hat{\boldsymbol{b}}_1 \right)^\top & \cdots & \hat{\boldsymbol{r}}_1 \left( \hat{\boldsymbol{b}}_n \right)^\top \\ \vdots & \ddots & \vdots \\ \hat{\boldsymbol{r}}_m \left( \hat{\boldsymbol{b}}_1 \right)^\top & \cdots & \hat{\boldsymbol{r}}_m \left( \hat{\boldsymbol{b}}_n \right)^\top \end{pmatrix}. \end{aligned} \tag{2.52}$$

The interpretation of each vector is summarized in **Appendix 1—table 9**.

### 2.1.4 Relation between LDA Raman and $\Omega_{\mathrm{B}}$

Here, we discuss the spatial correspondence between the Raman distribution in LDA space and the normalized Raman-proteome coefficient proteome structure (**Figure 6B and C**).

## Connecting LDA Raman and Raman-omics transformation coefficients

Let us consider $\left( R_E \right)^\top R_E \left( B_E \right)^\top$ in two ways. In this first approach,

$$\left( R_E \right)^\top \left( R_E \left( B_E \right)^\top \right) = \left( R_E \right)^\top P \tag{2.53}$$

$$= \left[ \left( \hat{\boldsymbol{r}}_1 \right)^\top \cdots \left( \hat{\boldsymbol{r}}_m \right)^\top \right] \begin{pmatrix} p_{11} & \cdots & p_{1n} \\ \vdots & \ddots & \vdots \\ p_{m1} & \cdots & p_{mn} \end{pmatrix} \tag{2.54}$$

$$= \left[ \sum_{i=1}^m p_{i1} \left( \hat{\boldsymbol{r}}_i \right)^\top \quad \cdots \quad \sum_{i=1}^m p_{in} \left( \hat{\boldsymbol{r}}_i \right)^\top \right]. \tag{2.55}$$

In the second approach,

$$\left( \left( R_E \right)^\top R_E \right) \left( B_E \right)^\top$$
$$= \left( \Sigma_{R_E} \right)^2 \left( B_E \right)^\top \tag{2.56}$$

$$= \begin{pmatrix} m & & & \\ & (m-1)\lambda_{\text{LDA1}} & & \\ & & \ddots & \\ & & & (m-1)\lambda_{\text{LDA}(m-1)} \end{pmatrix} \left[ \left( \hat{\boldsymbol{b}}_1 \right)^\top \cdots \left( \hat{\boldsymbol{b}}_n \right)^\top \right]. \tag{2.57}$$

Comparing the two calculations yields

$$\left( R_E \right)^\top P = \left( \Sigma_{R_E} \right)^2 \left( B_E \right)^\top. \tag{2.58}$$

This is equivalent to

$$\sum_{i=1}^{m} p_{ij} \left( \hat{\boldsymbol{r}}_i \right)^\top = \begin{pmatrix} m & & & \\ & (m-1)\lambda_{\text{LDA1}} & & \\ & & \ddots & \\ & & & (m-1)\lambda_{\text{LDA}(m-1)} \end{pmatrix} \left( \hat{\boldsymbol{b}}_j \right)^\top \tag{2.59}$$

for any protein $j$.

## Normalization with constant terms

Next, we consider the normalization of the matrices in *Equation 2.58* with constant terms. We first define the normalized coefficient matrix as

$$B_E^{\text{norm}} := \begin{pmatrix} 1 & \frac{b_{11}}{b_{10}} & \cdots & \frac{b_{1(m-1)}}{b_{10}} \\ \vdots & \vdots & \ddots & \vdots \\ 1 & \frac{b_{n1}}{b_{n0}} & \cdots & \frac{b_{n(m-1)}}{b_{n0}} \end{pmatrix} = \text{diag} \left( \boldsymbol{b}_0 \right)^{-1} B_E. \tag{2.60}$$

This is normalization of the coefficients by the constant terms. Furthermore, we normalize $\Sigma_{R_E}$ as

$$\Sigma_{R_E}^{\text{norm}} := \frac{1}{\left( \Sigma_{R_E} \right)_{11}} \Sigma_{R_E} = \frac{1}{\sqrt{m}} \Sigma_{R_E} \tag{2.61}$$

$$= \begin{pmatrix} 1 & & & \\ & \sqrt{\frac{m-1}{m}} \lambda_{\text{LDA1}} & & \\ & & \ddots & \\ & & & \sqrt{\frac{m-1}{m}} \lambda_{\text{LDA}(m-1)} \end{pmatrix}. \tag{2.62}$$

Thus, the right-hand side of *Equation 2.58* is

$$\left( \Sigma_{R_E} \right)^2 \left( B_E \right)^\top = m \left( \Sigma_{R_E}^{\text{norm}} \right)^2 \left( B_E^{\text{norm}} \right)^\top \text{diag} \left( \boldsymbol{b}_0 \right). \tag{2.63}$$

Since the first row of $\left( R_E \right)^\top$ is $\left( \boldsymbol{1}_m \right)^\top$, one can rewrite the left-hand side of *Equation 2.58* as

$$(R_E)^\top P = \left[ \frac{\sum_{i=1}^m p_{i1} (\hat{r}_i)^\top}{\sum_{i=1}^m p_{i1}} \quad \cdots \quad \frac{\sum_{i=1}^m p_{in} (\hat{r}_i)^\top}{\sum_{i=1}^m p_{in}} \right] \begin{pmatrix} \sum_{i=1}^m p_{i1} & & \\ & \ddots & \\ & & \sum_{i=1}^m p_{in} \end{pmatrix} \tag{2.64}$$

$$= \left[ \frac{\sum_{i=1}^m p_{i1} (\hat{r}_i)^\top}{\sum_{i=1}^m p_{i1}} \quad \cdots \quad \frac{\sum_{i=1}^m p_{in} (\hat{r}_i)^\top}{\sum_{i=1}^m p_{in}} \right] \mathrm{diag}\left( (\mathbf{1}_m)^\top P \right). \tag{2.65}$$

Here,

$$(\mathbf{1}_m)^\top P = (\mathbf{1}_m)^\top R_E (B_E)^\top \tag{2.66}$$

$$= \begin{bmatrix} m & (\mathbf{0}_{m-1})^\top \end{bmatrix} (B_E)^\top \tag{2.67}$$

$$= m (\mathbf{b}_0)^\top. \tag{2.68}$$

Therefore,

$$\mathbf{b}_0 = \frac{1}{m} \begin{pmatrix} (\mathbf{1}_m)^\top \mathbf{p}_1 \\ \vdots \\ (\mathbf{1}_m)^\top \mathbf{p}_n \end{pmatrix} = \begin{pmatrix} \frac{1}{m} \sum_{i=1}^m p_{i1} \\ \vdots \\ \frac{1}{m} \sum_{i=1}^m p_{in} \end{pmatrix}. \tag{2.69}$$

This relation indicates that the constant term for each protein is its average abundance. Consequently,

$$(R_E)^\top P = m \left[ \frac{\sum_{i=1}^m p_{i1} (\hat{r}_i)^\top}{\sum_{i=1}^m p_{i1}} \quad \cdots \quad \frac{\sum_{i=1}^m p_{in} (\hat{r}_i)^\top}{\sum_{i=1}^m p_{in}} \right] \mathrm{diag}(\mathbf{b}_0). \tag{2.70}$$

Therefore, we obtain

$$\left[ \frac{\sum_{i=1}^m p_{i1} (\hat{r}_i)^\top}{\sum_{i=1}^m p_{i1}} \quad \cdots \quad \frac{\sum_{i=1}^m p_{in} (\hat{r}_i)^\top}{\sum_{i=1}^m p_{in}} \right] = \left( \Sigma_{R_E}^{\mathrm{norm}} \right)^2 \left( B_E^{\mathrm{norm}} \right)^\top. \tag{2.71}$$

Equivalently,

$$\frac{\sum_{i=1}^m p_{ij} (\hat{r}_i)^\top}{\sum_{i=1}^m p_{ij}} = \begin{pmatrix} 1 & & & \\ & \frac{m-1}{m} \lambda_{\mathrm{LDA1}} & & \\ & & \ddots & \\ & & & \frac{m-1}{m} \lambda_{\mathrm{LDA}(m-1)} \end{pmatrix} \begin{pmatrix} 1 \\ \frac{b_{j1}}{b_{j0}} \\ \vdots \\ \frac{b_{j(m-1)}}{b_{j0}} \end{pmatrix} \tag{2.72}$$

for any protein $j$. This means that the normalized coefficients of each protein are mainly determined by the weighted averages of the Raman vectors, where the weights are the abundances of the protein.

As we already saw in *Equation 7* or *Equation 8* in Materials and methods, this equation also shows that protein pairs whose abundance ratio remains constant over all the conditions have identical normalized coefficients.

## Special case – condition-specific protein

Consider an imaginary condition-specific protein $\gamma$ whose abundance is $c$ ($> 0$) under condition $\Gamma$ and zero under the other conditions, i.e.,

$$p_{i\gamma} = \begin{cases} 0 & (i \neq \Gamma) \\ c & (i = \Gamma) \end{cases}. \tag{2.73}$$

From *Equation 2.59*,

$$\begin{pmatrix} c & & & \\ & c & & \\ & & \ddots & \\ & & & c \end{pmatrix} \left(\hat{\boldsymbol{r}}_\Gamma\right)^\top = \begin{pmatrix} m & & & \\ & (m-1)\lambda_{\mathrm{LDA1}} & & \\ & & \ddots & \\ & & & (m-1)\lambda_{\mathrm{LDA}(m-1)} \end{pmatrix} \left(\hat{\boldsymbol{b}}_\gamma\right)^\top, \tag{2.74}$$

which indicates that the LDA Raman of condition $\Gamma$, $\left(\hat{\boldsymbol{r}}_\Gamma\right)^\top$ and Raman-proteome coefficients for $\Gamma$-specific protein $\gamma$, $\left(\hat{\boldsymbol{b}}_\gamma\right)^\top$ are in the same orthant. The normalized version is obtained by dividing both sides by the first row (or from *Equation 2.72*):

$$\left(\hat{\boldsymbol{r}}_\Gamma\right)^\top = \begin{pmatrix} 1 & & & \\ & \dfrac{m-1}{m}\lambda_{\mathrm{LDA1}} & & \\ & & \ddots & \\ & & & \dfrac{m-1}{m}\lambda_{\mathrm{LDA}(m-1)} \end{pmatrix} \begin{pmatrix} 1 \\ \dfrac{b_{\gamma 1}}{b_{\gamma 0}} \\ \vdots \\ \dfrac{b_{\gamma(m-1)}}{b_{\gamma 0}} \end{pmatrix} \tag{2.75}$$

$$= \left(\Sigma_{R_E}^{\mathrm{norm}}\right)^2 \frac{\left(\hat{\boldsymbol{b}}_\gamma\right)^\top}{b_{\gamma 0}}, \tag{2.76}$$

which shows that the LDA Raman of condition $\Gamma$, $\left(\hat{\boldsymbol{r}}_\Gamma\right)^\top$ and the normalized Raman-proteome coefficients of $\Gamma$-specific protein $\gamma$, $\left(\hat{\boldsymbol{b}}_\gamma\right)^\top / b_{\gamma 0}$ are in the same orthant.

## Application to main data

The LDA Raman distribution shown in *Figure 6B* corresponds to $\hat{r}_i$ (scatterplots between different columns of $R_E$). On the other hand, the normalized coefficient proteome structure in *Figure 6C* is the scatterplots between different columns of $B_E^{\mathrm{norm}}$. The above linear algebra explains the correspondence between the two. In addition, from *Equation 2.72*, one can understand that the homeostatic core distributes around the center of the structure because its member proteins are expressed in all the conditions.

*Equation 2.76* was obtained by considering an imaginary protein whose expression levels were zero under all conditions except for one condition. To confirm this relation with actual data, we picked an almost-condition-specific protein (PaaE, highly expressed in LB condition) and a non-condition-specific protein (AcrR), and confirmed that the former approximately satisfied *Equation 2.76*, while the relation did not hold for the latter (*Appendix 1—figure 10*).

## 2.1.5 Relation between $\Omega_{\mathrm{B}}$ and $\Omega_{\mathrm{LE}}$

Here, we discuss the spatial correspondence between the normalized Raman-proteome coefficient proteome structure and the csLE proteome structure (*Figure 6C and D*).

## csLE proteome structure

Consider an undirected graph where each node corresponds to one type of protein. As previously explained, let the graph be a complete graph, namely every pair of nodes is connected by an edge. Each edge is weighted with cosine similarity between the two types of protein connected by the edge. Cosine similarity of protein $i$ and protein $j$ is given by

$$\cos\theta_{\boldsymbol{p}_i\boldsymbol{p}_j} := \frac{\boldsymbol{p}_i \cdot \boldsymbol{p}_j}{\|\boldsymbol{p}_i\|_2 \|\boldsymbol{p}_j\|_2}, \tag{2.77}$$

where $\boldsymbol{p}_i \cdot \boldsymbol{p}_j \left( = (\boldsymbol{p}_i)^\top \boldsymbol{p}_j \right)$ is the inner product of $\boldsymbol{p}_i$ and $\boldsymbol{p}_j$, and $\|\boldsymbol{p}_i\|_2 \left( = \sqrt{(\boldsymbol{p}_i)^\top \boldsymbol{p}_i} \right)$ is the $L^2$-norm (Euclidean norm) of $\boldsymbol{p}_i$. Cosine similarity $\cos\theta_{\boldsymbol{p}_i\boldsymbol{p}_j}$ evaluates how similar the expression patterns are between protein $i$ and protein $j$. When the abundance ratio between protein $i$ and protein $j$ remains constant over all the $m$ conditions, $\cos\theta_{\boldsymbol{p}_i\boldsymbol{p}_j}$ takes the maximum value 1.

The adjacency matrix of this graph is given by

$$A := \left( \cos\theta_{\boldsymbol{p}_i\boldsymbol{p}_j} \right)_{1\leq i,j\leq n} = \left( \frac{\boldsymbol{p}_i \cdot \boldsymbol{p}_j}{\|\boldsymbol{p}_i\|_2 \|\boldsymbol{p}_j\|_2} \right)_{1\leq i,j\leq n} \tag{2.78}$$

and the degree matrix of this graph is

$$D = \mathrm{diag}\left( A\mathbf{1}_n \right). \tag{2.79}$$

For simplicity, diagonal element $(i,i)$ of $A$ is $\cos\theta_{\boldsymbol{p}_i\boldsymbol{p}_i} = 1$ for any protein $i$, i.e., each node has a loop. $A$ is $n \times n$ and real symmetric and $D$ is $n \times n$ and diagonal. Then, the Laplacian matrix is given by

$$L := D - A, \tag{2.80}$$

which is an $n \times n$ symmetric matrix, and the symmetric normalized Laplacian is given by

$$L_{\mathrm{sym}} = D^{-1/2} L D^{-1/2} = I - D^{-1/2} A D^{-1/2}, \tag{2.81}$$

which is an $n \times n$ symmetric matrix.

Here, we define $\hat{P}$ by normalizing the columns of $\mathrm{P} = \begin{bmatrix} \boldsymbol{p}_1 & \cdots & \boldsymbol{p}_n \end{bmatrix}$:

$$\hat{P} := \begin{bmatrix} \dfrac{\boldsymbol{p}_1}{\|\boldsymbol{p}_1\|_2} & \cdots & \dfrac{\boldsymbol{p}_n}{\|\boldsymbol{p}_n\|_2} \end{bmatrix} = P \,\mathrm{diag}\left( P^\top P \right)^{-1/2} \tag{2.82}$$

By using $\hat{P}$, $A$ is rewritten as

$$A = \left( \cos\theta_{\boldsymbol{p}_i\boldsymbol{p}_j} \right)_{1\leq i,j\leq n} = \left( \frac{\boldsymbol{p}_i \cdot \boldsymbol{p}_j}{\|\boldsymbol{p}_i\|_2 \|\boldsymbol{p}_j\|_2} \right)_{1\leq i,j\leq n} \tag{2.83}$$

$$= \left( P\,\mathrm{diag}\left( P^\top P \right)^{-1/2} \right)^\top P\,\mathrm{diag}\left( P^\top P \right)^{-1/2} \tag{2.84}$$

$$= \mathrm{diag}\left( P^\top P \right)^{-1/2} P^\top P\,\mathrm{diag}\left( P^\top P \right)^{-1/2} \tag{2.85}$$

$$= \hat{P}^\top \hat{P}. \tag{2.86}$$

Consider an eigenproblem

$$L_{\mathrm{sym}} V_{\mathrm{sym}} = V_{\mathrm{sym}} \Lambda_{\mathrm{LE}}, \tag{2.87}$$

where $\Lambda_{\mathrm{LE}}$ is an $n \times n$ diagonal matrix in which the eigenvalues of $L_{\mathrm{sym}}$ are arranged in increasing order from the upper left, and columns of $V_{\mathrm{sym}}$ are the normalized eigenvectors of $L_{\mathrm{sym}}$ corresponding to the eigenvalues. Denote

$$\Lambda_{\mathrm{LE}} = \begin{pmatrix} \lambda_{\mathrm{LE}0} & & \\ & \ddots & \\ & & \lambda_{\mathrm{LE}(n-1)} \end{pmatrix} \tag{2.88}$$

Here, $L_{\mathrm{sym}}$ has the following four characteristics:

- $L_{\mathrm{sym}}$ is positive semi-definite. See, for example, **von Luxburg, 2007** for the proof.

- In an undirected graph with non-negative weights, the number of separated graph components equals the multiplicity of the eigenvalue zero of $L_{\mathrm{sym}}$. See, for example, **von Luxburg, 2007** for the proof. Since our proteome graph is connected, our $L_{\mathrm{sym}}$ has the single eigenvalue zero.

- From **Equation 2.86**, $\mathrm{rank}(A) = \mathrm{rank}(\hat{P})$. Here, it is obvious that $\mathrm{diag}\left(P^\top P\right)^{-1/2}$ has full rank, hence, $\mathrm{rank}(\hat{P}) = \mathrm{rank}(P) = m$. Therefore, $\mathrm{rank}(A) = m$. Obviously, $D$ has full rank by definition and thus, $\mathrm{rank}\left(D^{-1/2}AD^{-1/2}\right) = \mathrm{rank}(A) = m$. Therefore, $D^{-1/2}AD^{-1/2}$ has $n - m$ singular values of zero. Since $D^{-1/2}AD^{-1/2}$ is symmetric, its singular values and eigenvalues are the same. Thus, $D^{-1/2}AD^{-1/2}$ has $n - m$ singular values of zero. Therefore, $L_{\mathrm{sym}}\left(= I - D^{-1/2}AD^{-1/2}\right)$ has $n - m$ singular values of $1(= 1 - 0)$.

- For any $n$-dimensional vector $x$, $x^\top D^{-1/2}AD^{-1/2}x = \left(\hat{P}D^{-1/2}x\right)^\top \hat{P}D^{-1/2}x = \left\|\hat{P}D^{-1/2}x\right\|_2^2 \geq 0$. Therefore, $D^{-1/2}AD^{-1/2}$ is positive semi-definite, and all of the eigenvalues of $L_{\mathrm{sym}}\left(= I - D^{-1/2}AD^{-1/2}\right)$ are less than or equal to one.

By these four points, we see that the eigenvalues of $L_{\mathrm{sym}}\left(= I - D^{-1/2}AD^{-1/2}\right)$ satisfy

$$0 = \lambda_{\mathrm{LE0}} < \lambda_{\mathrm{LE1}} \leq \cdots \leq \lambda_{\mathrm{LE}(m-1)} < \lambda_{\mathrm{LE}m} = \lambda_{\mathrm{LE}(m+1)} = \cdots = \lambda_{\mathrm{LE}(n-1)} = 1. \tag{2.89}$$

Now we define an $m \times m$ matrix $\tilde{\Lambda}_{\mathrm{LE}}$ as

$$\tilde{\Lambda}_{\mathrm{LE}} = \begin{pmatrix} 0 & & & \\ & \lambda_{\mathrm{LE1}} & & \\ & & \ddots & \\ & & & \lambda_{\mathrm{LE}(m-1)} \end{pmatrix}. \tag{2.90}$$

Then, we can write

$$\Lambda_{\mathrm{LE}} = \begin{pmatrix} \tilde{\Lambda}_{\mathrm{LE}} & \\ & I \end{pmatrix}. \tag{2.91}$$

Let $\tilde{V}_{\mathrm{sym}}$ be the first $m$ columns of $V_{\mathrm{sym}}$:

$$\tilde{V}_{\mathrm{sym}} = \begin{bmatrix} \hat{v}_{\mathrm{sym},1} \\ \vdots \\ \hat{v}_{\mathrm{sym},n} \end{bmatrix} = \begin{bmatrix} v_{\mathrm{sym},10} & \cdots & v_{\mathrm{sym},1(m-1)} \\ \vdots & \ddots & \vdots \\ v_{\mathrm{sym},n0} & \cdots & v_{\mathrm{sym},n(m-1)} \end{bmatrix}. \tag{2.92}$$

The truncated version of the eigenproblem is

$$L_{\mathrm{sym}}\tilde{V}_{\mathrm{sym}} = \tilde{V}_{\mathrm{sym}}\tilde{\Lambda}_{\mathrm{LE}}. \tag{2.93}$$

$\hat{v}_{\mathrm{sym},i} := \left(v_{\mathrm{sym},i0} \quad \cdots \quad v_{\mathrm{sym},i(m-1)}\right)$ is the $i$-th row of $\tilde{V}_{\mathrm{sym}}$ and provides a new $m$-dimensional representation of the $i$-th protein. This representation of the proteome reflects distance between each protein pair in terms of cosine similarity.

Here we clarify the correspondence of the eigenproblems defined by the normalized Laplacian matrices $L_{\mathrm{sym}}$ and $L_{\mathrm{rw}}$. Let

$$\tilde{V}_{\mathrm{rw}} = D^{-1/2}\tilde{V}_{\mathrm{sym}}. \tag{2.94}$$

Then, this eigenproblem can also be regarded as the following generalized eigenproblem,

$$L\tilde{V}_{\mathrm{rw}} = D\tilde{V}_{\mathrm{rw}}\tilde{\Lambda}_{\mathrm{LE}}. \tag{2.95}$$

Remembering that $L_{\mathrm{rw}} = D^{-1}L$, we can further transform it into an eigneproblem

$$L_{\mathrm{rw}}\tilde{V}_{\mathrm{rw}} = \tilde{V}_{\mathrm{rw}}\tilde{\Lambda}_{\mathrm{LE}}. \tag{2.96}$$

This is the form of eigenproblem that we explained previously in 'Global proteome structures based on stoichiometric balance' in Materials and methods. In this section, we discuss it later.

The eigenproblem (*Equation 2.87*) can be transformed to

$$D^{-1/2}AD^{-1/2}V_{\mathrm{sym}} = V_{\mathrm{sym}}M, \tag{2.97}$$

where

$$M := I - \Lambda_{\mathrm{LE}} = \begin{pmatrix} I - \tilde{\Lambda}_{\mathrm{LE}} & \\ & O \end{pmatrix}. \tag{2.98}$$

This means that the columns of $V_{\mathrm{sym}}$ are also the (normalized) eigenvectors of $D^{-1/2}AD^{-1/2}$, and $M$ is the corresponding eigenvalue matrix.

Defining an $m \times m$ matrix $\tilde{M}$ as

$$\tilde{M} = I - \tilde{\Lambda}_{\mathrm{LE}} = \begin{pmatrix} 1 & & & & \\ & 1 - \lambda_{\mathrm{LE}1} & & & \\ & & & \ddots & \\ & & & & 1 - \lambda_{\mathrm{LE}(m-1)} \end{pmatrix}, \tag{2.99}$$

we can write

$$M = \begin{pmatrix} \tilde{M} & \\ & O \end{pmatrix}. \tag{2.100}$$

Note that

$$1 > 1 - \lambda_{\mathrm{LE}1} \geq \cdots \geq 1 - \lambda_{\mathrm{LE}(m-1)} > 0. \tag{2.101}$$

*Equation 2.97* is further transformed into

$$A = D^{1/2}V_{\mathrm{sym}}M\left(V_{\mathrm{sym}}\right)^{\top}D^{1/2}. \tag{2.102}$$

Comparing *Equation 2.102* and *Equation 2.85* leads to

$$\mathrm{diag}\left(P^{\top}P\right)^{-1/2}P^{\top}P\,\mathrm{diag}\left(P^{\top}P\right)^{-1/2} = D^{1/2}V_{\mathrm{sym}}M\left(V_{\mathrm{sym}}\right)^{\top}D^{1/2}. \tag{2.103}$$

## Connecting Raman-proteome transformation coefficients and csLE proteome

We consider $P^{\top}P$ in two ways. First, from *Equation 2.103*,

$$
\begin{aligned}
P^\top P &= \mathrm{diag}\left(P^\top P\right)^{1/2} D^{1/2} V_{\mathrm{sym}} M \left(V_{\mathrm{sym}}\right)^\top D^{1/2} \mathrm{diag}\left(P^\top P\right)^{1/2} \\
&= \mathrm{diag}\left(P^\top P\right)^{1/2} D^{1/2} V_{\mathrm{sym}} M^{1/2} \left(\mathrm{diag}\left(P^\top P\right)^{1/2} D^{1/2} V_{\mathrm{sym}} M^{1/2}\right)^\top \\
&= \mathrm{diag}\left(P^\top P\right)^{1/2} D^{1/2} \left(\tilde{V}_{\mathrm{sym}} \tilde{M}^{1/2} | O\right) \left(\mathrm{diag}\left(P^\top P\right)^{1/2} D^{1/2} \left(\tilde{V}_{\mathrm{sym}} \tilde{M}^{1/2} | O\right)\right)^\top \\
&= \left(\mathrm{diag}\left(P^\top P\right)^{1/2} D^{1/2} \tilde{V}_{\mathrm{sym}} \tilde{M}^{1/2} | O\right) \left(\mathrm{diag}\left(P^\top P\right)^{1/2} D^{1/2} \tilde{V}_{\mathrm{sym}} \tilde{M}^{1/2} | O\right)^\top \\
&= \left(\mathrm{diag}\left(P^\top P\right)^{1/2} D^{1/2} \tilde{V}_{\mathrm{sym}} \tilde{M}^{1/2} | O\right) \begin{pmatrix} \left(\mathrm{diag}\left(P^\top P\right)^{1/2} D^{1/2} \tilde{V}_{\mathrm{sym}} \tilde{M}^{1/2}\right)^\top \\ O \end{pmatrix} \\
&= \mathrm{diag}\left(P^\top P\right)^{1/2} D^{1/2} \tilde{V}_{\mathrm{sym}} \tilde{M}^{1/2} \left(\mathrm{diag}\left(P^\top P\right)^{1/2} D^{1/2} \tilde{V}_{\mathrm{sym}} \tilde{M}^{1/2}\right)^\top.
\end{aligned}
\tag{2.104}
$$

Since

$$
D^{-1/2} A D^{-1/2} = D^{-1/2} \hat{P}^\top \hat{P} D^{-1/2} \quad (\because (2.86))
\tag{2.105}
$$

$$
= \left(\hat{P} D^{-1/2}\right)^\top \hat{P} D^{-1/2},
\tag{2.106}
$$

the diagonal elements of $\tilde{M}$, i.e., the positive eigenvalues of $D^{-1/2} A D^{-1/2}$ are the square of the singular values of $\hat{P} D^{-1/2}$. Compact SVD of $\hat{P} D^{-1/2}$ is expressed as

$$
\hat{P} D^{-1/2} = U_{\mathrm{LE}} \Sigma_{\mathrm{LE}} \left(V_{\mathrm{LE}}\right)^\top,
\tag{2.107}
$$

where $\Sigma_{\mathrm{LE}}$ is an $m \times m$ diagonal matrix whose diagonal elements are the singular values in decreasing order from the upper left, and $\left(U_{\mathrm{LE}}\right)^\top U_{\mathrm{LE}} = \left(V_{\mathrm{LE}}\right)^\top V_{\mathrm{LE}} = I$. We then obtain

$$
\Sigma_{\mathrm{LE}} = \tilde{M}^{1/2} = \begin{pmatrix} 1 & & & \\ & \sqrt{1 - \lambda_{\mathrm{LE}1}} & & \\ & & \ddots & \\ & & & \sqrt{1 - \lambda_{\mathrm{LE}(m-1)}} \end{pmatrix}.
\tag{2.108}
$$

Thus,

$$
P^\top P = \mathrm{diag}\left(P^\top P\right)^{1/2} D^{1/2} \tilde{V}_{\mathrm{sym}} \Sigma_{\mathrm{LE}} \left(\mathrm{diag}\left(P^\top P\right)^{1/2} D^{1/2} \tilde{V}_{\mathrm{sym}} \Sigma_{\mathrm{LE}}\right)^\top.
\tag{2.109}
$$

On the other hand, from *Equation 2.44* and *Equation 2.46*,

$$
P^\top P = \left(R_E \left(B_E\right)^\top\right)^\top R_E \left(B_E\right)^\top = B_E \left(R_E\right)^\top R_E \left(B_E\right)^\top
\tag{2.110}
$$

$$
= B_E \left(\Sigma_{R_E}\right)^2 B_E^\top
\tag{2.111}
$$

$$
= B_E \Sigma_{R_E} \left(B_E \Sigma_{R_E}\right)^\top.
\tag{2.112}
$$

Therefore, comparing *Equation 2.109* and *Equation 2.112* yields

$$
B_E \Sigma_{R_E} = \mathrm{diag}\left(P^\top P\right)^{1/2} D^{1/2} \tilde{V}_{\mathrm{sym}} \Sigma_{\mathrm{LE}} \Theta,
\tag{2.113}
$$

where $\Theta$ is an $m \times m$ orthogonal matrix. We define the estimate of $B_E$ as

$$
B_E^{\mathrm{est}} := \mathrm{diag}\left(P^\top P\right)^{1/2} D^{1/2} \tilde{V}_{\mathrm{sym}}.
\tag{2.114}
$$

By this notation, *Equation 2.113* can be written as

$$B_E \Sigma_{R_E} = B_E^{\text{est}} \Sigma_{\text{LE}} \Theta. \tag{2.115}$$

Here, the left-hand side represents Raman-proteome linear transformation, whereas the right-hand side except for $\Theta$ is derived only from proteome data. Note that in order to derive *Equation 2.113*, LDA does not need to be applied to Raman data because $P^\top P$ can be written in the form of $B_E \left(\Sigma_{R_E}\right)^2 \left(B_E\right)^\top$ even if LDA is not applied to Raman data.

## Normalization with constant terms

Now we consider normalizing both sides of *Equation 2.115* by the first columns.

With *Equation 2.60* and *Equation 2.61*, the left-hand side of *Equation 2.115* can be rewritten as

$$B_E \Sigma_{R_E} = \sqrt{m} \operatorname{diag}\left(\boldsymbol{b}_0\right) B_E^{\text{norm}} \Sigma_{R_E}^{\text{norm}}. \tag{2.116}$$

Likewise, for the right-hand side, the first column of $B_E^{\text{est}}$ (the estimated constant term) is

$$\boldsymbol{b}_0^{\text{est}} := B_E^{\text{est}}[:, 1] = B_E^{\text{est}} \begin{pmatrix} 1 \\ 0 \\ \vdots \\ 0 \end{pmatrix} = \operatorname{diag}\left(P^\top P\right)^{1/2} D^{1/2} \tilde{V}_{\text{sym}} \begin{pmatrix} 1 \\ 0 \\ \vdots \\ 0 \end{pmatrix}. \tag{2.117}$$

The first column of $\tilde{V}_{\text{sym}}$ is the normalized eigenvector corresponding to the eigenvalue zero of $L_{\text{sym}}$.

By the definition of $L$, $L\mathbf{1}_n = D\mathbf{1}_n - A\mathbf{1}_n = \operatorname{diag}(D) - \operatorname{diag}(D) = \mathbf{0}_n$. Hence, in general, $L$ has an eigenvalue zero and a corresponding eigenvector $\mathbf{1}_n$. Therefore, by the definition of $L_{\text{rw}}$, $L_{\text{rw}}\mathbf{1}_n = D^{-1}L\mathbf{1}_n = \mathbf{0}_n$; $L_{\text{rw}}$ also has an eigenvalue zero and a corresponding eigenvector $\mathbf{1}_n$. The eigenproblems $L_{\text{rw}}\tilde{V}_{\text{rw}} = \tilde{V}_{\text{rw}}\tilde{\Lambda}_{\text{LE}}$ and $L_{\text{sym}}\tilde{V}_{\text{sym}} = \tilde{V}_{\text{sym}}\tilde{\Lambda}_{\text{LE}}$ are equivalent because one can obtain the eigenproblem of $L_{\text{sym}}$ by left multiplying both sides of the eigenproblem of $L_{\text{rw}}$ by $D^{1/2}$ and that of $L_{\text{rw}}$ by left multiplying both sides of the eigenproblem of $L_{\text{sym}}$ by $D^{-1/2}$. Eigenvalues of $L_{\text{rw}}$ and $L_{\text{sym}}$ are identical, and $\tilde{V}_{\text{rw}} = D^{-1/2}\tilde{V}_{\text{sym}}$ holds for their eigenvectors. Therefore, $L_{\text{sym}}$ has an eigenvalue zero and a corresponding eigenvector $D^{1/2}\mathbf{1}_n$. The multiplicity of the eigenvalue zero of our $L_{\text{sym}}$ is one, and its corresponding eigenvector is limited to $D^{1/2}\mathbf{1}_n$.

By writing $D$ as

$$D = \begin{pmatrix} d_1 & & \\ & \ddots & \\ & & d_n \end{pmatrix}, \tag{2.118}$$

$$\tilde{V}_{\text{sym}}[:, 1] = \tilde{V}_{\text{sym}} \begin{pmatrix} 1 \\ 0 \\ \vdots \\ 0 \end{pmatrix} = \left(\sum_{i=1}^{n} d_i\right)^{-1/2} D^{1/2}\mathbf{1}_n. \tag{2.119}$$

Thus, *Equation 2.117* can be further transformed as

$$
\begin{aligned}
\boldsymbol{b}_0^{\text{est}} = B_E^{\text{est}}[:, 1] &= \left(\textstyle\sum_{i=1}^{n} d_i\right)^{-1/2} \operatorname{diag}\left(P^\top P\right)^{1/2} D^{1/2} D^{1/2}\mathbf{1}_n \\
&= \left(\textstyle\sum_{i=1}^{n} d_i\right)^{-1/2} \operatorname{diag}\left(P^\top P\right)^{1/2} D\mathbf{1}_n
\end{aligned}
\tag{2.120}
$$

$$= \frac{1}{\left(\sum_{i=1}^{n} d_i\right)^{1/2}} \begin{pmatrix} d_1 \sqrt{(\boldsymbol{p}_1)^\top \boldsymbol{p}_1} \\ \vdots \\ d_n \sqrt{(\boldsymbol{p}_n)^\top \boldsymbol{p}_n} \end{pmatrix}. \tag{2.121}$$

Remembering that both $\mathrm{diag}\left(P^\top P\right)$ and $D$ are diagonal, we obtain

$$\mathrm{diag}\left(\boldsymbol{b}_0^{\mathrm{est}}\right) = \mathrm{diag}\left(B_E^{\mathrm{est}}\left[:,1\right]\right) = \left(\sum_{i=1}^{n} d_i\right)^{-1/2} \mathrm{diag}\left(P^\top P\right)^{1/2} D. \tag{2.122}$$

Therefore, from *Equation 2.114* and *Equation 2.122*, the 'estimated coefficients' normalized with the 'estimated constants' are

$$B_E^{\mathrm{est,norm}} := \mathrm{diag}\left(\boldsymbol{b}_0^{\mathrm{est}}\right)^{-1} B_E^{\mathrm{est}} \tag{2.123}$$

$$= \mathrm{diag}\left(\boldsymbol{b}_0^{\mathrm{est}}\right)^{-1} \mathrm{diag}\left(P^\top P\right)^{1/2} D^{1/2} \tilde{V}_{\mathrm{sym}} \tag{2.124}$$

$$= \left(\sum_{i=1}^{n} d_i\right)^{1/2} D^{-1} \mathrm{diag}\left(P^\top P\right)^{-1/2} \mathrm{diag}\left(P^\top P\right)^{1/2} D^{1/2} \tilde{V}_{\mathrm{sym}} \tag{2.125}$$

$$= \left(\sum_{i=1}^{n} d_i\right)^{1/2} D^{-1/2} \tilde{V}_{\mathrm{sym}} \tag{2.126}$$

$$= \left(\sum_{i=1}^{n} d_i\right)^{1/2} \tilde{V}_{\mathrm{rw}}. \tag{2.127}$$

We remark that the eigenproblem of $L_{\mathrm{rw}}$, i.e.,

$$L_{\mathrm{rw}} \tilde{V}_{\mathrm{rw}} = \tilde{V}_{\mathrm{rw}} \tilde{\Lambda}_{\mathrm{LE}} \tag{2.128}$$

is equivalent to solving a minimization problem

$$\arg\min \sum_{i,j} \left\|\hat{\boldsymbol{v}}_{\mathrm{rw},i} - \hat{\boldsymbol{v}}_{\mathrm{rw},j}\right\|_2^2 \cos\theta_{\boldsymbol{p}_i \boldsymbol{p}_j} \quad \text{subject to} \quad \left(\tilde{V}_{\mathrm{rw}}\right)^\top D \tilde{V}_{\mathrm{rw}} = I. \tag{2.129}$$

Thus, the closer $\boldsymbol{p}_i$ and $\boldsymbol{p}_j$ are in terms of cosine similarity, the closer $\hat{\boldsymbol{v}}_{\mathrm{rw},i}$ and $\hat{\boldsymbol{v}}_{\mathrm{rw},j}$ (the $i$-th and $j$-th rows of $\tilde{V}_{\mathrm{rw}}$, respectively).

The relation between the minimization problem and the eigenproblem is the following. The objective function of the minimization problem is

$$\sum_{i,j} \left\|\hat{\boldsymbol{v}}_{\mathrm{rw},i} - \hat{\boldsymbol{v}}_{\mathrm{rw},j}\right\|_2^2 \cos\theta_{\boldsymbol{p}_i \boldsymbol{p}_j} = 2 \sum_{i} d_i \left\|\hat{\boldsymbol{v}}_{\mathrm{rw},i}\right\|_2^2 - 2 \sum_{i,j} A_{ij} \hat{\boldsymbol{v}}_{\mathrm{rw},i} \cdot \hat{\boldsymbol{v}}_{\mathrm{rw},j}. \tag{2.130}$$

Here,

$$\mathrm{tr}\left(\left(\tilde{V}_{\mathrm{rw}}\right)^\top D \tilde{V}_{\mathrm{rw}}\right) = \sum_{i} d_i \left\|\hat{\boldsymbol{v}}_{\mathrm{rw},i}\right\|_2^2, \tag{2.131}$$

$$\mathrm{tr}\left(\left(\tilde{V}_{\mathrm{rw}}\right)^\top A \tilde{V}_{\mathrm{rw}}\right) = \sum_{i,j} A_{ij} \hat{\boldsymbol{v}}_{\mathrm{rw},i} \cdot \hat{\boldsymbol{v}}_{\mathrm{rw},j}. \tag{2.132}$$

Therefore, *Equation 2.130* can be transformed into

$$\sum_{i,j} \left\|\hat{\boldsymbol{v}}_{\mathrm{rw},i} - \hat{\boldsymbol{v}}_{\mathrm{rw},j}\right\|_2^2 \cos\theta_{\boldsymbol{p}_i \boldsymbol{p}_j} = 2\,\mathrm{tr}\left(\left(\tilde{V}_{\mathrm{rw}}\right)^\top D \tilde{V}_{\mathrm{rw}}\right) - 2\,\mathrm{tr}\left(\left(\tilde{V}_{\mathrm{rw}}\right)^\top A \tilde{V}_{\mathrm{rw}}\right) \tag{2.133}$$

$$= 2 \operatorname{tr} \left( \left( \tilde{V}_{\mathrm{rw}} \right)^{\top} L \tilde{V}_{\mathrm{rw}} \right). \tag{2.134}$$

Thus, the minimization problem is

$$\arg\min \quad \operatorname{tr} \left( \left( \tilde{V}_{\mathrm{rw}} \right)^{\top} L \tilde{V}_{\mathrm{rw}} \right) \quad \text{subject to} \quad \left( \tilde{V}_{\mathrm{rw}} \right)^{\top} D \tilde{V}_{\mathrm{rw}} = I. \tag{2.135}$$

This can be transformed into the generalized eigenproblem $L\tilde{V}_{\mathrm{rw}} = D\tilde{V}_{\mathrm{rw}}\tilde{\Lambda}_{\mathrm{LE}}$ by the method of Lagrange multipliers (*Ghojogh et al., 2019*).

Remember that this property of $\hat{v}_{\mathrm{rw},i}$ and $\hat{v}_{\mathrm{rw},j}$ is analogous to that of $\hat{\boldsymbol{b}}_i^{\mathrm{norm}}$ and $\hat{\boldsymbol{b}}_j^{\mathrm{norm}}$ (the $i$-th and $j$-th rows of $B_E^{\mathrm{norm}}$, respectively) as explained in Section 2.1.4 in Appendix and 'Raman-proteome correspondence matrix as a low-dimensional representation of proteome changes' in Materials and methods.

By considering the first (upper-left) element of $\Sigma_{\mathrm{LE}}$ is one, the right side of *Equation 2.113* can be transformed into

$$B_E^{\mathrm{est}} \Sigma_{\mathrm{LE}} \Theta = \operatorname{diag} \left( \boldsymbol{b}_0^{\mathrm{est}} \right) B_E^{\mathrm{est,norm}} \Sigma_{\mathrm{LE}} \Theta \tag{2.136}$$

$$= \left[ \left( \sum_{i=1}^{n} d_i \right)^{-1/2} \operatorname{diag} \left( P^{\top} P \right)^{1/2} D \right] \left[ \left( \sum_{i=1}^{n} d_i \right)^{1/2} \tilde{V}_{\mathrm{rw}} \right] \Sigma_{\mathrm{LE}} \Theta. \tag{2.137}$$

Therefore, from *Equations 2.115, 2.116, 2.136, 2.137*,

$$\sqrt{m} \operatorname{diag} \left( \boldsymbol{b}_0 \right) B_E^{\mathrm{norm}} \Sigma_{R_E}^{\mathrm{norm}} = \operatorname{diag} \left( \boldsymbol{b}_0^{\mathrm{est}} \right) B_E^{\mathrm{est,norm}} \Sigma_{\mathrm{LE}} \Theta \tag{2.138}$$

$$= \left[ \left( \sum_{i=1}^{n} d_i \right)^{-1/2} \operatorname{diag} \left( P^{\top} P \right)^{1/2} D \right] \left[ \left( \sum_{i=1}^{n} d_i \right)^{1/2} \tilde{V}_{\mathrm{rw}} \right] \Sigma_{\mathrm{LE}} \Theta. \tag{2.139}$$

This is the equation that links the normalized Raman-proteome coefficient proteome structure and the csLE proteome structure.

## Mathematical interpretation of the obtained equation

From *Equation 2.139*, if the distributions of $B_E^{\mathrm{norm}}$ and $\tilde{V}_{\mathrm{rw}}$ are similar, the diagonal matrix $\Theta$ must be similar to the identity matrix because large off-diagonal elements of $\Theta$ makes lower dimensions 'mix' much with the higher dimensions. In addition, the directions of diagonal matrices $\operatorname{diag} \left( \boldsymbol{b}_0 \right)$ and $\operatorname{diag} \left( P^{\top} P \right)^{1/2} D$ must also be close to each other even if $\Theta$ is close to the identity matrix. Note that the first column of $\Theta$ also reflects the relation between $\boldsymbol{b}_0$ and $\boldsymbol{b}_0^{\mathrm{est}} \left( \propto \operatorname{diag} \left( P^{\top} P \right)^{1/2} D \mathbf{1}_n \right)$.

The obtained relation between Raman-proteome normalized coefficient structure and csLE structure is summarized in *Appendix 1—table 10*.

Note that the relation between $\Sigma_{R_E}^{\mathrm{norm}}$ and $\Sigma_{\mathrm{LE}}$ can change depending on normalization of $V_{\mathrm{LDA}}$. However, the difference is not important for the spatial correspondence between the two structures because they only affect scaling of the axes. Rather, $\Lambda'_{\mathrm{LDA}}$ and $\tilde{M} = \left( \Sigma_{\mathrm{LE}} \right)^2$, which determine the order of columns of $V_{\mathrm{LDA}}$ and $\tilde{V}_{\mathrm{rw}}$, are important.

## Application to main data

The normalized coefficient proteome structure in *Figure 6A and C* is the scatterplots between different columns of $B_E^{\mathrm{norm}}$. The cosine similarity proteome structures in *Figure 5L* and *Figure 6D* are the scatterplots between different columns of $\tilde{V}_{\mathrm{rw}}$.

In our data analysis, we calculated $\tilde{V}_{\mathrm{rw}}$ as $\tilde{V}_{\mathrm{rw}} = D^{-1/2} \tilde{V}_{\mathrm{sym}}$, where each column of $\tilde{V}_{\mathrm{sym}}$ was normalized.

On the basis of the results of the mathematical analysis, we compared $B_E^{\mathrm{norm}}$ and $B_E^{\mathrm{est,norm}} = \left( \sum_{i=1}^{n} d_i \right)^{1/2} \tilde{V}_{\mathrm{rw}}$ in *Appendix 1—figure 9G*.

The similarity between the projections of the two distinct omics structures onto low-dimensional subspaces suggests that $\Theta$ is close to the identity matrix (*Equation 2.138* and *Appendix 1—table 10*). *Appendix 1—figure 9A, C–E* shows that the actual $\Theta$ is indeed significantly close to the identity matrix. This suggests that the major changes in cellular Raman spectra detectable by LDA reflect

the major changes in the proteome characterized by LE based on stoichiometry balance (cosine similarity).

The structural similarity also suggests that directions of $\mathrm{diag}\,(\boldsymbol{b}_0)$ and $\mathrm{diag}\,(P^\top P)^{1/2}\,D$ are also similar (*Equation 2.138* and *Appendix 1—table 10*).

In fact, *Appendix 1—figure 9F* confirmed good agreement between $m^{1/2}\mathrm{diag}\,(\boldsymbol{b}_0)$ and $\left(\sum_{i=1}^n d_i\right)^{-1/2}\mathrm{diag}\left(P^\top P\right)^{1/2}D = \mathrm{diag}\,(\boldsymbol{b}_0^{\mathrm{est}})$. Since these two quantities are calculated only from proteome data, this agreement is a characteristic of proteome data. See the next section (Section 2.2 in Appendix) for further analyses and discussion on this point.

## 2.2 Quantitative constraint on omics profiles

### 2.2.1 From agreement between $\sqrt{m}\,\boldsymbol{b}_0$ and $\boldsymbol{b}_0^{\mathrm{est}}$ to proportionality between $L^1$ norm/$L^2$ norm ratio and degree

We observed above that constant terms $\boldsymbol{b}_0$ and the estimated constant terms $\boldsymbol{b}_0^{\mathrm{est}}$ were strongly correlated (*Appendix 1—figure 9F*). It is of note that both $\boldsymbol{b}_0$ and $\boldsymbol{b}_0^{\mathrm{est}}$ can be calculated only from omics data. Specifically, from *Equations 2.121 and 2.69*,

$$\boldsymbol{b}_0 \propto \begin{pmatrix} (\mathbf{1}_m)^\top \boldsymbol{p}_1 \\ \vdots \\ (\mathbf{1}_m)^\top \boldsymbol{p}_n \end{pmatrix} = \begin{pmatrix} \sum_{i=1}^m p_{i1} \\ \vdots \\ \sum_{i=1}^m p_{in} \end{pmatrix} = \begin{pmatrix} \sum_{i=1}^m |p_{i1}| \\ \vdots \\ \sum_{i=1}^m |p_{in}| \end{pmatrix} = \begin{pmatrix} \|\boldsymbol{p}_1\|_1 \\ \vdots \\ \|\boldsymbol{p}_n\|_1 \end{pmatrix}, \tag{2.140}$$

$$\boldsymbol{b}_0^{\mathrm{est}} \propto \begin{pmatrix} d_1 \sqrt{(\boldsymbol{p}_1)^\top \boldsymbol{p}_1} \\ \vdots \\ d_n \sqrt{(\boldsymbol{p}_n)^\top \boldsymbol{p}_n} \end{pmatrix} = \begin{pmatrix} d_1 \sqrt{\sum_{i=1}^m p_{i1}^2} \\ \vdots \\ d_n \sqrt{\sum_{i=1}^m p_{in}^2} \end{pmatrix} = \begin{pmatrix} d_1 \|\boldsymbol{p}_1\|_2 \\ \vdots \\ d_n \|\boldsymbol{p}_n\|_2 \end{pmatrix}. \tag{2.141}$$

Here, $\left\|\boldsymbol{p}_j\right\|_1$ and $\left\|\boldsymbol{p}_j\right\|_2$ are the $L^1$ and $L^2$ norms of $\boldsymbol{p}_j$ and reflect only the expression property of protein $j$. On the other hand, the degree $d_j$ is a measure for the relationships of protein $j$ with the other proteins because $d_j$ is the sum of cosine similarities, $d_j = \sum_{i=1}^n \cos\theta_{\boldsymbol{p}_i\boldsymbol{p}_j}$.

The observed relation

$$\sqrt{m}\,\boldsymbol{b}_0 \approx \boldsymbol{b}_0^{\mathrm{est}} \tag{2.142}$$

(*Appendix 1—figure 9F*) indicates that a proportionality relation

$$\frac{\|\boldsymbol{p}_j\|_1}{\|\boldsymbol{p}_j\|_2} \propto \sum_{i=1}^n \cos\theta_{\boldsymbol{p}_i\boldsymbol{p}_j} \tag{2.143}$$

must hold approximately.

As mentioned in the main text, we refer to the ratio of $L^1$ norm to $L^2$ norm in the left-hand side of *Equation 2.143* as expression generality score ($g_j$) because it can be interpreted as a measure of constancy and generality of the expression levels of the protein (see 'Interpretation of $L^1$ norm/$L^2$ norm ratio of an expression vector as a quantitative measure of expression generality' in Materials and methods, *Appendix 1—figure 8A and B*).

When a protein is perfectly condition-specific and expressed only in a particular condition, its $L^1$ norm equals its $L^2$ norm, and the ratio takes the minimum value one. When a protein is expressed equally across all the conditions, its $L^1$ norm is greater than its $L^2$ norm and the ratio takes the maximum value, the square root of the number of conditions (*Appendix 1—figure 8A and B*). On the other hand, the right-hand side of *Equation 2.143*, which we refer to as stoichiometry conservation centrality ($d_j$) in the main text, measures to what extent protein $j$ conserves its stoichiometry with the other proteins. Therefore, the proportionality relation (*Equation 2.143*) suggests a global quantitative constraint between condition specificity of expression patterns and stoichiometry conservation strength. Positions of SCGs and density of genes in the csLE structure (*Figure 5A, K, and L*) already suggested that genes with less condition-specific expression patterns have more genes with stoichiometrically similar expression patterns, and the proportionality here quantitatively

captures this property of omics dynamics. We remark that the proportionality relation was confirmed in all the omics data we analyzed in this paper (see Section 3.2 and *Appendix 1—figure 7I–N*).

## 2.2.2 Mathematics behind proportionality between stoichiometry conservation centrality $d_j$ and expression generality score $g_j$

The cosine similarity-based analyses involve normalization of expression vectors $\boldsymbol{p}_j$ by its $L^2$ norm $\left\|\boldsymbol{p}_j\right\|_2$ (*Equation 2.77*). Normalized expression vectors $\boldsymbol{p}_j/\left\|\boldsymbol{p}_j\right\|_2$ represent points on the first orthant division of a unit $(m-1)$-sphere in an $m$-dimensional space $(\sum_{i=1}^{m}(x_i)^2 = 1\ (x_1,\ldots,x_m \geq 0))$. $L^2$ normalization allows us to compare expression patterns without considering expression magnitudes, and an expression pattern is represented by a position on the unit $(m-1)$-sphere. Therefore, our cosine similarity-based quantification of stoichiometric balance in omics data is equivalent to evaluating distances (measured with angle) between points on the $(m-1)$-sphere.

Since stoichiometry conservation centrality $d_j$ is the sum of cosine similarities,

$$d_j = \sum_{i=1}^{n} \cos \theta_{\boldsymbol{p}_j \boldsymbol{p}_i} = \sum_{i=1}^{n} \frac{\boldsymbol{p}_j \cdot \boldsymbol{p}_i}{\|\boldsymbol{p}_j\|_2 \|\boldsymbol{p}_i\|_2} \tag{2.144}$$

$$= \frac{\boldsymbol{p}_j}{\|\boldsymbol{p}_j\|_2} \cdot \left( \sum_{i=1}^{n} \frac{\boldsymbol{p}_i}{\|\boldsymbol{p}_i\|_2} \right). \tag{2.145}$$

Defining

$$\tilde{\boldsymbol{p}}_{\text{tot}} := \sum_{i=1}^{n} \frac{\boldsymbol{p}_i}{\|\boldsymbol{p}_i\|_2}, \tag{2.146}$$

we obtain

$$\frac{1}{\sqrt{\sum_{i=1}^{n} d_i}} d_j = \frac{\boldsymbol{p}_j}{\|\boldsymbol{p}_j\|_2} \cdot \frac{\tilde{\boldsymbol{p}}_{\text{tot}}}{\|\tilde{\boldsymbol{p}}_{\text{tot}}\|_2}. \tag{2.147}$$

Note that $\|\tilde{\boldsymbol{p}}_{\text{tot}}\|_2 = \sqrt{\sum_{i=1}^{n} d_i}$. The last term $\tilde{\boldsymbol{p}}_{\text{tot}}/\|\tilde{\boldsymbol{p}}_{\text{tot}}\|_2$ is the normalized vector of the sum of all the normalized expression vectors, which we refer to as 'expression-pattern norm vector'. Therefore, *Equation 2.147* means that stoichiometry conservation centrality $d_j = \sum_i \cos \theta_{\boldsymbol{p}_j \boldsymbol{p}_i}$ is proportional to the cosine of the angle between the expression-pattern norm vector $\tilde{\boldsymbol{p}}_{\text{tot}}/\|\tilde{\boldsymbol{p}}_{\text{tot}}\|_2$ and protein $j$'s expression pattern $\boldsymbol{p}_j/\|\boldsymbol{p}_j\|_2$. The more distant the expression pattern of a protein is from that specified by the expression-pattern norm vector, the smaller $d_j$ is.

On the other hand, expression generality score $g_j$ is

$$g_j = \frac{\|\boldsymbol{p}_j\|_1}{\|\boldsymbol{p}_j\|_2} = \left\| \frac{\boldsymbol{p}_j}{\|\boldsymbol{p}_j\|_2} \right\|_1 \tag{2.148}$$

$$= \frac{\boldsymbol{p}_j}{\|\boldsymbol{p}_j\|_2} \cdot \mathbf{1}_m. \tag{2.149}$$

Therefore,

$$\frac{1}{\sqrt{m}} \frac{\|\boldsymbol{p}_j\|_1}{\|\boldsymbol{p}_j\|_2} = \frac{\boldsymbol{p}_j}{\|\boldsymbol{p}_j\|_2} \cdot \frac{\mathbf{1}_m}{\sqrt{m}}. \tag{2.150}$$

The last term $\mathbf{1}_m/\sqrt{m}$ is the normalized vector of $\mathbf{1}_m$, corresponding to the 'center' of the first orthant division of the unit $(m-1)$-sphere. In other words, $\mathbf{1}_m/\sqrt{m}$ represents 'perfectly even expression pattern' across conditions. Therefore, *Equation 2.150* means that the expression generality score $g_j = \|\boldsymbol{p}_j\|_1/\|\boldsymbol{p}_j\|_2$ is proportional to the cosine of the angle between the 'perfectly even expression pattern' $\mathbf{1}_m/\sqrt{m}$ and protein $j$'s expression pattern $\boldsymbol{p}_j/\|\boldsymbol{p}_j\|_2$. The more distant

the expression pattern of a protein is from the 'perfectly even expression pattern', the smaller the expression generality score is.

Comparing **Equations 2.147 and 2.150**, we see that if the expression-pattern norm vector $\tilde{\boldsymbol{p}}_{\text{tot}}/\|\tilde{\boldsymbol{p}}_{\text{tot}}\|_2$ and the perfectly even expression pattern $\mathbf{1}_m/\sqrt{m}$ are equal, a proportional relationship

$$\forall j, \; d_j = \sqrt{\frac{\sum_{i=1}^{n} d_i}{m}} \frac{\|\boldsymbol{p}_j\|_1}{\|\boldsymbol{p}_j\|_2} \tag{2.151}$$

holds. Note that this is equivalent to $\sqrt{m}\,\boldsymbol{b}_0 = \boldsymbol{b}_0^{\text{est}}$.

Conversely, if $\tilde{\boldsymbol{p}}_{\text{tot}}/\|\tilde{\boldsymbol{p}}_{\text{tot}}\|_2$ deviates from $\mathbf{1}_m/\sqrt{m}$, the proportional relation between $d_j$ and $g_j$ breaks. We found that the proteome data by **Schmidt et al., 2016** showed $\tilde{\boldsymbol{p}}_{\text{tot}}/\|\tilde{\boldsymbol{p}}_{\text{tot}}\|_2 \neq \mathbf{1}_m/\sqrt{m}$. Instead, we found that the values of the elements of $\tilde{\boldsymbol{p}}_{\text{tot}}$ increased approximately linearly with the population growth rates under corresponding conditions (**Appendix 1—figure 8D**). Such a strong positive correlation between the elements of $\tilde{\boldsymbol{p}}_{\text{tot}}$ and the population growth rates is nontrivial and suggests a new growth law constraining the total of relative expression level changes of all the proteins.

Next, we consider the consequence of this positive correlation between the elements of $\tilde{\boldsymbol{p}}_{\text{tot}}$ and the population growth rates. Let us consider the proteins whose expression generality score $g_j = \|\boldsymbol{p}_j\|_1/\|\boldsymbol{p}_j\|_2$ takes the minimum value one. Namely, the expression of these proteins is completely condition-specific (**Appendix 1—figure 8A and B**). For such proteins, only one component of the expression pattern vector $\boldsymbol{p}_j/\|\boldsymbol{p}_j\|_2$ is one, and the other components are zero. Thus, from **Equation 2.147**, their stoichiometry conservation centrality $d_j$ becomes proportional to the elements of $\tilde{\boldsymbol{p}}_{\text{tot}}$ corresponding to the conditions under which they are expressed. Since the values of the elements of $\tilde{\boldsymbol{p}}_{\text{tot}}$ are positively correlated with the population growth rates, $d_j$ of completely condition-specific proteins also exhibits a positive correlation with the growth rates under the conditions accompanying their expression (**Appendix 1—figure 8C**).

Such correlation can be confirmed for the proteins with nearly condition-specific expression patterns (PaaE, Asr, and DgoA in **Figure 7B and C**).

More generally, the deviation of $d_j$ from the perfect proportionality line can be understood by the relation

$$\frac{d_j}{\sqrt{\sum_{i=1}^{n} d_i}} - \frac{\|\boldsymbol{p}_j\|_1}{\sqrt{m}\|\boldsymbol{p}_j\|_2} = \frac{\boldsymbol{p}_j}{\|\boldsymbol{p}_j\|_2} \cdot \left( \frac{\tilde{\boldsymbol{p}}_{\text{tot}}}{\|\tilde{\boldsymbol{p}}_{\text{tot}}\|_2} - \frac{\mathbf{1}_m}{\sqrt{m}} \right), \tag{2.152}$$

which can be derived from **Equations 2.147, 2.150**. Note that the values of the elements of the last term $\tilde{\boldsymbol{p}}_{\text{tot}}/\|\tilde{\boldsymbol{p}}_{\text{tot}}\|_2 - \mathbf{1}_m/\sqrt{m}$ also increase with the population growth rates, being positive under fast growth conditions and negative under slow growth conditions (**Appendix 1—figure 8D**). Therefore, when protein $j$ tends to be expressed higher under fast growth conditions, i.e., when the elements of $\boldsymbol{p}_j/\|\boldsymbol{p}_j\|_2$ corresponding to the fast growth conditions are relatively larger than those corresponding to the slow growth conditions, the left-hand side of **Equation 2.152** becomes positive, and its $d_j$ resides above the perfect proportionality line. On the other hand, when protein $j$ tends to be expressed higher under slow growth conditions, its $d_j$ resides below the perfect proportionality line.

In the $g_j$-$d_j$ plot in **Figure 7A and B**, we find several stretches of protein clusters above and below the perfect proportionality line. As expected from the argument above, each cluster corresponds to a group of proteins with similar expression patterns, and their positions relative to the proportionality line characterize the condition under which they are expressed the most (**Figure 7C**).

In summary, visualizing omics data by using the stoichiometry conservation centrality $d_j$ and the expression generality score $g_j$ allows us to systematically characterize the condition-dependent expression pattern of each protein on the basis of its position in the plot. Interestingly, this systematic characterization of gene expression patterns (the relation between $d_j$ and $g_j$) was derived from our mathematical analyses of the correspondences between Raman and omics as we explained above.

# 3 Extended data analysis

## 3.1 Growth laws

### 3.1.1 Single-gene-level growth law

Bacterial growth law states that the total abundances of ribosomal components increase linearly with growth rate (**Neidhardt and Magasanik, 1960**; **Scott et al., 2010**; **Bremer and Dennis, 2008**). The homeostatic core (the largest SCG) identified in our analysis contains many ribosomal proteins. Hence, it is plausible that the total abundance of homeostatic core proteins also increases linearly with growth rate, which we indeed found true (**Appendix 1—figure 5A**). Furthermore, the abundance ratios of homeostatic core proteins are conserved across conditions. Therefore, the intracellular abundance of each protein species in the homeostatic core is expected to increase linearly with growth rate.

Let $p_{\epsilon j}$ be the abundance of protein $j$ in the homeostatic core in environment $\epsilon$. Since this protein conserves the stoichiometry with the other homeostatic core proteins across conditions,

$$\frac{p_{\epsilon i}}{p_{\epsilon j}} = \alpha_{ij} \tag{3.1}$$

in any environments $\epsilon$ ($\alpha_{ij}$ is the environment-independent abundance ratio of the homeostatic core protein $i$ to protein $j$).

Let $M_\epsilon = \sum_i p_{\epsilon i} = p_{\epsilon j} \sum_i \alpha_{ij}$ be the total abundance of homeostatic core proteins in environment $\epsilon$. The growth law for the homeostatic core is

$$M_\epsilon = a + b g_\epsilon, \tag{3.2}$$

where $g_\epsilon$ is the growth rate in environment $\epsilon$, $a$ is the $y$-intercept, and $b$ is the slope of the linear relation. Therefore,

$$p_{\epsilon j} = \frac{a}{\sum_i \alpha_{ij}} + \frac{b}{\sum_i \alpha_{ij}} g_\epsilon. \tag{3.3}$$

This shows that the abundance of homeostatic core proteins satisfies single-gene-level growth law.

### 3.1.2 Extended verification of stoichiometry conservation

When the abundance ratios between protein $i$ and protein $j$ are conserved,

$$\frac{p_{cj}}{p_{ci}} = \frac{p_{sj}}{p_{si}}, \tag{3.4}$$

where $c$ and $s$ specify the environments ($s$ signifies the standard environment). Hence,

$$\frac{p_{cj}}{p_{sj}} = \frac{p_{ci}}{p_{si}} = \gamma_c, \tag{3.5}$$

where $\gamma_c$ is the common abundance ratio of stoichiometry-conserving proteins with respect to the condition $c$. Note that $\gamma_c$ is common among all the proteins in a stoichiometry-conserving group.

From **Equation 3.5**,

$$\log p_{cj} = \log p_{sj} + \log \gamma_c. \tag{3.6}$$

Therefore, plotting $\log p_{cj}$ against $\log p_{sj}$ should find the stoichiometry-conserving proteins aligned on a straight line with a slope of 1. We indeed find such plots for the homeostatic core proteins (**Appendix 1—figure 5B and C**). This result confirms their stoichiometry conservation from a different perspective.

### 3.1.3 Linear dependence of common abundance ratio on growth rate

Since the total amount of homeostatic core proteins increases linearly with growth rate (**Appendix 1—figure 5A** and **Equation 3.2**),

$$\sum_j p_{cj} - \sum_j p_{sj} = b\left(g_c - g_s\right).$$ (3.7)

Since $\sum_j p_{cj} = \gamma_c \sum_j p_{sj}$,

$$\left(\gamma_c - 1\right) \sum_j p_{sj} = b\left(g_c - g_s\right).$$ (3.8)

Hence,

$$\gamma_c = \left(1 - \frac{bg_s}{M_s}\right) + \frac{b}{M_s}g_c,$$ (3.9)

where $M_s = \sum_j p_{sj}$. Therefore, the common abundance ratio $\gamma_c$ also increases linearly with growth rate.

Estimating $\Gamma_c := \log_{10} \gamma_c$ as the $y$-intercepts of the regression lines with a slope of 1 (*Appendix 1—figure 5B*), we confirmed this linear dependence of common abundance ratio of homeostatic core proteins on growth rate (*Appendix 1—figure 5D*).

## 3.2 Generality of the results

### 3.2.1 Additional datasets with Raman data

#### Correspondence of three types of space

The correspondences among LDA Raman, Raman-omics normalized coefficient omics structure, and csLE omics structure were also observed in other datasets. The datasets include Raman (this paper) and proteome (*Schmidt et al., 2016*) data of *E. coli* with different genotypes (BW25113, MG1655, and NCM3722) cultured in the 'LB' medium (*Appendix 1—figure 11A–E*) and in the 'Glucose' medium (*Appendix 1—figure 11F–J*), and Raman and transcriptome data of *S. pombe* cultured under 10 different environmental conditions (*Appendix 1—figure 11K–O*; *Kobayashi-Kirschvink et al., 2018*).

#### Comparison of matrices obtained by mathematical analyses

In addition to the comparison of three types of space, we also examined the matrices on the basis of results of the mathematical analyses (*Appendix 1—table 10*) using the aforementioned additional datasets; the closeness of $B_E^{\text{norm}}$ and $B_E^{\text{est,norm}}$ (*Appendix 1—figure 12F* for Raman-proteome of *E. coli* with different genotypes cultured in 'LB', *Appendix 1—figure 12L* for Raman-proteome of *E. coli* with different genotypes cultured in 'Glucose', and *Appendix 1—figure 12R* for Raman-transcriptome of *S. pombe* cultured in 10 environment conditions), the closeness of $\Theta$ to the identity matrix (*Appendix 1—figure 12A–D* for Raman-proteome of *E. coli* with different genotypes cultured in 'LB', *Appendix 1—figure 12G–J* for Raman-proteome of *E. coli* with different genotypes cultured in 'Glucose', and *Appendix 1—figure 12M–P* for Raman-transcriptome of *S. pombe* cultured in 10 environment conditions), and the correspondence between $\sqrt{m}\,\text{diag}\left(\boldsymbol{b}_0\right)$ and $\text{diag}\left(\boldsymbol{b}_0^{\text{est}}\right)$ (*Appendix 1—figure 12E* for Raman-proteome of *E. coli* with different genotypes cultured in 'LB', *Appendix 1—figure 12K* for Raman-proteome of *E. coli* with different genotypes cultured in 'Glucose', and *Appendix 1—figure 12Q* for Raman-transcriptome of *S. pombe* cultured under in 10 environment conditions).

We confirmed that the same results hold for these additional datasets. Note that the correspondence between $\sqrt{m}\,\text{diag}\left(\boldsymbol{b}_0\right)$ and $\text{diag}\left(\boldsymbol{b}_0^{\text{est}}\right)$ does not involve Raman data. It is an intrinsic property of the omics data.

#### Proportionality between expression generality and stoichiometry conservation centrality

The correspondence between $\sqrt{m}\,\text{diag}\left(\boldsymbol{b}_0\right)$ and $\text{diag}\left(\boldsymbol{b}_0^{\text{est}}\right)$ suggested the proportionality between expression generality score $g_j$ and stoichiometry conservation centrality $d_j$ in these omics data. We confirmed that the same results hold for these additional datasets (*Appendix 1—figure 7I–N*).

#### Correlation between $\tilde{p}_{\text{tot}}$ and growth rates

For the proteome data of *E. coli* with different genotypes (BW25113, MG1655, and NCM3722) cultured in 'LB' and in 'Glucose', growth rates were also reported (*Schmidt et al., 2016*). We

confirmed a positive correlation between the elements of $\tilde{\boldsymbol{p}}_{\text{tot}}$ and growth rates for these datasets (*Appendix 1—figure 8E*). See also the deviation from the proportionality line in the $g_j$-$d_j$ plot (*Appendix 1—figure 7I–J*).

## Biological relevance of centrality of csLE structure

We also confirmed centrality-essentiality correlation and centrality-evolutionary conservation correlation in the *S. pombe* transcriptome data (*Appendix 1—figure 6B and E–G*).

## Degree distribution and its destruction by randomization

Degree (stoichiometry conservation centrality) distributions of csLE structure of the additional datasets also showed a similar pattern as the main data, and randomization of the omics data breaks the strong correlation of expression patterns in the actual data (*Appendix 1—figure 7C, D, and H*).

### 3.2.2 Additional datasets without Raman data

Examining proteome structures with csLE does not require Raman data. Therefore, we additionally analyzed publicly available proteome data of *M. tuberculosis* and *M. bovis* under the growth conditions with distinct oxygen levels (*Schubert et al., 2015*), and the proteome data of *S. cerevisiae* under various environmental conditions (*Lahtvee et al., 2017*).

We characterized csLE structures of these datasets (*Appendix 1—figure 13*). Furthermore, we confirmed the proportionality between expression generality score $g_j$ and stoichiometry conservation centrality $d_j$ (*Appendix 1—figure 7K–M*). For the proteome data of *S. cerevisiae*, growth rates were also reported (*Lahtvee et al., 2017*). The *S. cerevisiae* cells were cultured in chemostat at the same dilution rate in any condition. In fact, we observed little variation of $\tilde{\boldsymbol{p}}_{\text{tot}}$ (*Appendix 1—figure 8F*), which leads to little deviation from the proportionality line (*Appendix 1—figure 7M*).

Degree (stoichiometry conservation centrality) distributions of csLE structure of the additional datasets also showed a similar pattern as the main data, and randomization of the omics data breaks the strong correlation of expression patterns in the actual data (*Appendix 1—figure 7E–G*).

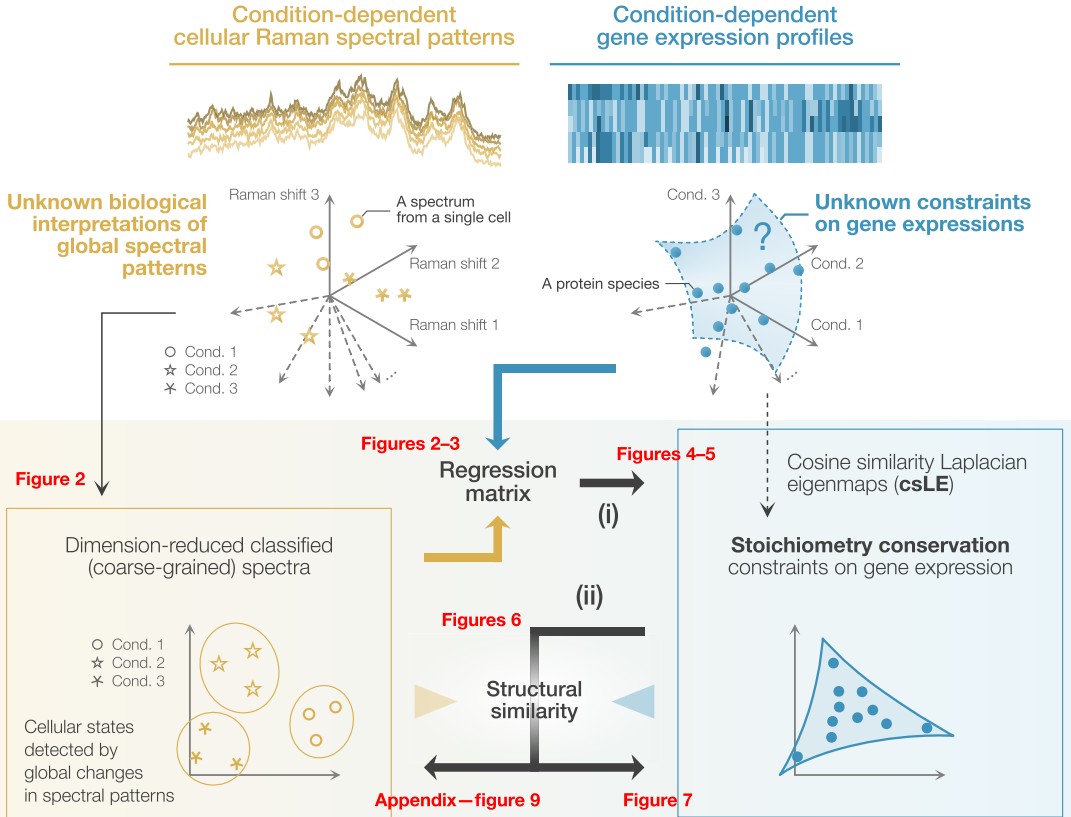

(i) develops csLE and uncovers stoichiometry conservation architecture of gene expressions, on which gene hierarchy correlates with its essentiality and evolutionary conservation.

(ii) finds a structural similarity, which indicates that spectra reflect the changes in omics profiles under the constraints of stoichiometry conservation and that stoichiometry conservation strength of each gene is proportional to expression generality score.

**Appendix 1—figure 1.** Schematic illustration of the approach in this study. Related to *Figure 1*. Raman spectra and gene expression profiles are both high-dimensional vectors and can be represented as points in high-dimensional spaces. Coarse-graining Raman spectra by dimensional reduction finds condition-dependent differences in their global spectral patterns (see *Figure 2*). The dimension-reduced spectra were linked to and used to predict condition-dependent global gene expression profiles (see *Figure 2*), which implies that global changes in spectral patterns detect differences in cellular physiological states. The analysis of this linkage led us to discover a stoichiometry-conserving constraint on gene expression, which enabled us to represent gene expression profiles in a functionally relevant low-dimensional space (i; see also *Figures 3–5*). Then, we find a nontrivial correspondence between these low-dimensional Raman and gene expression spaces (ii; see also *Figure 6*). This correspondence provides an omics-level interpretation of global Raman spectral patterns and a quantitative constraint between expression generality and stoichiometry conservation centrality (ii; see also *Figure 7*, *Appendix 1—figure 9*).

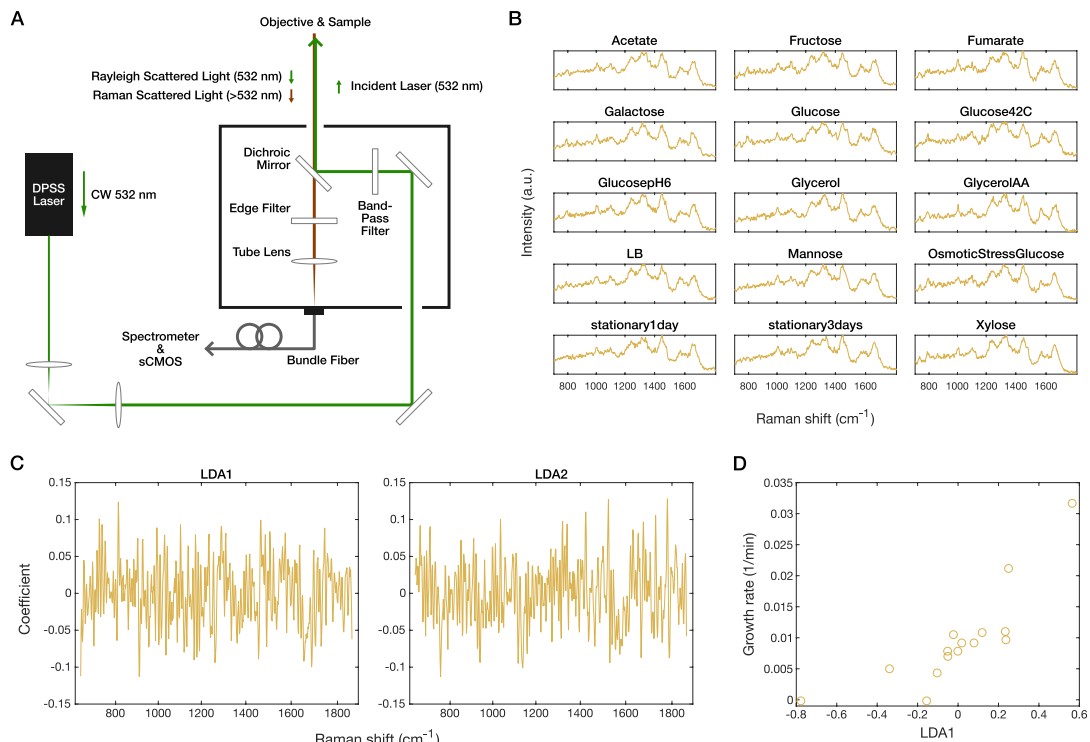

**Appendix 1—figure 2.** Custom-built Raman microscope and analyses of *E. coli* Raman spectra. Related to *Figure 2*. (**A**) Schematic diagram of the Raman microscope used in this study. (**B**) Representative Raman spectra from single *E. coli* cells. The fingerprint region of one spectrum is shown for each condition. (**C**) Linear superposition of Raman shifts. Each linear discriminant analysis (LDA) axis is a linear superposition of Raman shifts. These figures show the coefficients for LDA1 (left) and LDA2 (right). (**D**) Relationship between Raman LDA1 axis and growth rates. The horizontal axis represents Raman LDA1 axis. The vertical axis represents growth rates measured in *Schmidt et al., 2016*. Each point corresponds to the data for one condition. Pearson correlation coefficient is 0.81±0.09.

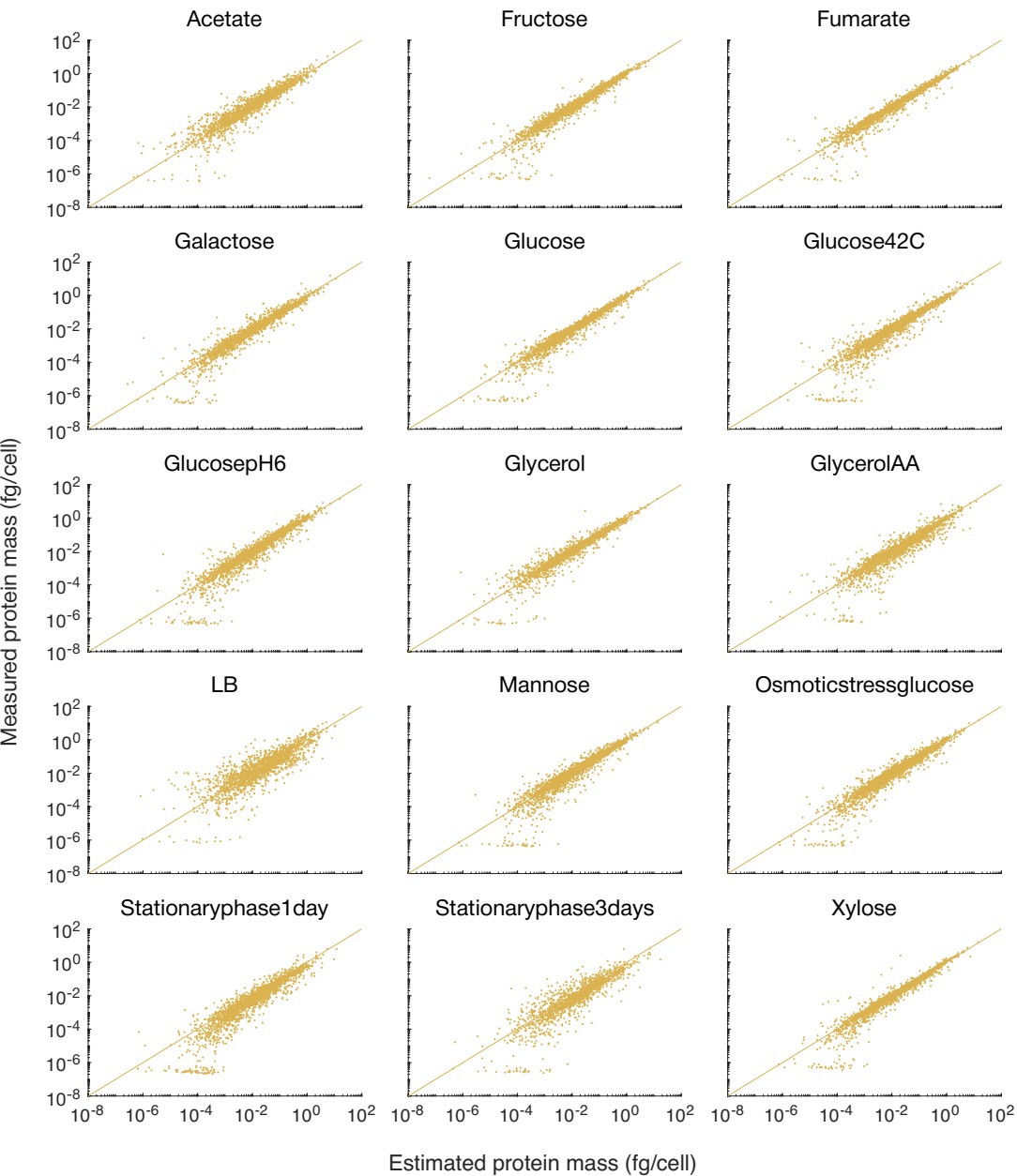

**Appendix 1—figure 3.** Estimation of proteomes from Raman spectra. Related to *Figure 2*. Comparing the measured proteomes with those estimated from Raman spectra. The horizontal and vertical axes represent the estimated and measured proteomes, respectively. Proteins with negative estimated abundance are not shown in these figures. The conditions with the largest and the second largest numbers of proteins with negative estimated abundance were 'stationary3days' (666 proteins) and 'LB' (359 proteins). The conditions with the fewest and the second fewest negatively estimated proteins were 'GlucosepH6' (0 proteins) and 'Xylose' (7 proteins).

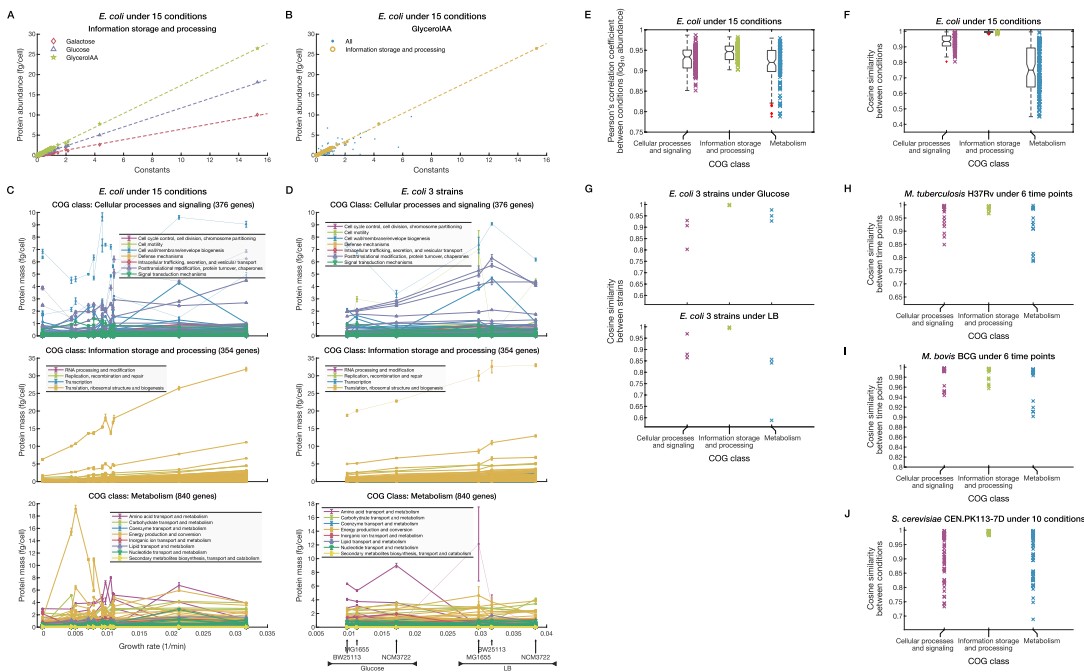

**Appendix 1—figure 4.** Comparison of stoichiometry conservation among Clusters of Orthologous Group (COG) classes. Related to **Figure 3**. (**A and B**) Relations between protein abundance and constant terms of Raman-proteome coefficients. The horizontal axes are $\boldsymbol{b}_0$ (constant terms), and the vertical axes are $\hat{\boldsymbol{p}}_i^\top$ (protein abundance). Dashed lines are the least squares regression lines with intercept zero for information storage and processing (ISP) COG class members. The average of $B_{-i}^{\text{est}}$ was used as an estimate of $\boldsymbol{B}$ here. In (**A**), only ISP COG class members are shown for three representative conditions: 'Galactose', 'Glucose', and 'GlycerolAA'. In (**B**), all proteins are shown for a representative condition, 'GlycerolAA'. (**C**) Relations between protein abundance and growth rates of *E. coli* under 15 environmental conditions. We analyzed the absolute quantitative proteome data, growth rate data, and COG annotation reported by **Schmidt et al., 2016**. Lines represent different protein species. Error bars are standard errors. The top panel is for the Cellular Processes and Signaling COG class; the middle is for the ISP COG class; and the bottom is for the Metabolism COG class. (**D**) Relations between protein abundance and growth rates of three *E. coli* strains (BW25113, MG1655, and NCM3722) under two culture conditions. We again analyzed the data by **Schmidt et al., 2016**. Lines represent different protein species. Error bars are standard errors. (**E and F**) COG class-dependent expression pattern similarity of *E. coli* proteomes between conditions. The *E. coli* proteome data under the 15 different environmental conditions were analyzed. The similarity is evaluated by Pearson correlation coefficients of log expression levels in (**E**) and by cosine similarity in (**F**). We consider all the combinations of the 15 conditions. Thus, there are 105 data points for each COG class. The box-and-whisker plots summarize the distributions of the points. The lines inside the boxes denote the medians. The top and bottom edges of the boxes denote the 25th percentiles and 75th percentiles, respectively. Note that (**E**) and (**F**) are evaluations of the same data used in **Figure 3B** in the main text with different similarity indices. (**G**) COG class-dependent expression pattern similarity between different strains of *E. coli* (BW25113, MG1655, and NCM3722). The absolute quantitative proteome data and COG annotation were taken from **Schmidt et al., 2016**. The similarity was evaluated by cosine similarity. The data contain three strains. Thus, there are three points for each COG class. The top panel is for the 'Glucose' condition, and the bottom is for the 'LB' condition. (**H–J**) COG class-dependent expression pattern similarity in other organisms. (**H**) is for *M. tuberculosis* (data from **Schubert et al., 2015**; six environmental conditions [time points]), (**I**) for *M. bovis* (data from **Schubert et al., 2015**; six environmental conditions [time points]), and (**J**) for *S. cerevisiae* (data from **Lahtvee et al., 2017**; 10 environmental conditions). The COG annotations were taken from the December 2014 release of 2003-2014 COGs (**Galperin et al., 2015**) and the Release 3 of 'Mycobrowser' (**Kapopoulou et al., 2011**) for (**H**) and (**I**) and from the Comprehensive Sake Yeast Genome Database (S288C strain) (**Akao et al., 2011**) for (**J**). The unit for protein abundance was fg/cell for (**H**) and (**I**) and fg in pg dry cell weight for (**J**).

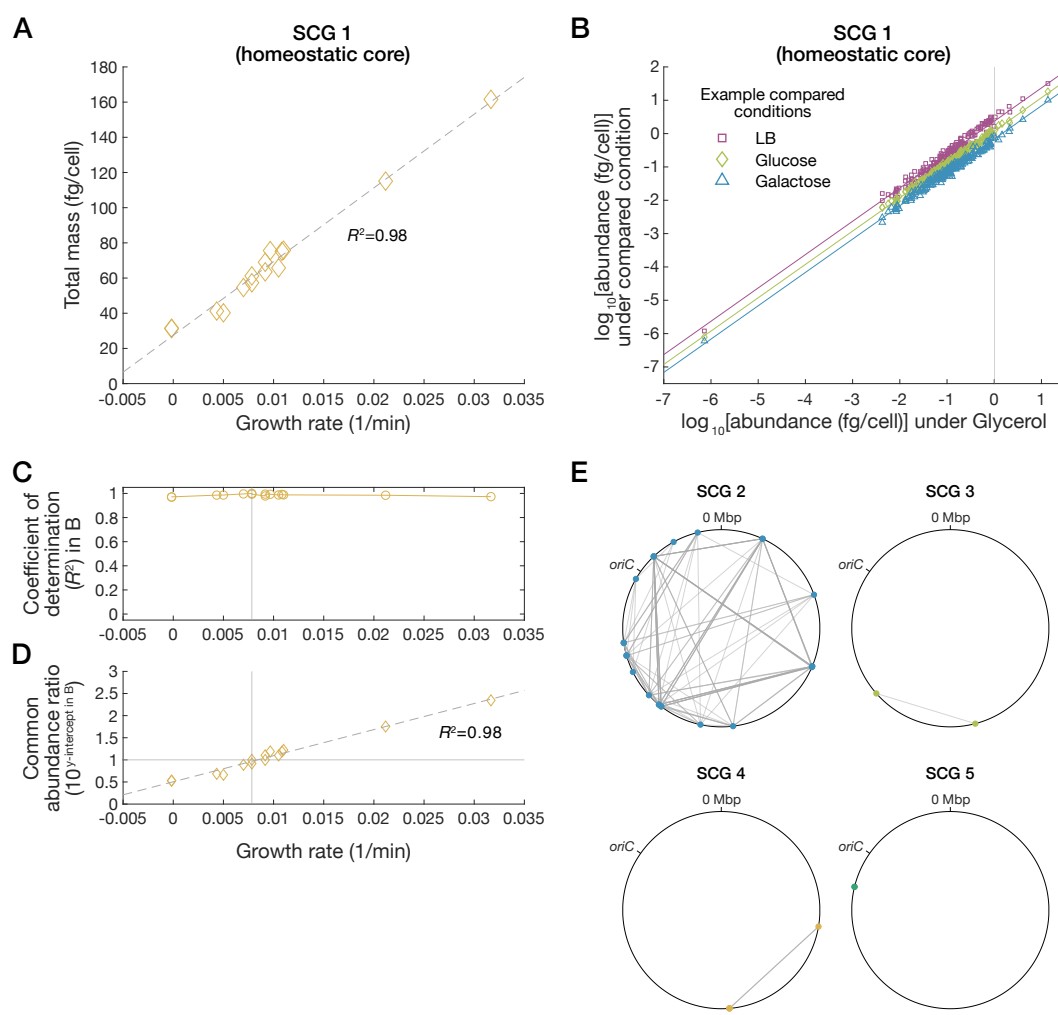

**Appendix 1—figure 5.** Single-gene-level growth law in the homeostatic core. Related to *Figure 4*. (**A**) Relationship between population growth rates and total abundance of SCG 1 (homeostatic core) proteins. Here, we analyzed the *E. coli* proteome data (*Schmidt et al., 2016*), focusing on the 15 conditions for which we obtained Raman data. The dashed line is the least squares regression line. (**B**) Scatterplots of log abundance of SCG 1 (homeostatic core) proteins. Here, the proteomes under three representative conditions, 'LB', 'Glucose', and 'Galactose', are compared with that under the standard condition 'Glycerol'. Each colored line is the linear regression line with slope one for the points with the same color. The vertical line is $x = 0$. (**C**) Relationship between population growth rate and coefficient of determination of linear regression in (**B**). The vertical line represents the growth rate under the standard condition ('Glycerol'). (**D**) Linear relationship between common abundance ratio and growth rates. The vertical axis represents $10^{\Gamma_c}$, where $\Gamma_c$ is the $y$-intercepts in (**B**) (see Section 3.1.2 in Appendix). The dashed line is the linear regression line. The horizontal line is $y = 1$, and the $x$ coordinate of the vertical line is the growth rate under the standard condition ('Glycerol'). (**E**) The gene loci of the proteins belonging to the condition-specific stoichiometrically conserved groups (SCGs) on the chromosome (ASM75055v1.46; *Howe et al., 2020*). Colored dots are nodes (genes), and gray lines are edges (high cosine similarity relationships). The edge in the map of SCG 5 cannot be seen because their gene loci are clustered in close proximity in the same operon.

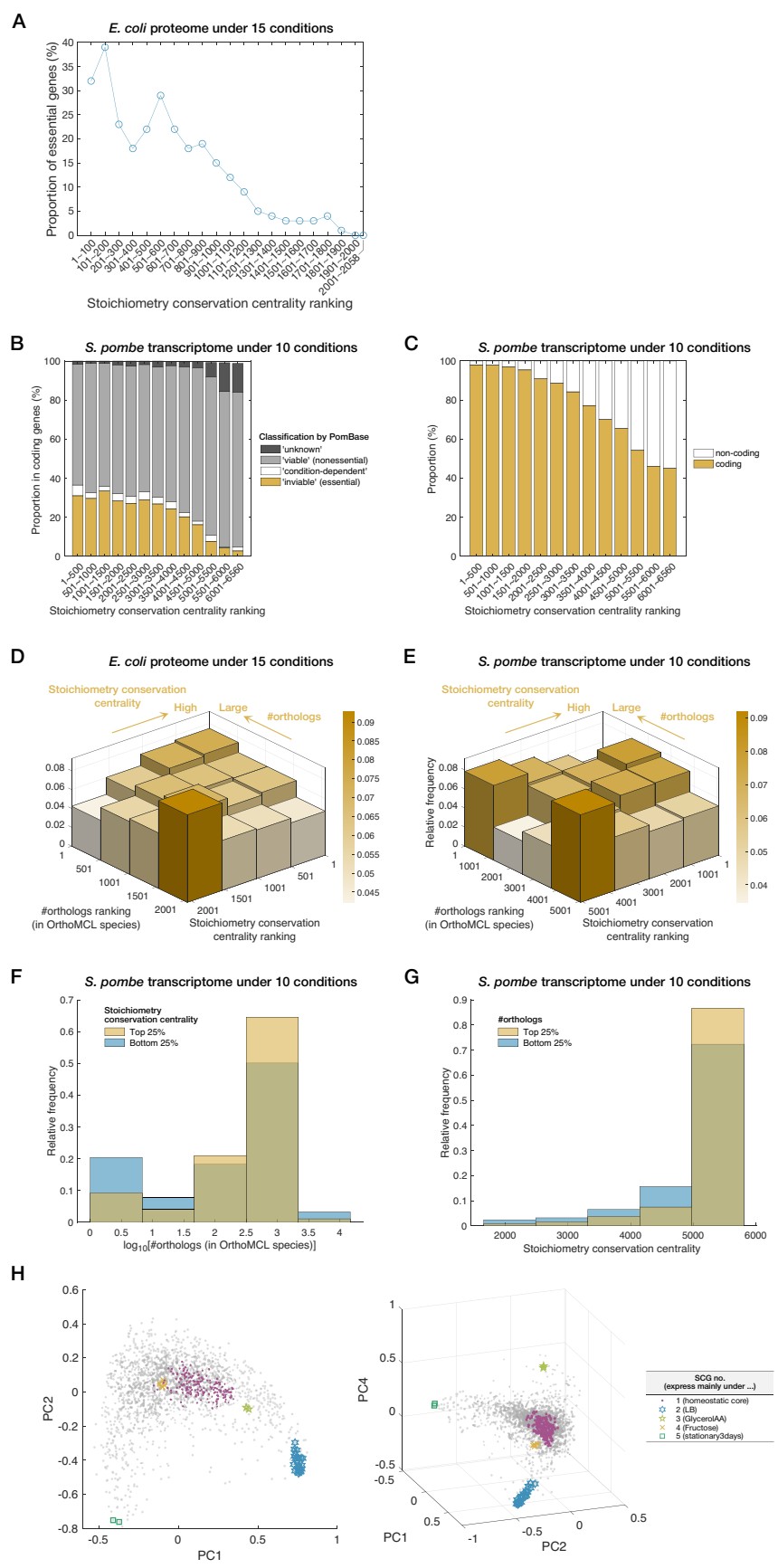

**Appendix 1—figure 6.** Functional relevance of stoichiometry conservation centrality. Related to *Figure 5*. (**A**) Relationship between gene essentiality and stoichiometry conservation centrality in *E. coli*. The proportion of essential genes is plotted for each stoichiometry conservation centrality rank range. In this plot, we calculated stoichiometry conservation centrality based on the *E. coli* proteome data (*Schmidt et al., 2016*) under the 15 conditions for which we obtained Raman data. The list of essential genes was downloaded from EcoCyc (*Keseler et al., 2017*). (**B**) Relationship between gene essentiality and stoichiometry conservation centrality in *S. pombe*. We calculated stoichiometry conservation centrality based on the *S. pombe* transcriptome data reported in *Kobayashi-Kirschvink et al., 2018*. Only coding genes are considered in this plot, though stoichiometry conservation centrality values were calculated using both coding and non-coding genes. Gene classification is based on PomBase (*Harris et al., 2022*). Some bins do not reach 100% in sum because 11 coding genes in the *S. pombe* transcriptome data were not found in the current PomBase. (**C**) Relationship between ratio of coding genes and stoichiometry conservation centrality in the *S. pombe* transcriptome data. The coding/non-coding assignment is based on PomBase (*Harris et al., 2022*). (**D**) Correlation between stoichiometry conservation and evolutionary conservation. In this plot, we calculated stoichiometry conservation centrality based on the *E. coli* proteome data (*Schmidt et al., 2016*) under the 15 conditions for which we obtained Raman data. Colors represent the height of each bar. The distributions of stoichiometry conservation centrality were compared between the top 25% and the bottom 25% fractions in the number of orthologs rankings. The fraction with many orthologs tends to have higher stoichiometry conservation centrality (one-sided Brunner-Munzel test, $p = 7.84 \times 10^{-15}$). The distributions of the number of orthologs were compared between the top 25% and the bottom 25% stoichiometry conservation centrality fractions. The high centrality fraction tends to have more orthologs (one-sided Brunner-Munzel test, $p = 1.46 \times 10^{-11}$). Ortholog data were taken from OrthoMCL-DB (*Chen et al., 2006*). (**E–G**) Correlation between stoichiometry conservation and evolutionary conservation in *S. pombe*. We calculated stoichiometry conservation centrality based on the *S. pombe* transcriptome data reported in *Kobayashi-Kirschvink et al., 2018*. In (**E**), the result is shown by a two-dimensional histogram. Colors represent the height of each bar. The distributions of the number of orthologs were compared between the top 25% and the bottom 25% stoichiometry conservation centrality fractions. The high centrality fraction tends to have more orthologs (one-sided Brunner-Munzel test, $p = 0.00548$). The direct comparison between the two fractions is shown in (**F**). The distributions of stoichiometry conservation centrality were compared between the top 25% and the bottom 25% fractions in the number of orthologs rankings. The fraction with many orthologs tends to have higher stoichiometry conservation centrality (one-sided Brunner-Munzel test, $p = 0.00270$). The direct comparison between the two fractions is shown in (**G**). Ortholog data were taken from OrthoMCL-DB (*Chen et al., 2006*). (**H**) Applying principal component analysis (PCA) to $L^2$-normalized proteomes. PCA (with mean centering) was applied to $L^2$-normalized proteome data $\begin{bmatrix} \boldsymbol{p}_1 / \|\boldsymbol{p}_1\|_2 & \cdots & \boldsymbol{p}_n / \|\boldsymbol{p}_n\|_2 \end{bmatrix}$. Here, we analyzed the *E. coli* proteome data under the 15 conditions for which we obtained Raman data. The left is a projection onto a two-dimensional space, and the right is a projection onto a three-dimensional space. The axes for visualization were selected by considering similarity to the cosine similarity LE (csLE) structure.

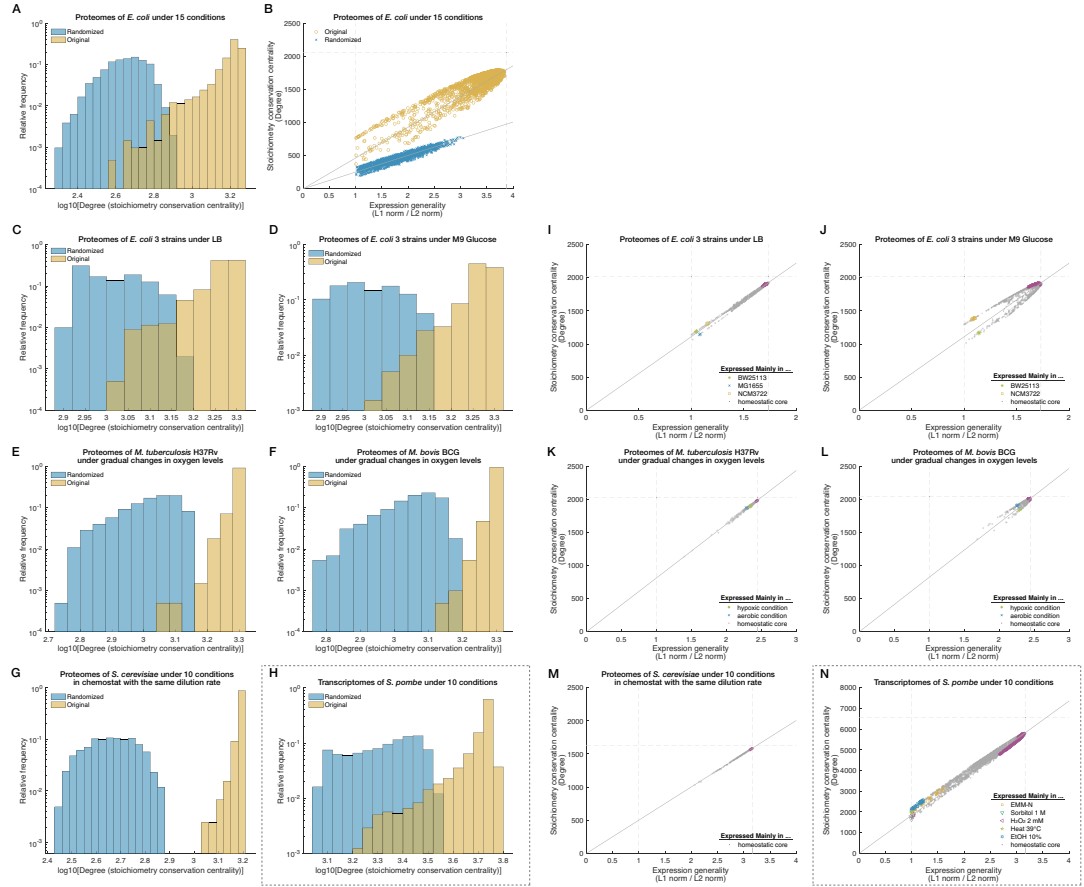

**Appendix 1—figure 7.** Distributions and constraints with respect to stoichiometry conservation centrality (degree). Related to **Figure 5** and **Figure 7**. (**A**) Comparison of degree (stoichiometry conservation centrality) distributions between original (yellow) and randomized (blue) *E. coli* proteome data. We created randomized proteome data by shuffling the expression levels across the protein species within each condition. We used the *E. coli* proteome data (**Schmidt et al., 2016**) under the 15 conditions for which we obtained Raman data. (**B**) Comparison of the $g_j$-$d_j$ relationships between original (yellow) and randomized data (blue). The horizontal axis is expression generality score ($g_j = L^1$ norm/$L^2$ norm), and the vertical axis is stoichiometry conservation centrality ($d_j$: degree). Each dot represents a protein species. The dashed lines are $y = n$, $x = 1$, $\sqrt{m}$ ($n = 2058, m = 15$). The solid lines are $y = \sqrt{\sum_i d_i / m}\ x$. (**C–H**) Degree (stoichiometry conservation centrality) distributions for additional datasets. Yellow histograms are for the original data, and blue histograms are for the randomized data. (**C**) For the proteomes of three *E. coli* strains (BW25113, MG1655, and NCM3722) in LB (**Schmidt et al., 2016**); (**D**) for the proteomes of the three *E. coli* strains in M9 Glucose (**Schmidt et al., 2016**); (**E**) for the proteomes of *M. tuberculosis* (**Schubert et al., 2015**); (**F**) for the proteomes of *M. bovis* (**Schubert et al., 2015**); (**G**) for the proteomes of *S. cerevisiae* (**Lahtvee et al., 2017**); and (**H**) for the transcriptomes of *S. pombe* (**Kobayashi-Kirschvink et al., 2018**). (**I–N**) $g_j$-$d_j$ relationships for additional datasets. Each gray dot represents a protein species. The proteins belonging to the homeostatic core in each dataset are shown in magenta; those belonging to condition-specific stoichiometrically conserved groups (SCGs) are indicated in different colors in each plot. See the caption of **Appendix 1—figures 11 and 13** for the cosine similarity threshold to specify the homeostatic core and the condition-specific SCGs in each dataset. The dashed lines are $y = n, x = 1, \sqrt{m}$. The solid lines through the origins are $y = \sqrt{\sum_{i=1}^{n} d_i / m}\ x$ (**I**) for the proteomes of the three *E. coli* strains in LB (**Schmidt et al., 2016**); (**J**) for the proteomes of the three *E. coli* strains in M9 Glucose (**Schmidt et al., 2016**); (**K**) for the proteomes of *M. tuberculosis* (**Schubert et al., 2015**); (**L**) for the proteomes of *M. bovis* (**Schubert et al., 2015**) (**M**) for the proteomes of *S. cerevisiae* (**Lahtvee et al., 2017**); and (**N**) for the transcriptomes of *S. pombe* (**Kobayashi-Kirschvink et al., 2018**).

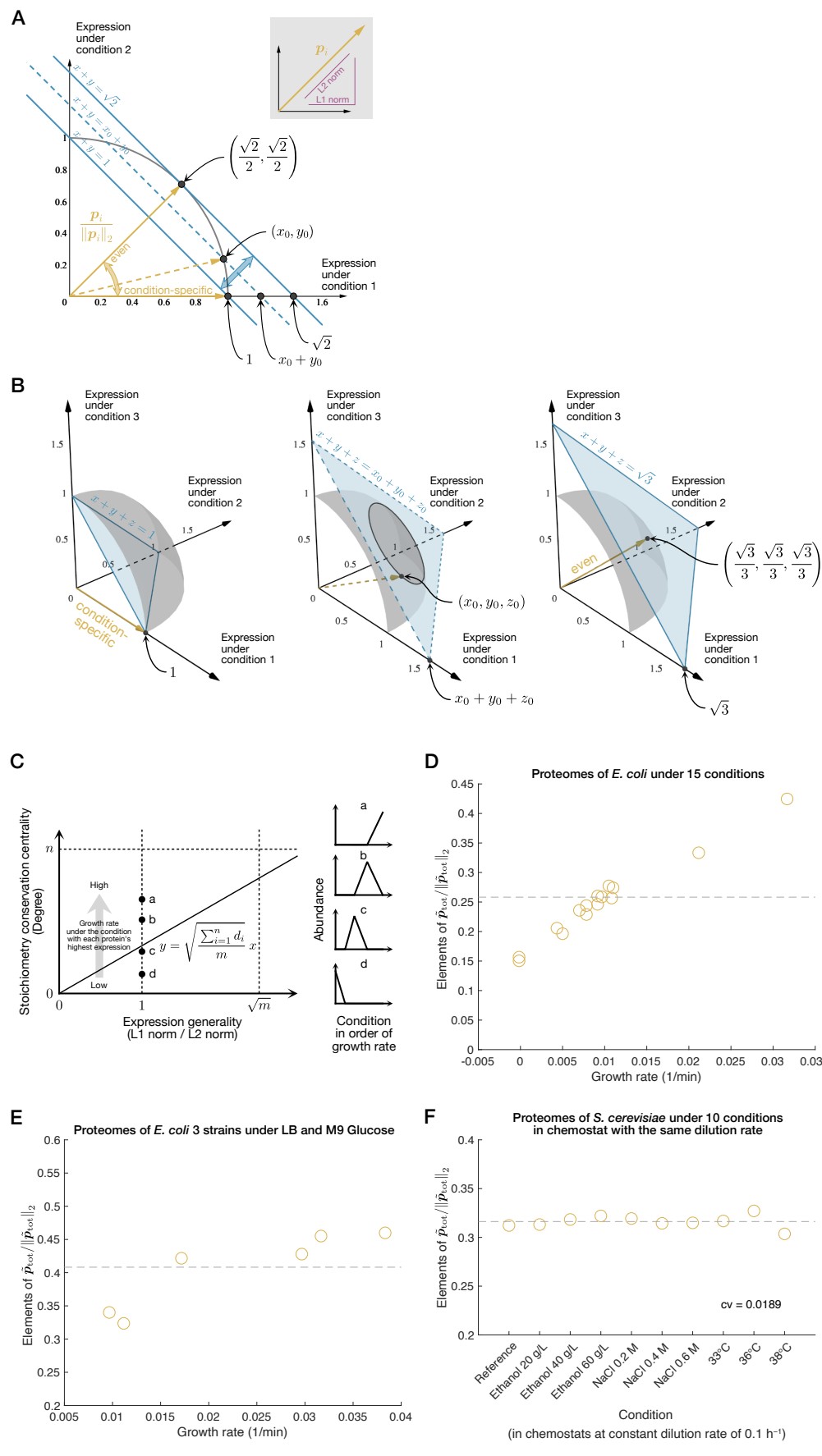

**Appendix 1—figure 8.** Properties of normalized expression vectors. Related to *Figure 7*. (**A and B**) Schematic explanation for the interpretation of the $L^1$ norm/$L^2$ norm ratio of expression vectors as an index of expression generality. (**A**) is a two-dimensional case, and (**B**) is a three-dimensional case. The inset in (**A**) schematically explains $L^1$ norm and $L^2$ norm of an expression vector. See 'Interpretation of $L^1$ norm/$L^2$ norm ratio of an expression vector as a quantitative measure of expression generality' in Materials and methods for details. (**C**) Schematic explanation for deviations of points from the proportionality line in the $g_j$-$d_j$ plots. Here, we consider four condition-specific protein species a, b, c, and d labeled in the descending order of growth rates under the conditions accompanying their expression. Note that their $L^1$ norm/$L^2$ norm ratios are all one on the horizontal axis. One can show that the degree (stoichiometry conservation centrality) $d_j$ is proportional to the inner product of $L^2$-normalized expression vector $\boldsymbol{p}_j/\|\boldsymbol{p}_j\|_2$ and the expression norm vector $\tilde{\boldsymbol{p}}_{\mathrm{tot}}/\|\tilde{\boldsymbol{p}}_{\mathrm{tot}}\|_2$ (see *Equation 2.147* in Section 2.2.2). Since the elements of $\tilde{\boldsymbol{p}}_{\mathrm{tot}}/\|\tilde{\boldsymbol{p}}_{\mathrm{tot}}\|_2$ increase approximately linearly with growth rates of the corresponding conditions (see **D**), the degrees (stoichiometry conservation centrality values) decrease from a to d in the order of growth rates. (**D–F**) Correlation between elements of $\tilde{\boldsymbol{p}}_{\mathrm{tot}}$ and population growth rates. The vertical axis represents the elements of $\tilde{\boldsymbol{p}}_{\mathrm{tot}}/\|\tilde{\boldsymbol{p}}_{\mathrm{tot}}\|_2$, and the horizontal axis represents the population growth rates. The dashed lines are $y = 1/\sqrt{m}$. (**D**) is the result from the analysis of the *E. coli* proteome data (*Schmidt et al., 2016*) under the 15 conditions for which we obtained Raman data ($m = 15$). (**E**) is the result from the analysis of the proteome data of three strains of *E. coli* (BW25113, MG1655, and NCM3722) under 'LB' and 'Glucose' conditions ($m = 6$) (*Schmidt et al., 2016*). (**F**) is the result from the analysis of the proteome data of *S. cerevisiae* under 10 different conditions ($m = 10$) (*Lahtvee et al., 2017*). The cells were cultured in a chemostat with the same dilution rate. The numbers of analyzable protein species and the numbers of conditions were different between (**D**) and (**E**). Thus, the values of the vertical axes cannot be compared directly between them.

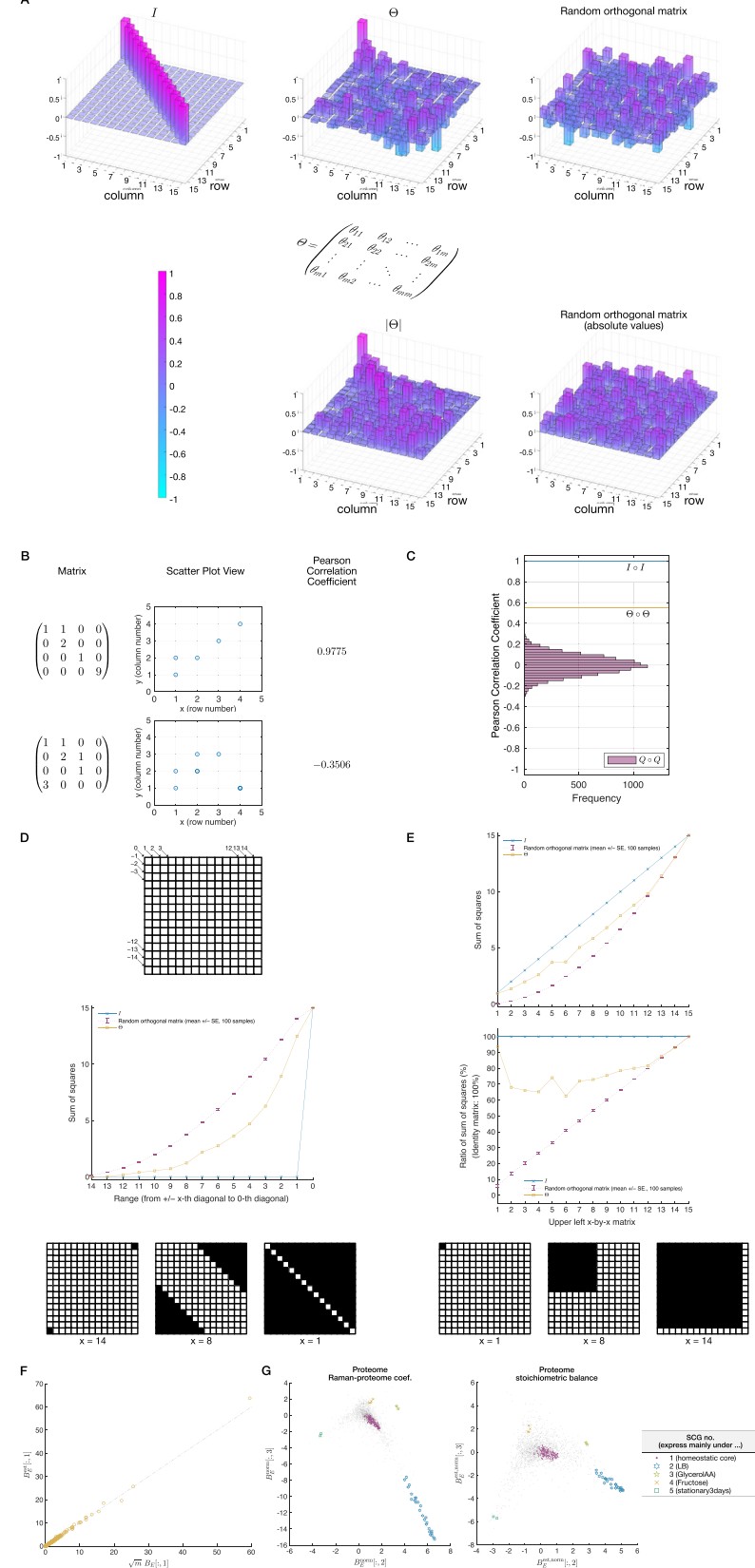

**Appendix 1—figure 9.** Mathematical analyses of the main Raman-proteome data. Related to *Figure 6*. Proteomes of *E. coli* under 15 conditions (*Schmidt et al., 2016*) and corresponding Raman data we measured

*Appendix 1—figure 9 continued on next page*

*Appendix 1—figure 9 continued*

in this study were analyzed in this figure. (**A**) Visual comparison of the unit matrix $I$, the orthogonal matrix $\Theta$ obtained from the data, and a random orthogonal matrix. Height of each bar indicates the value of each element. Colors represent the height of each bar. For clarifying the position of each element, a component form of matrix $\Theta$ is shown in the middle ($m = 15$). For $\Theta$ (middle) and a random orthogonal matrix (right), the original matrices are displayed in the upper row, and matrices whose elements are the absolute values of the corresponding elements of the original matrices are displayed in the lower row. (In this figure, $|\Theta|$ represents a matrix of which the $(i, j)$ element is the absolute value of the $(i, j)$ element of $\Theta$.) (**B**) Representation of matrices as scatterplots. See 'Evaluating similarity between orthogonal matrix $\Theta$ and identity matrix' in Materials and methods for details. (**C**) Comparison of the unit matrix $I$, the orthogonal matrix $\Theta$ obtained from the data, and random orthogonal matrices $Q$ by Pearson correlation coefficients. Pearson correlation coefficient of the element-wise squared matrix of each matrix can be regarded as a measure of closeness to the identity matrix (o represents element-wise multiplication). The probability of finding a random orthogonal matrix $Q$ with Pearson correlation coefficient greater than the Pearson correlation coefficient of $\Theta$ was $< 1 \times 10^{-5}$ (no occurrence in $10^5$ samplings). See 'Evaluating similarity between orthogonal matrix $\Theta$ and identity matrix' in Materials and methods for details. (**D**) Comparison of magnitudes of off-diagonal elements among the unit matrix $I$, the orthogonal matrix $\Theta$ obtained from the data, and random orthogonal matrices $Q$. The lattice on the top explains the numbering of $k$-diagonals ($-m < k < m$, $m = 15$). In the lattices on the bottom, black color indicates areas in which the elements are squared and summed at the corresponding steps (i.e. areas represented by $x$ in the graph). The sum of the squared values in each step is shown in the middle graph. Error bars of the random matrix line are standard errors of 100 samplings. See 'Evaluating similarity between orthogonal matrix $\Theta$ and identity matrix' in Materials and methods for details. (**E**) Comparison of magnitudes of elements of leading principal submatrices among the unit matrix $I$, the orthogonal matrix $\Theta$ obtained from the data, and random orthogonal matrices $Q$. In the lattices on the bottom, black color indicates an area in which elements are squared and summed at the corresponding step (i.e. an area represented by $x$ in the graphs). The sum of the squared values in each area is shown in the top graph. The results shown in the top graph are converted into ratios to the identity matrix $I$ and are shown in the middle graph. Error bars of the random matrix line are standard errors of 100 samplings. See 'Evaluating similarity between orthogonal matrix $\Theta$ and identity matrix' in Materials and methods for details. (**F**) Comparison of $\sqrt{m}\,\mathrm{diag}\left(\boldsymbol{b}_0\right)$ and $\mathrm{diag}\left(\boldsymbol{b}_0^{\mathrm{est}}\right)$. $x$ axis represents $\sqrt{m}\,\boldsymbol{b}_0$ and $y$ axis represents $\boldsymbol{b}_0^{\mathrm{est}}$. The dashed line indicates $y = x$. (**G**) Comparison between $B_E^{\mathrm{norm}}$ (left) and $B_E^{\mathrm{est,norm}}$ (right). Note that while $B_E^{\mathrm{norm}}$ figure (left) is the same as *Figure 6C*, the right figure shows $B_E^{\mathrm{est,norm}} = \left(\sum_{i=1}^{n} d_i\right)^{1/2} \tilde{V}_{\mathrm{rw}}$, where $\tilde{V}_{\mathrm{rw}}$ is shown in *Figure 6D*.

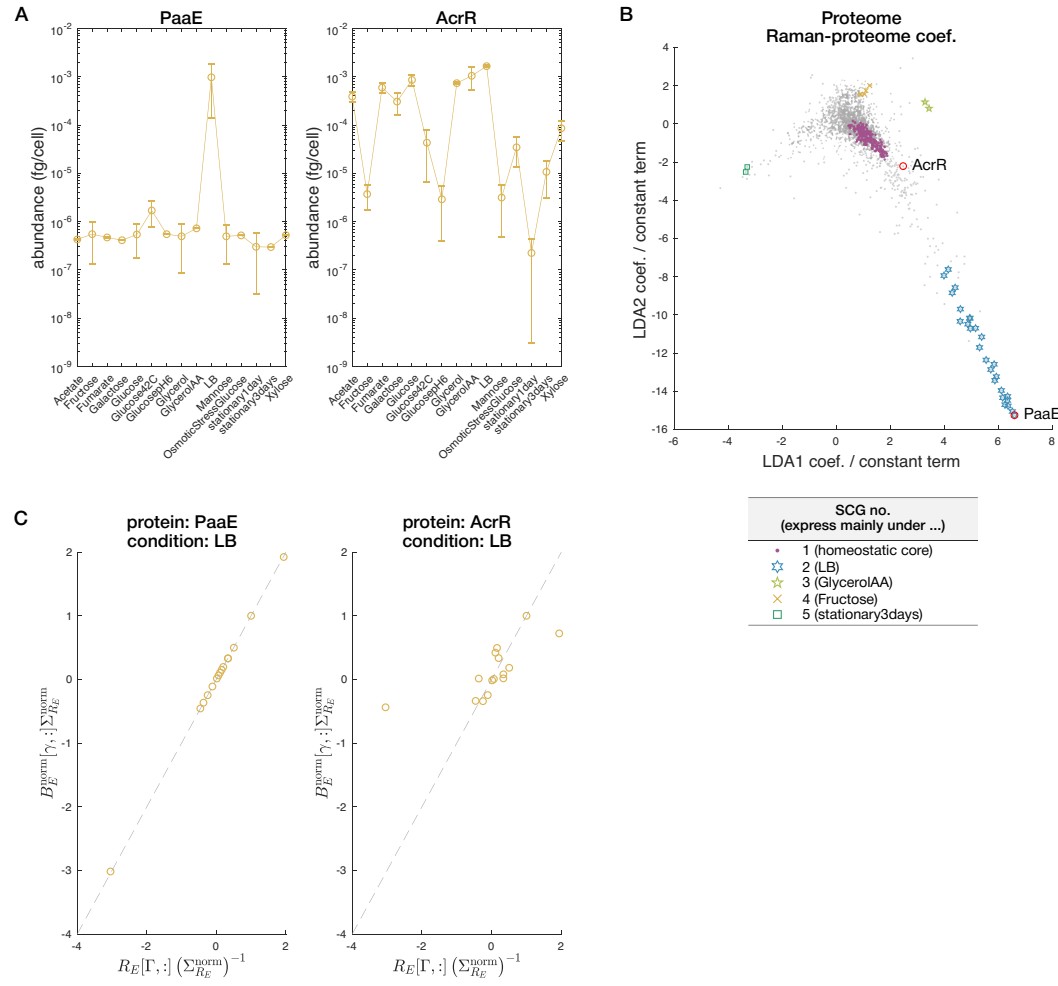

**Appendix 1—figure 10.** Orthant correspondences between Raman spectra in linear discriminant analysis (LDA) space and condition-specific proteins in Raman-proteome coefficient proteome space. Related to *Figure 6*. Using the main Raman and proteome data of *E. coli* under the 15 conditions, we examine the orthant correspondence between Raman spectra in the LDA space and condition-specific proteins in the Raman-proteome coefficient proteome space $\Omega_{\mathbf{B}}$. Here, we focus on two proteins PaaE and AcrR. (**A**) Expression patterns of PaaE (left) and AcrR (right) across conditions. Error bars are standard errors. PaaE is expressed under the 'LB' condition in a condition-specific manner, whereas AcrR is expressed at high levels not only under 'LB' condition but also under several other conditions. (**B**) Positions of PaaE and AcrR in the Raman-proteome coefficient-based proteome space $\Omega_{\mathbf{B}}$. (**C**) Verification of orthant correspondence. We verified the orthant correspondence described by *Equation 2.76*. We multiplied both sides of *Equation 2.76* by $\left(\Sigma_{R_E}^{\mathrm{norm}}\right)^{-1}$, and the elements of the vectors of both sides were compared by scatterplots. The horizontal axes are related to the coordinates in the Raman LDA space; the vertical axes are related to the coordinates in the Raman-proteome coefficient proteome space. The dashed lines are $y = x$. The nearly perfect agreement of the elements confirms the orthant correspondence for the condition-specific protein PaaE (left). Deviations from the diagonal agreement line are found for AcrR (right).

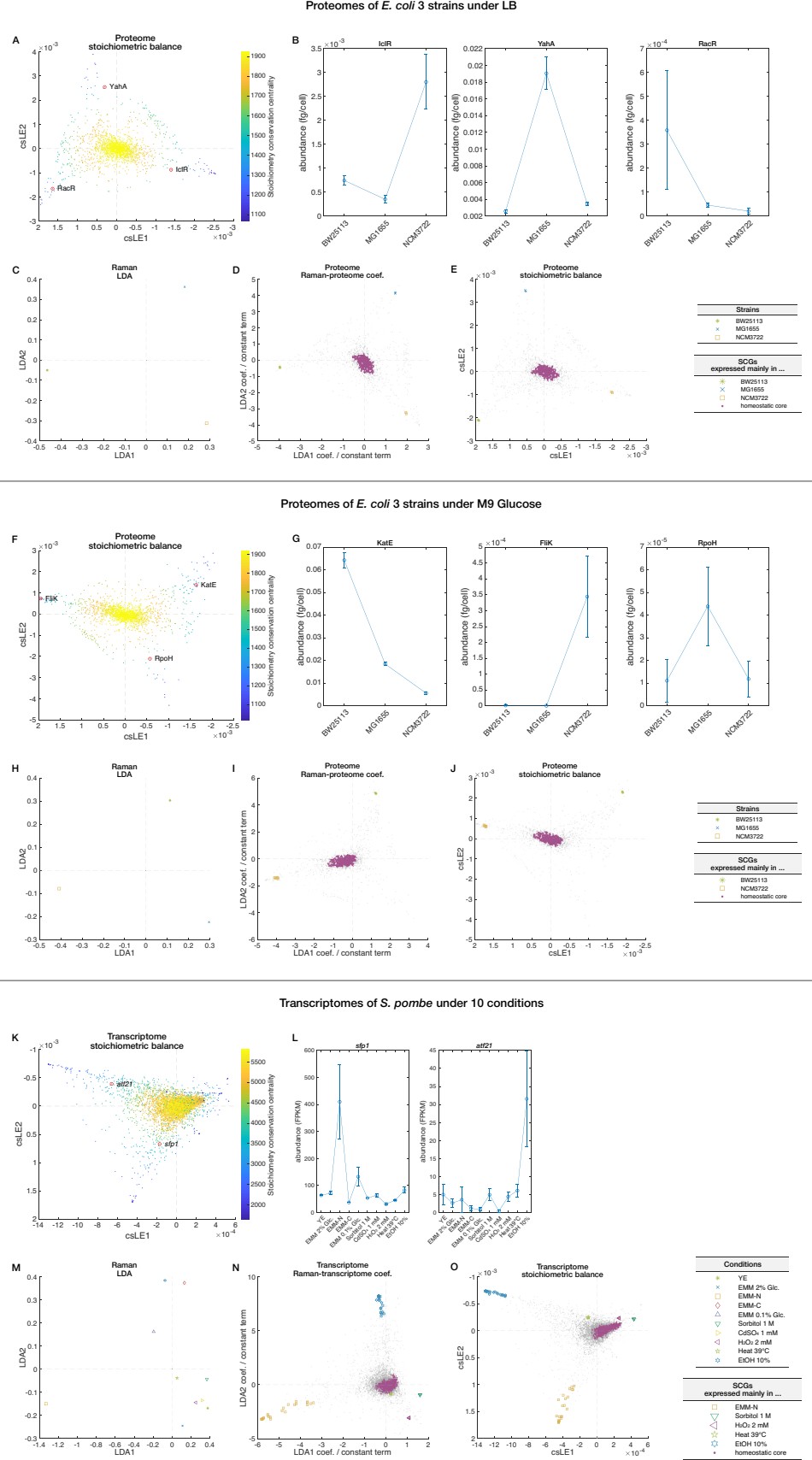

**Appendix 1—figure 11.** Stoichiometry-based omics structures and their correspondences to Raman-based omics structures for additional datasets. Related to *Figures 4–6*. This figure summarizes the results on omics structures characterized by stoichiometry conservation relations and their correspondences to those characterized by Raman-omics relations for additional datasets. (**A–E**) show the results from the analyses of the Raman and proteome data of three *E. coli* strains (BW25113, MG1655, and NCM3722) in LB; (**F–J**) from the analyses of the Raman and proteome data of the three *E. coli* strains in M9 Glucose; and (**K–O**) from the analyses of the Raman and transcriptome data of *S. pombe* under 10 conditions. We used the *E. coli* proteome data reported in *Schmidt et al., 2016*, and the *S. pombe* transcriptome data reported in *Kobayashi-Kirschvink et al., 2018*, in the analyses. (**A**), (**F**), and (**K**) show distributions of omics components in cosine similarity LE (csLE) space. Stoichiometry conservation centrality of each component is indicated by color. (**B**), (**G**), and (**L**) show expression patterns of representative condition-specific omics components indicated in the previous figures of omics structures in the csLE spaces. Error bars are standard errors in (**B**) and (**G**), and maximum-minimum ranges (two replicates) in (**L**). (**C**), (**H**), and (**M**) show positions of averaged cellular Raman spectra under different conditions in the linear discriminant analysis (LDA) spaces. (**D**), (**I**), and (**N**) show omics structures in the spaces specified by the Raman-omics coefficients with the homeostatic cores and condition-specific stoichiometrically conserved groups (SCGs) indicated by colored points. (**E**), (**J**), and (**O**) show the omics structures in the csLE omics spaces with the homeostatic cores and condition-specific SCGs indicated by colored points. Columns $v_{\mathrm{rw},1}$ (the eigenvector corresponding to $L_{\mathrm{rw}}$'s smallest eigenvalue except for zero) and $v_{\mathrm{rw},2}$ (the eigenvector corresponding to $L_{\mathrm{rw}}$'s second smallest eigenvalue except for zero) are shown. We used the cosine similarity thresholds of 0.99993 to specify SCGs both for the three *E. coli* strains under LB data (**D** and **E**) and for the three *E. coli* strains under M9 Glucose data (**I** and **J**), and 0.9967 for the *S. pombe* transcriptome data (**N** and **O**).

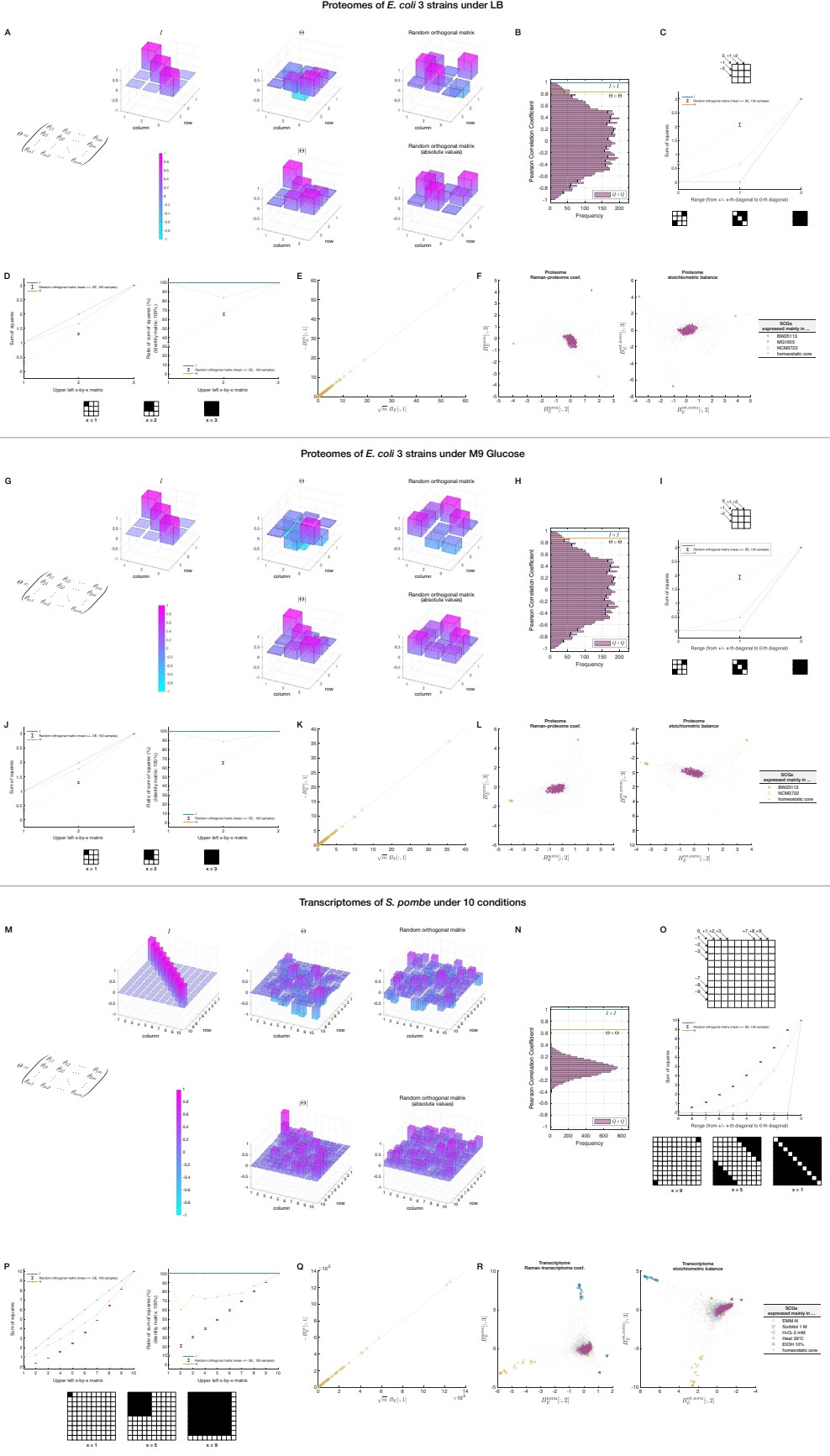

**Appendix 1—figure 12.** Analyses of the mathematical relation connecting two types of omics spaces. Related to *Figure 6*. This figure shows the analyses of mathematical relation that connects coordinates of omics components in the two types of omics spaces (see *Figure 6E* and Section 2 in Appendix) using additional datasets. (**A–F**) show the results from the analyses of the Raman and proteome data of three *E. coli* strains (BW25113, MG1655, and NCM3722) in LB; (**G–L**) from the analyses of the Raman and proteome data of the three *E. coli* strains in M9 Glucose; and (**M–R**) from the analyses of the Raman and transcriptome data of *S. pombe* under 10 conditions. We used the *E. coli* proteome data reported in *Schmidt et al., 2016*, and the *S. pombe* transcriptome data reported in *Kobayashi-Kirschvink et al., 2018* in the analyses. See the caption of *Appendix 1—figure 9* for the explanation of each panel. The stoichiometrically conserved groups (SCGs) in (**F**), (**L**), and (**R**) are the same as in *Appendix 1—figure 11*. The probability of finding a random orthogonal matrix $Q$ with Pearson correlation coefficient greater than the Pearson correlation coefficient of $\Theta$ was 0.022 in (**B**), 0.013 in (**H**), and $< 1 \times 10^{-5}$ (no occurrence in $10^5$ samplings) in (**N**).

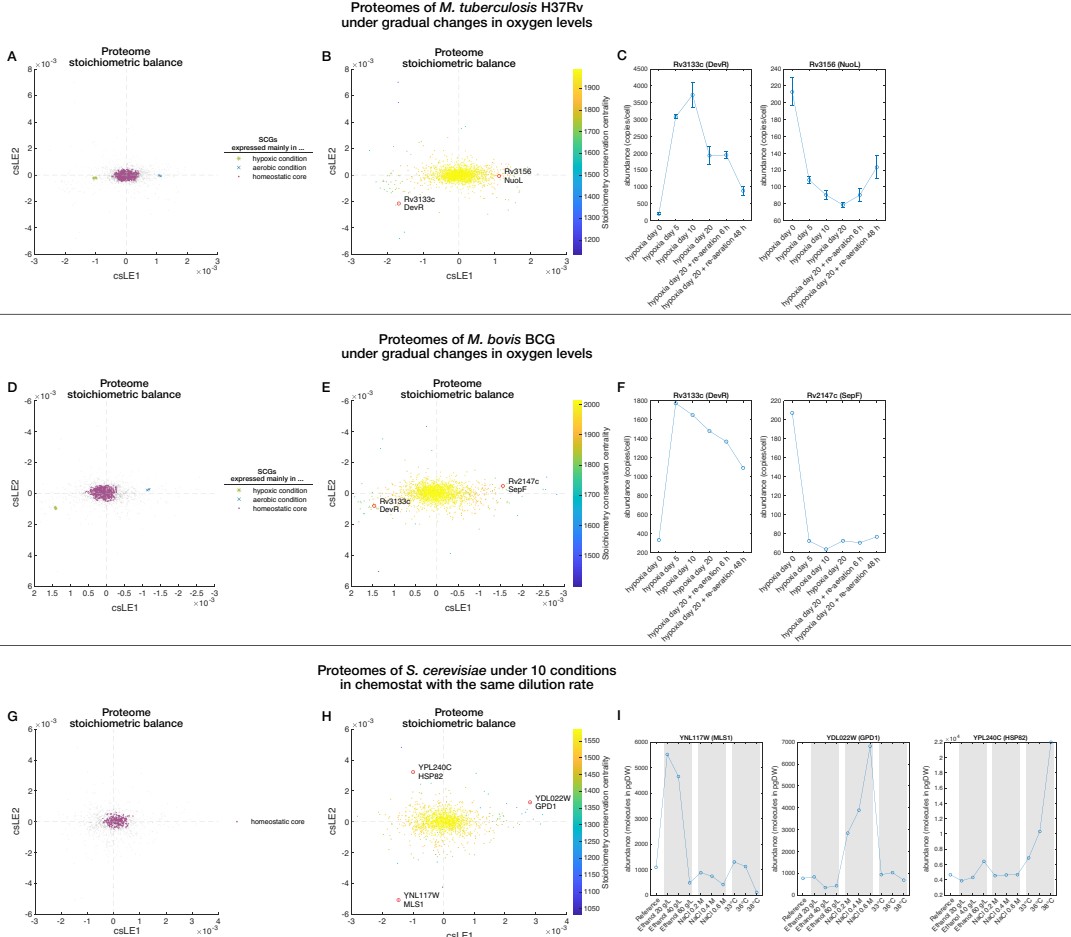

**Appendix 1—figure 13.** Stoichiometry-based proteome structures for additional datasets. Related to *Figures 4 and 5*. This figure shows proteome structures in the cosine similarity LE (csLE) proteome spaces for additional datasets. (**A–C**) show the results from the analyses of the proteome data of *M. tuberculosis* H37Rv under gradual changes in oxygen levels (*Schubert et al., 2015*); (**D–F**) shows the results from the analyses of the proteome data of *M. bovis* BCG under gradual changes in oxygen levels (*Schubert et al., 2015*); and (**G–I**) show the results from the analyses of the proteome data of *S. cerevisiae* under 10 conditions in chemostat with the same dilution rate (*Lahtvee et al., 2017*). (**A**), (**D**), and (**G**) show the proteome structures in the csLE spaces. The thresholds used to specify the stoichiometrically conserved groups (SCGs) were 0.99965 for (**A**), 0.9997 for (**D**), and 0.9989 for (**G**). (**B**), (**E**), and (**H**) show the same proteome structures as in the previous panels, but with stoichiometry conservation centrality of each protein species indicated by the color. (**C**), (**F**), and (**I**) show expression patterns of representative proteins indicated by the red circles in the previous panels. Error bars in (**C**) are standard errors.

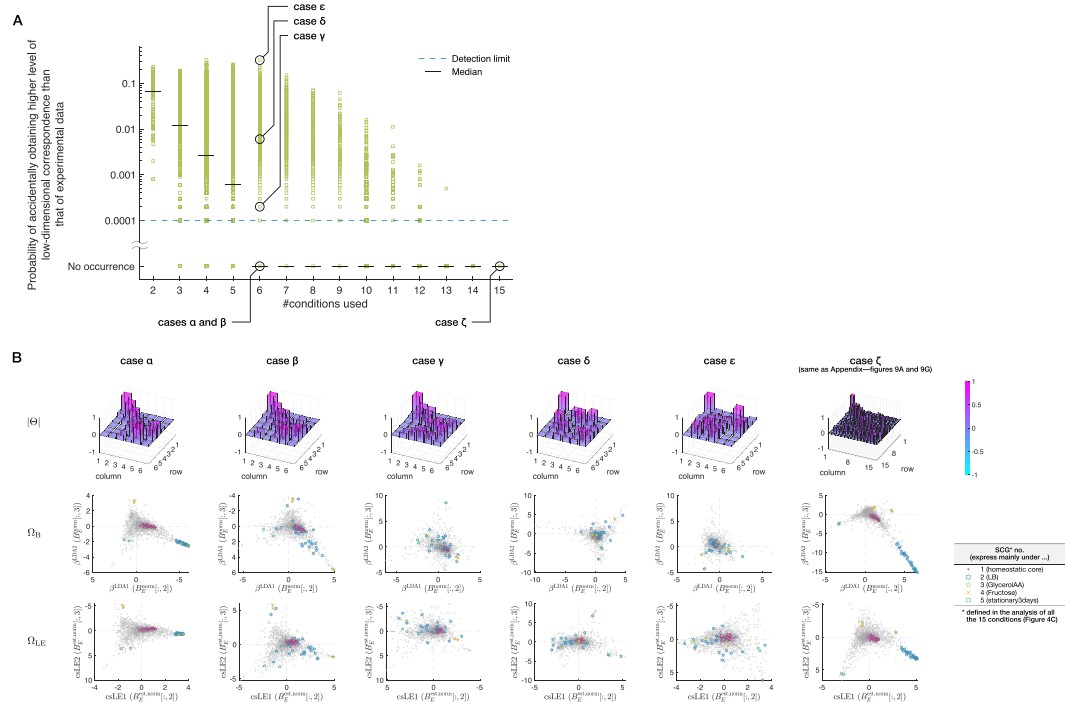

**Appendix 1—figure 14.** Dependence of low-dimensional correspondence between Raman spectra and proteomes on the number of conditions. Related to *Figure 6*. The dependence of the low-dimensional correspondence between Raman spectra and proteomes on the number of analyzed conditions was systematically investigated by evaluating the similarity of the orthogonal matrix Θ to the identity matrix for all subsampled condition sets. Proteomes of *E. coli* under 15 conditions (*Schmidt et al., 2016*) and corresponding Raman data we measured in this study were analyzed in this figure. (**A**) The relationship between the number of conditions and the probability of obtaining higher level of low-dimensional correspondence than that of experimental data by chance. This probability is calculated as the probability of finding a random orthogonal matrix with Pearson correlation coefficient greater than the Pearson correlation coefficient of Θ by creating $10^4$ random orthogonal matrices. See 'Evaluating similarity between orthogonal matrix Θ and identity matrix' in Materials and methods and *Appendix 1—figure 9* for details of the evaluation method. Each green square corresponds to one subsample, and each short horizontal black line represents the median of all the $\binom{15}{x}$ combinations of conditions (i.e. $\binom{15}{x}$ green squares) for each subsample size $x$. The blue dashed line indicates the detection limit (i.e. one over the number of generated random orthogonal matrices). The non-subsampled case (i.e. the case with all 15 conditions) in this figure corresponds to *Appendix 1—figure 9C*. (**B**) Visual comparison of Θ, $B_E^{\mathrm{norm}}$ and $B_E^{\mathrm{est,norm}}$ for six representative subsamples indicated in (**A**). As in *Appendix 1—figure 9A*, Θ is visualized using |Θ|, whose element is the absolute value of the corresponding element of Θ, and height of each bar in the figures of |Θ| indicates the value of each element of |Θ|. Colors reflect the height of each bar. Spaces created with columns of $B_E^{\mathrm{norm}}$ and $B_E^{\mathrm{est,norm}}$ are $\Omega_{\mathrm{B}}$ and $\Omega_{\mathrm{LE}}$, respectively. As Θ deviates from the identity matrix from the cases $\alpha$ and $\beta$ to the case of $\epsilon$, the low-dimensional correspondence between $\Omega_{\mathrm{B}}$ and $\Omega_{\mathrm{LE}}$ collapses naturally. Since the case $\zeta$ is the non-subsampled case, the figure of |Θ| is the same as *Appendix 1—figure 9A*, and those of $B_E^{\mathrm{norm}}$ and $B_E^{\mathrm{est,norm}}$ are the same as *Appendix 1—figure 9G*. Note that the figure of $\Omega_{\mathrm{B}}$ of the case $\zeta$ is also exactly the same as *Figure 6C*, and that of $\Omega_{\mathrm{LE}}$ of the case $\zeta$ is equal to *Figure 6D* up to a factor of $\left(\sum_{i=1}^{n} d_i\right)^{1/2}$. The stoichiometrically conserved groups (SCGs) shown in this figure were defined in the analysis of the proteomes of all the 15 conditions (*Figure 4C*).

**Appendix 1—table 1.** List of culture conditions.

M9 m.m. and a.a. in this table are the abbreviations for M9 minimal media and amino acids, respectively.

| Phase | Overview of composition | Temperature | pH | Name in this paper |
|---|---|---|---|---|
| | M9 m.m. + acetate | | | Acetate |
| | M9 m.m. + fructose | | | Fructose |
| | M9 m.m. + fumarate | 37°C | | Fumarate |
| | M9 m.m. + galactose | | | Galactose |
| | M9 m.m. + glucose | | 7 | Glucose |
| | M9 m.m. + glucose | 42°C | | Glucose42C |
| Exponential | M9 m.m. + glucose | | 6 | GlucosepH6 |
| | M9 m.m. + glycerol | | | Glycerol |
| | M9 m.m. + glycerol + a.a. | | | GlycerolAA |
| | M9 m.m. + glucose + NaCl | | | OsmoticStressGlucose |
| | M9 m.m. + mannose | 37°C | 7 | Mannose |
| | M9 m.m. + xylose | | | Xylose |
| | LB | | | LB |
| Stationary for 1 day | M9 m.m. + glucose | | | stationary1day |
| Stationary for 3 days | | | | stationary3days |

**Appendix 1—table 2.** Evaluation of the overall estimation error with various distance measures (the case where LDA1 to LDA4 axes were used).

The sum of estimation errors $\sum_i \mathrm{dist}(\hat{p}_i, \hat{p}_i^{\mathrm{est}})$ was calculated, and a permutation test ($10^5$ permutations) was conducted. In this table, LDA1 to LDA4 axes were used. $\bar{x}$ represents a vector whose all elements are the mean of all elements of $x$. $x_j$ is the $j$-th element of $x$. $\mathrm{median}_j\, x_j$ represents the median of scalers $x_j$.

| Metric | Definition of $\mathbf{dist}(x, y)$ | $\sum_i \mathbf{dist}\left(\hat{p}_i, \hat{p}_i^{\mathbf{est}}\right)$ | $p$-value |
|---|---|---|---|
| Square of $L^2$ norm (PRESS) | $\|x - y\|_2^2 = \sum_j (x - y)_j^2$ | $2.34 \times 10^3$ | 0.00005 |
| $L^1$ norm | $\|x - y\|_1 = \sum_j \left|(x - y)_j\right|$ | $1.40 \times 10^3$ | 0.00002 |
| Cosine distance | $1 - \dfrac{x \cdot y}{\|x\|_2 \|y\|_2}$ | 1.52 | 0.0014 |
| 1 – Pearson correlation coefficient | $1 - \dfrac{(x - \bar{x}) \cdot (y - \bar{y})}{\|x - \bar{x}\|_2 \|y - \bar{y}\|_2}$ | 1.57 | 0.0012 |
| Median of relative error | $\mathrm{median}_j \dfrac{\left|(x - y)_j\right|}{x_j + 1}$ | 0.0536 | 0.00022 |

**Appendix 1—table 3.** Evaluation of the overall estimation error with various distance measures (the case where all the 14 LDA axes were used).

The results obtained by using all the 14 LDA axes are presented. See *Appendix 1—table 2* for notations. Note that the system is underdetermined in this case; thus, we adopted the minimum-norm solution from among all least-squares solutions.

| Metric | Definition of $\mathbf{dist}(x, y)$ | $\sum_i \mathbf{dist}\left(\hat{p}_i, \hat{p}_i^{\mathbf{est}}\right)$ | $p$-value |
|---|---|---|---|
| Square of $L^2$ norm (PRESS) | $\|x - y\|_2^2 = \sum_j (x - y)_j^2$ | $1.63 \times 10^3$ | 0.0019 |

*Appendix 1—table 3 Continued on next page*

*Appendix 1—table 3 Continued*

| Metric | Definition of $\mathbf{dist}(x,y)$ | $\sum_i \mathbf{dist}\left(\hat{p}_i, \hat{p}_i^{\text{est}}\right)$ | $p$-value |
|---|---|---|---|
| $L^1$ norm | $\|x - y\|_1 = \sum_j \left|(x - y)_j\right|$ | $1.19 \times 10^3$ | 0.00066 |
| Cosine distance | $1 - \frac{x \cdot y}{\|x\|_2 \|y\|_2}$ | 1.18 | 0.0879 |
| 1 – Pearson correlation coefficient | $1 - \frac{(x - \bar{x}) \cdot (y - \bar{y})}{\|x - \bar{x}\|_2 \|y - \bar{y}\|_2}$ | 1.23 | 0.085 |
| Median of relative error | $\text{median}_j \frac{\left|(x - y)_j\right|}{x_j + 1}$ | 0.0418 | 0.00082 |

**Appendix 1—table 4.** Gene list of SCG 1 (homeostatic core).
Members of homeostatic core (*Figure 4*, cosine similarity threshold: 0.995). The description of each gene is cited from *Schmidt et al., 2016*.

| Name | Description |
|---|---|
| rpoC | DNA-directed RNA polymerase subunit beta' |
| rpoB | DNA-directed RNA polymerase subunit beta |
| tufA | Elongation factor Tu 1 |
| infB | Translation initiation factor IF-2 |
| fusA | Elongation factor G |
| glyS | Glycyl-tRNA synthetase beta subunit |
| rpsA | 30S ribosomal protein S1 |
| leuS | Leucyl-tRNA synthetase |
| pheT | Phenylalanyl-tRNA synthetase beta chain |
| aspS | Aspartyl-tRNA synthetase |
| valS | Valyl-tRNA synthetase |
| secA | Protein translocase subunit SecA |
| gyrA | DNA gyrase subunit A |
| pepN | Aminopeptidase N |
| tsf | Elongation factor Ts |
| tig | Trigger factor |
| pta | Phosphate acetyltransferase |
| bamA | Outer membrane protein assembly factor YaeT |
| rne | Ribonuclease E |
| ftsZ | Cell division protein FtsZ |
| gyrB | DNA gyrase subunit B |
| polA | DNA polymerase I |
| rplB | 50S ribosomal protein L2 |
| prlC | Oligopeptidase A |
| rho | Transcription termination factor Rho |
| ftsH | ATP-dependent zinc metalloprotease FtsH |
| nusA | Transcription elongation protein NusA |
| lysS | Lysyl-tRNA synthetase |

*Appendix 1—table 4 Continued on next page*

*Appendix 1—table 4 Continued*

| Name | Description |
| --- | --- |
| metG | Methionyl-tRNA synthetase |
| glnS | Glutaminyl-tRNA synthetase |
| lpdA | Dihydrolipoyl dehydrogenase |
| serS | Seryl-tRNA synthetase |
| surA | Chaperone SurA |
| rpsB | 30S ribosomal protein S2 |
| gltX | Glutamyl-tRNA synthetase |
| lptD | LPS-assembly protein LptD |
| argS | Arginyl-tRNA synthetase |
| fabB | 3-Oxoacyl-[acyl-carrier-protein] synthase 1 |
| pheS | Phenylalanyl-tRNA synthetase alpha chain |
| clpX | ATP-dependent Clp protease ATP-binding subunit ClpX |
| accC | Biotin carboxylase |
| pyrG | CTP synthase |
| tolC | Outer membrane protein TolC |
| rplE | 50S ribosomal protein L5 |
| accA | Acetyl-coenzyme A carboxylase carboxyl transferase subunit alpha |
| hflK | Modulator of FtsH protease HflK |
| pdxB | Erythronate-4-phosphate dehydrogenase |
| ygfZ | tRNA-modifying protein YgfZ |
| pmbA | Protein PmbA |
| rplA | 50S ribosomal protein L1 |
| hldD | ADP-L-glycero-D-manno-heptose-6-epimerase |
| mreB | Rod shape-determining protein MreB |
| acrA | Acriflavine resistance protein A |
| gor | Glutathione reductase |
| hisS | Histidyl-tRNA synthetase |
| rpsC | 30S ribosomal protein S3 |
| glmM | Phosphoglucosamine mutase |
| lepA | Elongation factor 4 |
| ffh | Signal recognition particle protein |
| secD | Protein-export membrane protein SecD |
| lpoA | Penicillin-binding protein activator LpoA |
| rhlB | ATP-dependent RNA helicase RhlB |
| rpsG | 30S ribosomal protein S7 |
| rpsD | 30S ribosomal protein S4 |
| minD | Septum site-determining protein MinD |
| cyoA | Ubiquinol oxidase subunit 2 |

*Appendix 1—table 4 Continued on next page*

*Appendix 1—table 4 Continued*

| Name | Description |
| --- | --- |
| mdoG | Glucans biosynthesis protein G |
| rplC | 50S ribosomal protein L3 |
| glmU | Bifunctional protein GlmU |
| rpsF | 30S ribosomal protein S6 |
| rpsE | 30S ribosomal protein S5 |
| hemL | Glutamate-1-semialdehyde 2,1-aminomutase |
| hldE | Bifunctional protein HldE |
| ubiE | Ubiquinone/menaquinone biosynthesis methyltransferase UbiE |
| sspA | Stringent starvation protein A |
| nusG | Transcription antitermination protein NusG |
| prfB | Peptide chain release factor 2 |
| dacA | D-alanyl-D-alanine carboxypeptidase DacA |
| rplF | 50S ribosomal protein L6 |
| fabG | 3-Oxoacyl-[acyl-carrier-protein] reductase |
| ftsY | Cell division protein FtsY |
| dcrB | Protein DcrB |
| mlaC | Probable phospholipid-binding protein MlaC |
| hflC | Modulator of FtsH protease HflC |
| coaB | Coenzyme A biosynthesis bifunctional protein CoaBC |
| ybiT | Uncharacterized ABC transporter ATP-binding protein YbiT |
| oxyR | Hydrogen peroxide-inducible genes activator |
| rpsH | 30S ribosomal protein S8 |
| fkpA | FKBP-type peptidyl-prolyl cis-trans isomerase FkpA |
| frr | Ribosome-recycling factor |
| fabD | Malonyl CoA-acyl carrier protein transacylase |
| hslO | 33 kDa chaperonin |
| ybeZ | PhoH-like protein |
| hemX | Putative uroporphyrinogen-III C-methyltransferase |
| rplY | 50S ribosomal protein L25 |
| rplK | 50S ribosomal protein L11 |
| rpsI | 30S ribosomal protein S9 |
| bamB | Lipoprotein YfgL |
| bamD | UPF0169 lipoprotein YfiO |
| kdgR | Transcriptional regulator KdgR |
| glnD | [Protein-PII] uridylyltransferase |
| yniC | Phosphatase YniC |
| rpsJ | 30S ribosomal protein S10 |
| rplX | 50S ribosomal protein L24 |

*Appendix 1—table 4 Continued*

| Name | Description |
|------|-------------|
| rplD | 50S ribosomal protein L4 |
| rplQ | 50S ribosomal protein L17 |
| ppa | Inorganic pyrophosphatase |
| rpsM | 30S ribosomal protein S13 |
| rplN | 50S ribosomal protein L14 |
| ybaB | UPF0133 protein YbaB |
| yidC | Inner membrane protein OxaA |
| lptB | Lipopolysaccharide export system ATP-binding protein LptB |
| suhB | Inositol-1-monophosphatase |
| yejK | Nucleoid-associated protein YejK |
| ghrA | Glyoxylate/hydroxypyruvate reductase A |
| rsmI | Ribosomal RNA small subunit methyltransferase I |
| hemY | Protein HemY |
| uup | ABC transporter ATP-binding protein Uup |
| hrpA | ATP-dependent RNA helicase HrpA |
| rplJ | 50S ribosomal protein L10 |
| rplM | 50S ribosomal protein L13 |
| fur | Ferric uptake regulation protein |
| rplS | 50S ribosomal protein L19 |
| rcsB | Capsular synthesis regulator component B |
| mrp | Protein Mrp |
| glyQ | Glycyl-tRNA synthetase alpha subunit |
| greA | Transcription elongation factor GreA |
| nrdB | Ribonucleoside-diphosphate reductase 1 subunit beta |
| wbbI | Uncharacterized protein YefG |
| udk | Uridine kinase |
| mnmG | tRNA uridine 5-carboxymethylaminomethyl modification enzyme MnmG |
| rplL | 50S ribosomal protein L7/L12 |
| rplI | 50S ribosomal protein L9 |
| rpoZ | DNA-directed RNA polymerase subunit omega |
| ybbN | Uncharacterized protein YbbN |
| yfiF | Uncharacterized tRNA/rRNA methyltransferase YfiF |
| yedD | Uncharacterized lipoprotein YedD |
| rpmD | 50S ribosomal protein L30 |
| tatB | Sec-independent protein translocase protein TatB |
| yfgM | UPF0070 protein YfgM |
| kdsB | 3-Deoxy-manno-octulosonate cytidylyltransferase |
| rpoN | RNA polymerase sigma-54 factor |

*Appendix 1—table 4 Continued*

| Name | Description |
| --- | --- |
| *fdx* | 2Fe-2S ferredoxin |
| *rplV* | 50S ribosomal protein L22 |
| *rplO* | 50S ribosomal protein L15 |
| *fabZ* | (3R)-hydroxymyristoyl-[acyl-carrier-protein] dehydratase |
| *mipA* | MltA-interacting protein |
| *ssb* | Single-stranded DNA-binding protein |
| *yiaF* | Uncharacterized protein YiaF |
| *secY* | Preprotein translocase subunit SecY |
| *rbfA* | Ribosome-binding factor A |
| *potA* | Spermidine/putrescine import ATP-binding protein PotA |
| *rimM* | Ribosome maturation factor RimM |
| *trxA* | Thioredoxin-1 |
| *rpsS* | 30S ribosomal protein S19 |
| *rpsU* | 30S ribosomal protein S21 |
| *accB* | Biotin carboxyl carrier protein of acetyl-CoA carboxylase |
| *engB* | Probable GTP-binding protein EngB |
| *tatA* | Sec-independent protein translocase protein TatA |
| *rfbD* | dTDP-4-dehydrorhamnose reductase |
| *ribF* | Riboflavin biosynthesis protein RibF |
| *folP* | Dihydropteroate synthase |
| *lepB* | Signal peptidase I |
| *sspB* | Stringent starvation protein B |
| *hupA* | DNA-binding protein HU-alpha |
| *rpsP* | 30S ribosomal protein S16 |
| *rplP* | 50S ribosomal protein L16 |
| *rpsT* | 30S ribosomal protein S20 |
| *rpsK* | 30S ribosomal protein S11 |
| *rplU* | 50S ribosomal protein L21 |
| *rplR* | 50S ribosomal protein L18 |
| *lpxA* | Acyl-[acyl-carrier-protein]–UDP-N-acetylglucosamine O-acyltransferase |
| *yceD* | Uncharacterized protein YceD |
| *queC* | 7-Cyano-7-deazaguanine synthase |
| *rpmA* | 50S ribosomal protein L27 |
| *rpmG* | 50S ribosomal protein L33 |
| *rpmF* | 50S ribosomal protein L32 |
| *rpsN* | 30S ribosomal protein S14 |
| *rplT* | 50S ribosomal protein L20 |
| *nudK* | GDP-mannose pyrophosphatase NudK |

*Appendix 1—table 4 Continued on next page*

*Appendix 1—table 4 Continued*

| Name | Description |
| --- | --- |
| rplW | 50S ribosomal protein L23 |
| trmB | tRNA (guanine-N(7)-)-methyltransferase |
| rluB | Ribosomal large subunit pseudouridine synthase B |
| rpsR | 30S ribosomal protein S18 |
| secG | Protein-export membrane protein SecG |
| rlmE | Ribosomal RNA large subunit methyltransferase E |
| yfaY | CinA-like protein |
| trmA | tRNA (uracil-5-)-methyltransferase |
| rpmH | 50S ribosomal protein L34 |
| yajC | UPF0092 membrane protein YajC |
| yheU | UPF0270 protein YheU |

**Appendix 1—table 5.** Gene list of SCG 2.
Members in SCG 2 (*Figure 4*, cosine similarity threshold: 0.995). The description of each gene is cited from *Schmidt et al., 2016*.

| Name | Description |
| --- | --- |
| fdoG | Formate dehydrogenase-O major subunit |
| dsdA | D-serine dehydratase |
| treC | Trehalose-6-phosphate hydrolase |
| sdaB | L-serine dehydratase 2 |
| nanA | N-acetylneuraminate lyase |
| garD | D-galactarate dehydratase |
| proV | Glycine betaine/L-proline transport ATP-binding protein ProV |
| garR | 2-Hydroxy-3-oxopropionate reductase |
| nanK | N-acetylmannosamine kinase |
| fdoH | Formate dehydrogenase-O iron-sulfur subunit |
| aphA | Class B acid phosphatase |
| nanE | Putative N-acetylmannosamine-6-phosphate 2-epimerase |
| srlB | Glucitol/sorbitol-specific phosphotransferase enzyme IIA component |
| ibpB | Small heat shock protein IbpB |
| hybC | Hydrogenase-2 large chain |
| proW | Glycine betaine/L-proline transport system permease protein ProW |
| srlE | Glucitol/sorbitol-specific phosphotransferase enzyme IIB component |
| fdoI | Formate dehydrogenase, cytochrome b556(fdo) subunit |
| preT | Uncharacterized oxidoreductase YeiT |
| garL | 5-Keto-4-deoxy-D-glucarate aldolase |
| paaB | Phenylacetic acid degradation protein PaaB |
| paaK | Phenylacetate-coenzyme A ligase |
| paaE | Probable phenylacetic acid degradation NADH oxidoreductase PaaE |
| ykgE | Uncharacterized protein YkgE |

*Appendix 1—table 5 Continued on next page*

*Appendix 1—table 5 Continued*

| Name | Description |
| --- | --- |
| ybjT | Uncharacterized protein YbjT |
| ykgG | Uncharacterized protein YkgG |

**Appendix 1—table 6.** Gene list of SCG 3.
Members in SCG 3 (*Figure 4*, cosine similarity threshold: 0.995). The description of each gene is cited from *Schmidt et al., 2016*.

| Name | Description |
| --- | --- |
| wzc | Tyrosine-protein kinase Wzc |
| amiC | N-acetylmuramoyl-L-alanine amidase AmiC |

**Appendix 1—table 7.** Gene list of SCG 4.
Members in SCG 4 (*Figure 4*, cosine similarity threshold: 0.995). The description of each gene is cited from *Schmidt et al., 2016*.

| Name | Description |
| --- | --- |
| fruB | Multiphosphoryl transfer protein |
| fruK | 1-Phosphofructokinase |
| fruA | PTS system fructose-specific EIIBC component |
| narI | Respiratory nitrate reductase 1 gamma chain |

**Appendix 1—table 8.** Gene list of SCG 5.
Members in SCG 5 (*Figure 4*, cosine similarity threshold: 0.995). The description of each gene is cited from *Schmidt et al., 2016*.

| Name | Description |
| --- | --- |
| hdeB | Protein HdeB |
| hdeA | Chaperone-like protein HdeA |

**Appendix 1—table 9.** Interpretations of $r_h, \hat{r}_i, b_h,$ and $\hat{b}_j$.
Interpretations of the columns and rows of $R_E$ and $B_E$ are summarized.

| Matrix | Vector | | Dimension | Description |
| --- | --- | --- | --- | --- |
| $R_E$ | Column | $r_h \left(h = 0, \ldots, m-1\right)$ | $m$ | List of $h$-th LDA coordinates of mean LDA Raman of all the conditions |
| | Row | $\hat{r}_i \left(i = 1, \ldots, m\right)$ | $m$ | Mean LDA Raman of condition $i$ |
| $B_E$ | Column | $b_h \left(h = 0, \ldots, m-1\right)$ | $n$ | List of coefficients of all the proteins for the $h$-th LDA axis |
| | Row | $\hat{b}_j \left(j = 1, \ldots, n\right)$ | $m$ | Coefficients for protein $j$ |

**Appendix 1—table 10.** Mathematical relation between Raman-proteome coefficients and cosine similarity LE (csLE) proteomes.
The matrices in the left-hand side of *Equation 2.138* (a proteome structure based on Raman-proteome coefficients) and their counterparts in the right-hand side of *Equation 2.138* (a proteome structure obtained with csLE) are listed.

| Raman-omics coef. structure | csLE | Size and type of matrix | Description |
| --- | --- | --- | --- |
| $B_E^{\text{norm}}$ | $\left(\sum_{i=1}^n d_i\right)^{1/2} \tilde{V}_{\text{rw}}$ $\left(= B_E^{\text{est,norm}}\right)$ | $n \times m$ matrix | Coefficients normalized by constants |

*Appendix 1—table 10 Continued on next page*

*Appendix 1—table 10 Continued*

| Raman-omics coef. structure | csLE | Size and type of matrix | Description |
|---|---|---|---|
| $I$ | $\Theta$ | $m \times m$ orthogonal matrix | Orthogonal transformation |
| $m^{-1/2}\mathrm{diag}\left(\left(\mathbf{1}_m\right)^\top P\right)$ $\left(= m^{1/2}\mathrm{diag}\left(\boldsymbol{b}_0\right)\right)$ | $\left(\sum_{i=1}^n d_i\right)^{-1/2}\mathrm{diag}\left(P^\top P\right)^{1/2} D$ $\left(= \mathrm{diag}\left(\boldsymbol{b}_0^{\mathrm{est}}\right)\right)$ | $n \times n$ diagonal matrix | Constant terms |
| $\Sigma_{R_E}^{\mathrm{norm}}$ | $\Sigma_{\mathrm{LE}}$ | $m \times m$ diagonal matrix | Singular values |

