## [Editor Report · eLife Assessment]

This paper reports the **fundamental** finding of how Raman spectral patterns correlate with proteome profiles using Raman spectra of *E. coli* cells from different physiological conditions and found global stoichiometric regulation on proteomes. The authors' findings provide **compelling** evidence that stoichiometric regulation of proteomes is general through analysis of both bacterial and human cells. In the future, similar methodology can be applied on various tissue types and microbial species for studying proteome composition with Raman spectral patterns.

---

## [Referee Report · Reviewer #1 (Public review)]

Summary

This work performed Raman spectral microscopy for *E. coli* cells with 15 different culture conditions. The author developed a theoretical framework to construct a regression matrix which predicts proteome composition by Raman data. Specifically, this regression matrix is obtained by statistical inference from various experimental conditions. With this model, the authors categorized co-expressed genes and illustrate how proteome stoichiometry is regulated among different culture conditions. Co-expressed gene clusters were investigated and identified as homeostasis core, carbon-source dependent, and stationary phase dependent genes. Overall, the author demonstrates a strong and comprehensive data analysis scheme for the joint analysis of Raman and proteome datasets.

Strengths and major contributions

Major contributions: (1) Experimentally, the authors contributed Raman datasets of *E. coli* with various growth conditions. (2) In data analysis, the authors developed a scheme to compare proteome and Raman datasets. Protein co-expression clusters were identified, and their biological meaning were investigated.

Discussion and impact for the field

Raman signature contains both proteomic and metabolomic information and is an orthogonal method to infer the composition biomolecules. This work is a strong initiative for introducing the powerful technique to systems biology and provide a rigorous pipeline for future data analysis. The regression matrix can be used for cross-comparison among future experimental results on proteome-Raman datasets.

Comments on revisions:

The authors addressed all my questions nicely. In particular, the subsampling test demonstrated that with enough "distinct" physiological condition (even for m=5) one could already explore the major mode of proteome regulation and Raman signature. The main text has been streamlined and the clarity is improved. I have a minor suggestion:

(i) For equation (1), it is important to emphasize that the formula works for every j=1,...,15, and the regression matrix B is obtained by statistical inference by summarizing data from all 15 conditions.

---

## [Author Response]

The following is the authors’ response to the original reviews.

**Reviewer #1 (Public review):**
SummaryThis work performed Raman spectral microscopy at the single-cell level for 15 different culture conditions in *E. coli*. The Raman signature is systematically analyzed and compared with the proteome dataset of the same culture conditions. With a linear model, the authors revealed correspondence between Raman pattern and proteome expression stoichiometry indicating that spectrometry could be used for inferring proteome composition in the future. With both Raman spectra and proteome datasets, the authors categorized co-expressed genes and illustrated how proteome stoichiometry is regulated among different culture conditions. Co-expressed gene clusters were investigated and identified as homeostasis core, carbon-source dependent, and stationary phase-dependent genes. Overall, the authors demonstrate a strong and solid data analysis scheme for the joint analysis of Raman and proteome datasets.Strengths and major contributions(1) Experimentally, the authors contributed Raman datasets of *E. coli* with various growth conditions.(2) In data analysis, the authors developed a scheme to compare proteome and Raman datasets. Protein co-expression clusters were identified, and their biological meaning was investigated.WeaknessesThe experimental measurements of Raman microscopy were conducted at the single-cell level; however, the analysis was performed by averaging across the cells. The author did not discuss if Raman microscopy can used to detect cell-to-cell variability under the same condition.

We thank the reviewer for raising this important point. Though this topic is beyond the scope of our study, some of our authors have addressed the application of single-cell Raman spectroscopy to characterizing phenotypic heterogeneity in individual *Staphylococcus aureus* cells in another paper (Kamei et al., bioRxiv, doi: 10.1101/2024.05.12.593718). Additionally, one of our authors demonstrated that single-cell RNA sequencing profiles can be inferred from Raman images of mouse cells (Kobayashi-Kirschvink et al., Nat. Biotechnol. 42, 1726–1734, 2024). Therefore, detecting cell-to-cell variability under the same conditions has been shown to be feasible. Whether averaging single-cell Raman spectra is necessary depends on the type of analysis and the available dataset. We will discuss this in more detail in our response to Comment (1) by Reviewer #1 (Recommendation for the authors).

Discussion and impact on the fieldRaman signature contains both proteomic and metabolomic information and is an orthogonal method to infer the composition of biomolecules. It has the advantage that single-cell level data could be acquired and both in vivo and in vitro data can be compared. This work is a strong initiative for introducing the powerful technique to systems biology and providing a rigorous pipeline for future data analysis.
**Reviewer #2 (Public review):**
Summary and strengths:Kamei et al. observe the Raman spectra of a population of single *E. coli* cells in diverse growth conditions. Using LDA, Raman spectra for the different growth conditions are separated. Using previously available protein abundance data for these conditions, a linear mapping from Raman spectra in LDA space to protein abundance is derived. Notably, this linear map is condition-independent and is consequently shown to be predictive for held-out growth conditions. This is a significant result and in my understanding extends the earlier Raman to RNA connection that has been reported earlier.They further show that this linear map reveals something akin to bacterial growth laws (ala Scott/Hwa) that the certain collection of proteins shows stoichiometric conservation, i.e. the group (called SCG - stoichiometrically conserved group) maintains their stoichiometry across conditions while the overall scale depends on the conditions. Analyzing the changes in protein mass and Raman spectra under these conditions, the abundance ratios of information processing proteins (one of the large groups where many proteins belong to "information and storage" - ISP that is also identified as a cluster of orthologous proteins) remain constant. The mass of these proteins deemed, the homeostatic core, increases linearly with growth rate. Other SCGs and other proteins are condition-specific.Notably, beyond the ISP COG the other SCGs were identified directly using the proteome data. Taking the analysis beyond they then how the centrality of a protein - roughly measured as how many proteins it is stoichiometric with - relates to function and evolutionary conservation. Again significant results, but I am not sure if these ideas have been reported earlier, for example from the community that built protein-protein interaction maps.

As pointed out, past studies have revealed that the function, essentiality, and evolutionary conservation of genes are linked to the topology of gene networks, including protein-protein interaction networks. However, to the best of our knowledge, their linkage to stoichiometry conservation centrality of each gene has not yet been established.

Previously analyzed networks, such as protein-protein interaction networks, depend on known interactions. Therefore, as our understanding of the molecular interactions evolves with new findings, the conclusions may change. Furthermore, analysis of a particular interaction network cannot account for effects from different types of interactions or multilayered regulations affecting each protein species.

In contrast, the stoichiometry conservation network in this study focuses solely on expression patterns as the net result of interactions and regulations among all types of molecules in cells. Consequently, the stoichiometry conservation networks are not affected by the detailed knowledge of molecular interactions and naturally reflect the global effects of multilayered interactions. Additionally, stoichiometry conservation networks can easily be obtained for non-model organisms, for which detailed molecular interaction information is usually unavailable. Therefore, analysis with the stoichiometry conservation network has several advantages over existing methods from both biological and technical perspectives.

We added a paragraph explaining this important point to the Discussion section, along with additional literature.

Finally, the paper built a lot of "machinery" to connect ¥Omega_LE, built directly from proteome, and ¥Omega_B, built from Raman, spaces. I am unsure how that helps and have not been able to digest the 50 or so pages devoted to this.

The mathematical analyses in the supplementary materials form the basis of the argument in the main text. Without the rigorous mathematical discussions, Fig. 6E — one of the main conclusions of this study — and Fig. 7 could never be obtained. Therefore, we believe the analyses are essential to this study. However, we clarified why each analysis is necessary and significant in the corresponding sections of the Results to improve the manuscript's readability.

Please see our responses to comments (2) and (7) by Reviewer #1 (Recommendations for the authors) and comments (5) and (6) by Reviewer #2 (Recommendations for the authors).

Strengths:The rigorous analysis of the data is the real strength of the paper. Alongside this, the discovery of SCGs that are condition-independent and that are condition-dependent provides a great framework.Weaknesses:Overall, I think it is an exciting advance but some work is needed to present the work in a more accessible way.

We edited the main text to make it more accessible to a broader audience. Please see our responses to comments (2) and (7) by Reviewer #1 (Recommendations for the authors) and comments (5) and (6) by Reviewer #2 (Recommendations for the authors).

**Reviewer #1 (Recommendations for the authors):**
(1) The Raman spectral data is measured from single-cell imaging. In the current work, most of the conclusions are from averaged data. From my understanding, once the correspondence between LDA and proteome data is established (i.e. the matrix B) one could infer the single-cell proteome composition from B. This would provide valuable information on how proteome composition fluctuates at the single-cell level.

We can calculate single-cell proteomes from single-cell Raman spectra in the manner suggested by the reviewer. However, we cannot evaluate the accuracy of their estimation without single-cell proteome data under the same environmental conditions. Likewise, we cannot verify variations of estimated proteomes of single cells. Since quantitatively accurate single-cell proteome data is unavailable, we concluded that addressing this issue was beyond the scope of this study.

Nevertheless, we agree with the reviewer that investigating how proteome composition fluctuates at the single-cell level based on single-cell Raman spectra is an intriguing direction for future research. In this regard, some of our authors have studied the phenotypic heterogeneity of *Staphylococcus aureus* cells using single-cell Raman spectra in another paper (Kamei et al., bioRxiv, doi: 10.1101/2024.05.12.593718), and one of our authors has demonstrated that single-cell RNA sequencing profiles can be inferred from Raman images of mouse cells (Kobayashi-Kirschvink et al., Nat. Biotechnol. 42, 1726–1734, 2024). Therefore, it is highly plausible that single-cell Raman spectroscopy can also characterize proteomic fluctuations in single cells. We have added a paragraph to the Discussion section to highlight this important point.

(2) The establishment of matrix B is quite confusing for readers who only read the main text. I suggest adding a flow chart in Figure 1 to explain the data analysis pipeline, as well as state explicitly what is the dimension of B, LDA matrix, and proteome matrix.

We thank the reviewer for the suggestion. Following the reviewer's advice, we have explicitly stated the dimensions of the vectors and matrices in the main text. We have also added descriptions of the dimensions of the constructed spaces. Rather than adding another flow chart to Figure 1, we added a new table (Table 1) to explain the various symbols representing vectors and matrices, thereby improving the accessibility of the explanation.

(3) One of the main contributions for this work is to demonstrate how proteome stoichiometry is regulated across different conditions. A total of m=15 conditions were tested in this study, and this limits the rank of LDA matrix as 14. Therefore, maximally 14 "modes" of differential composition in a proteome can be detected.As a general reader, I am wondering in the future if one increases or decreases the number of conditions (say m=5 or m=50) what information can be extracted? It is conceivable that increasing different conditions with distinct cellular physiology would be beneficial to "explore" different modes of regulation for cells. As proof of principle, I am wondering if the authors could test a lower number (by sub-sampling from m=15 conditions, e.g. picking five of the most distinct conditions) and see how this would affect the prediction of proteome stoichiometry inference.

We thank the reviewer for bringing an important point to our attention. To address the issue raised, we conducted a new subsampling analysis (Fig. S14).

As we described in the main text (Fig. 6E) and the supplementary materials, the *m* x *m* orthogonal matrix, Θ, represents to what extent the two spaces Ω_LE_ and Ω_B_ are similar (*m* is the number of conditions; in our main analysis, *m* = 15). Thus, the low-dimensional correspondence between the two spaces connected by an orthogonal transformation, such as an *m*-dimensional rotation, can be evaluated by examining the elements of the matrix Θ. Specifically, large off-diagonal elements of the matrix mix higher dimensions and lower dimensions, making the two spaces spanned by the first few major axes appear dissimilar. Based on this property, we evaluated the vulnerability of the low-dimensional correspondence between Ω_LE_ and Ω_B_ to the reduced number of conditions by measuring how close Θ was to the identity matrix when the analysis was performed on the subsampled datasets.

In the new figure (Fig. S14), we first created all possible smaller condition sets by subsampling the conditions. Next, to evaluate the closeness between the matrix Θ and the identity matrix for each smaller condition set, we generated 10,000 random orthogonal matrices of the same size as . We then evaluated the probability of obtaining a higher level of low-dimensional correspondence than that of the experimental data by chance (see section 1.8 of the Supplementary Materials). This analysis was already performed in the original manuscript for the non-subsampled case (*m* = 15) in Fig. S9C; the new analysis systematically evaluates the correspondence for the subsampled datasets.

The results clearly show that low-dimensional correspondence is more likely to be obtained with more conditions (Fig. S14). In particular, when the number of conditions used in the analysis exceeds five, the median of the probability that random orthogonal matrices were closer to the identity matrix than the matrix Θ calculated from subsampled experimental data became lower than 10^-4^. This analysis provides insight into the number of conditions required to find low-dimensional correspondence between Ω_LE_ and Ω_B_.

What conditions are used in the analysis can change the low-dimensional structures of Ω_LE_ and Ω_B_ . Therefore, it is important to clarify whether including more conditions in the analysis reduces the dependence of the low-dimensional structures on conditions. We leave this issue as a subject for future study. This issue relates to the effective dimensionality of omics profiles needed to establish the diverse physiological states of cells across conditions. Determining the minimum number of conditions to attain the condition-independent low-dimensional structures of Ω_LE_ and Ω_B_ would provide insight into this fundamental problem. Furthermore, such an analysis would identify the range of applications of Raman spectra as a tool for capturing macroscopic properties of cells at the system level.

We now discuss this point in the Discussion section, referring to this analysis result (Fig. S14). Please also see our reply to the comment (1) by Reviewer #2 (Recommendations for the authors).

(4) In *E. coli* cells, total proteome is in mM concentration while the total metabolites are between 10 to 100 mM concentration. Since proteins are large molecules with more functional groups, they may contribute to more Raman signal (per molecules) than metabolites. Still, the meaningful quantity here is the "differential Raman signal" with different conditions, not the absolute signal. I am wondering how much percent of differential Raman signature are from proteome and how much are from metabolome.

It is an important and interesting question to what extent changes in the proteome and metabolome contribute to changes in Raman spectra. Though we concluded that answering this question is beyond the scope of this study, we believe it is an important topic for future research.

Raman spectral patterns convey the comprehensive molecular composition spanning the various omics layers of target cells. Changes in the composition of these layers can be highly correlated, and identifying their contributions to changes in Raman spectra would provide insight into the mutual correlation of different omics layers. Addressing the issue raised by the reviewer would expand the applications of Raman spectroscopy and highlight the advantage of cellular Raman spectra as a means of capturing comprehensive multi-omics information.

We note that some studies have evaluated the contributions of proteins, lipids, nucleic acids, and glycogen to the Raman spectra of mammalian cells and how these contributions change in different states (e.g., Mourant et al., J Biomed Opt, 10(3), 031106, 2005). Additionally, numerous studies have imaged or quantified metabolites in various cell types (see, for example, Cutshaw et al., Chemical Reviews, 123(13), 8297–8346, 2023, for a comprehensive review). Extending these approaches to multiple omics layers in future studies would help resolve the issue raised by the reviewer.

(5) It is known that *E. coli* cells in different conditions have different cell sizes, where cell width increases with carbon source quality and growth rate. Does this effect be normalized when processing the Raman signal?

Each spectrum was normalized by subtracting the average and dividing it by the standard deviation. This normalization minimizes the differences in signal intensities due to different cell sizes and densities. This information is shown in the Materials and Methods section of the Supplementary Materials.

(6) I have a question about interpretation of the centrality index. A higher centrality indicates the protein expression pattern is more aligned with the "mainstream" of the other proteins in the proteome. However, it is possible that the proteome has multiple" mainstream modes" (with possibly different contributions in magnitudes), and the centrality seems to only capture the "primary mode". A small group of proteins could all have low centrality but have very consistent patterns with high conservation of stoichiometry. I wondering if the author could discuss and clarify with this.

We thank the reviewer for drawing our attention to the insufficient explanation in the original manuscript. First, we note that stoichiometry conserving protein groups are not limited to those composed of proteins with high stoichiometry conservation centrality. The SCGs 2–5 are composed of proteins that strongly conserve stoichiometry within each group but have low stoichiometry conservation centrality (Fig. 5A, 5K, 5L, and 7A). In other words, our results demonstrate the existence of the "primary mainstream mode" (SCG 1, i.e., the homeostatic core) and condition-specific "non-primary mainstream modes" (SCGs 2–5). These primary and non-primary modes are distinguishable by their position along the axis of stoichiometry conservation centrality (Fig. 5A, 5K, and 5L).

However, a single one-dimensional axis (centrality) cannot capture all characteristics of stoichiometry-conserving architecture. In our case, the "non-primary mainstream modes" (SCGs 2–5) were distinguished from each other by multiple csLE axes.

To clarify this point, we modified the first paragraph of the section where we first introduce csLE (Revealing global stoichiometry conservation architecture of the proteomes with csLE). We also added a paragraph to the Discussion section regarding the condition-specific SCGs 2–5.

(7) Figures 3, 4, and 5A-I are analyses on proteome data and are not related to Raman spectral data. I am wondering if this part of the analysis can be re-organized and not disrupt the mainline of the manuscript.

We agree that the structure of this manuscript is complicated. Before submitting this manuscript to eLife, we seriously considered reorganizing it. However, we concluded that this structure was most appropriate because our focus on stoichiometry conservation cannot be explained without analyzing the coefficients of the Raman-proteome correspondence using COG classification (see Fig. 3; note that Fig. 3A relates to Raman data). This analysis led us to examine the global stoichiometry conservation architecture of proteomes (Figs. 4 and 5) and discover the unexpected similarity between the low-dimensional structures of Ω_LE_ and Ω_B_

Therefore, we decided to keep the structure of the manuscript as it is. To partially resolve this issue, however, we added references to Fig. S1, the diagram of this paper’s mainline, to several places in the main text so that readers can more easily grasp the flow of the manuscript.

(8) Supplementary Equation (2.6) could be wrong. From my understanding of the coordinate transformation definition here, it should be [w1 ... ws] X : = RHS terms in big parenthesis.

We checked the equation and confirmed that it is correct.

**Reviewer #2 (Recommendations for the authors):**
(1) The first main result or linear map between raman and proteome linked via B is intriguing in the sense that the map is condition-independent. A speculative question I have is if this relationship may become more complex or have more condition-dependent corrections as the number of conditions goes up. The 15 or so conditions are great but it is not clear if they are often quite restrictive. For example, they assume an abundance of most other nutrients. Now if you include a growth rate decrease due to nitrogen or other limitations, do you expect this to work?

In our previous paper (Kobayashi-Kirschvink et al., Cell Systems 7(1): 104–117.e4, 2018), we statistically demonstrated a linear correspondence between cellular Raman spectra and transcriptomes for fission yeast under 10 environmental conditions. These conditions included nutrient-rich and nutrient-limited conditions, such as nitrogen limitation. Since the Raman-transcriptome correspondence was only statistically verified in that study, we analyzed the data from the standpoint of stoichiometry conservation in this study. The results (Fig. S11 and S12) revealed a correspondence in lower dimensions similar to that observed in our main results. In addition, similar correspondences were obtained even for different *E. coli* strains under common culture conditions (Fig. S11 and S12). Therefore, it is plausible that the stoichiometry-conservation low-dimensional correspondence between Raman and gene expression profiles holds for a wide range of external and internal perturbations.

We agree with the reviewer that it is important to understand how Raman-omics correspondences change with the number of conditions. To address this issue, we examined how the correspondence between Ω_LE_ and Ω_B_ changes by subsampling the conditions used in the analysis. We focused on , which was introduced in Fig. 5E, because the closeness of Θ to the identity matrix represents correspondence precision. We found a general trend that the low-dimensional correspondence becomes more precise as the number of conditions increases (Fig. S14). This suggests that increasing the number of conditions generally improves the correspondence rather than disrupting it.

We added a paragraph to the Discussion section addressing this important point. Please also refer to our response to Comment (3) of Reviewer #1 (Recommendations for the authors).

(2) A little more explanation in the text for 3C/D would help. I am imagining 3D is the control for 3C. Minor comment - 3B looks identical to S4F but the y-axis label is different.

We thank the reviewer for pointing out the insufficient explanation of Fig. 3C and 3D in the main text. Following this advice, we added explanations of these plots to the main text. We also added labels ("ISP COG class" and "non-ISP COG class") to the top of these two figures.

Fig. 3B and S4F are different. For simplicity, we used the Pearson correlation coefficient in Fig. 3B. However, cosine similarity is a more appropriate measure for evaluating the degree of conservation of abundance ratios. Thus, we presented the result using cosine similarity in a supplementary figure (Fig. S4F). Please note that each point in Fig. S4F is calculated between proteome vectors of two conditions. The dimension of each proteome vector is the number of genes in each COG class.

(3) Can we see a log-log version of 4C to see how the low-abundant proteins are behaving? In fact, the same is in part true for Figure 3A.

We added the semi-log version of the graph for SCG1 (the homeostatic core) in Fig. 4C to make low-abundant proteins more visible. Please note that the growth rates under the two stationary-phase conditions were zero; therefore, plotting this graph in log-log format is not possible.

Fig. 3A cannot be shown as a log-log plot because many of the coefficients are negative. The insets in the graphs clarify the points near the origin.

(4) In 5L, how should one interpret the other dots that are close to the center but not part of the SCG1? And this theme continues in 6ACD and 7A.

The SCGs were obtained by setting a cosine similarity threshold. Therefore, proteins that are close to SCG 1 (the homeostatic core) but do not belong to it have a cosine similarity below the threshold with any protein in SCG 1. Fig. 7 illustrates the expression patterns of the proteins in question.

(5) Finally, I do not fully appreciate the whole analysis of connecting ¥Omega_csLE and ¥Omega_B and plots in 6 and 7. This corresponds to a lot of linear algebra in the 50 or so pages in section 1.8 in the supplementary. If the authors feel this is crucial in some way it needs to be better motivated and explained. I philosophically appreciate developing more formalism to establish these connections but I did not understand how this (maybe even if in the future) could lead to a new interpretation or analysis or theory.

The mathematical analyses included in the supplementary materials are important for readers who are interested in understanding the mathematics behind our conclusions. However, we also thought these arguments were too detailed for many readers when preparing the original submission and decided to show them in the supplemental materials.

To better explain the motivation behind the mathematical analyses, we revised the section “Representing the proteomes using the Raman LDA axes”.

Please also see our reply to the comment (6) by Reviewer #2 (Recommendations for the authors) below.

(6) Along the lines of the previous point, there seems to be two separate points being made: (a) there is a correspondence between Raman and proteins, and (b) we can use the protein data to look at centrality, generality, SCGs, etc. And the two don't seem to be linked until the formalism of ¥Omegas?

The reviewer is correct that we can calculate and analyze some of the quantities introduced in this study, such as stoichiometry conservation centrality and expression generality, without Raman data. However, it is difficult to justify introducing these quantities without analyzing the correspondence between the Raman and proteome profiles. Moreover, the definition of expression generality was derived from the analysis of Raman-proteome correspondence (see section 2.2 of the Supplementary Materials). Therefore, point (b) cannot stand alone without point (a) from its initial introduction.

To partially improve the readability and resolve the issue of complicated structure of this manuscript, we added references to Fig. S1, which is a diagram of the paper’s mainline, to several places in the main text. Please also see our reply to the comment (7) by Reviewer #1 (Recommendations for the authors).